# HiFo-Prompt: Prompting with Hindsight and Foresight for LLM-based Automatic Heuristic Design

**Chentong Chen**[1]*, **Mengyuan Zhong**[1]*, **Ye Fan**[2], **Jialong Shi**[1]†, **Jianyong Sun**[1]†

[1]Xi'an Jiaotong University
[2]Northwest Polytechnic University
{chengtong.chen, my.zhong}@stu.xjtu.edu.cn,
fanye@nwpu.edu.cn, {jialong.shi, jy.sun}@xjtu.edu.cn

## Abstract

This paper investigates the application of Large Language Models (LLMs) in Automated Heuristic Design (AHD), where their integration into evolutionary frameworks reveals a significant gap in global control and long-term learning. We propose the Hindsight-Foresight Prompt (HiFo-Prompt), a novel framework for LLM-based AHD designed to overcome these limitations. This is achieved through two synergistic strategies: **Foresight** and **Hindsight**. Foresight acts as a high-level meta-controller, monitoring population dynamics(e.g., stagnation and diversity collapse) to switch the global search strategy between exploration and exploitation explicitly. Hindsight builds a persistent knowledge base by distilling successful design principles from past generations, making this knowledge reusable. This dual mechanism ensures that the LLM is not just a passive operator but an active reasoner, guided by a global plan (Foresight) while continuously improving from its cumulative experience (Hindsight). Empirical results demonstrate that HiFo-Prompt significantly outperforms a comprehensive suite of state-of-the-art AHD methods, discovering higher-quality heuristics with substantially improved convergence speed and query efficiency. Our code is available at https://github.com/Challenger-XJTU/HiFo-Prompt.

## 1 Introduction

Combinatorial Optimization (CO) problems, which involve finding an optimal solution from a discrete set of possibilities, are ubiquitous in science and engineering. Because of their NP-hardness, designing effective heuristics for these problems is a complex task, traditionally based on extensive human experience and intuition (Camacho-Villalón et al., 2023).

The advent of Large Language Models (LLMs) has catalyzed a paradigm shift toward Automated Heuristic Design (AHD) (Wang & Chen, 2023; Liu et al., 2024c). A particularly potent approach marries LLMs with Evolutionary Computation (EC), casting the LLM as a high-level semantic mutation operator. Foundational works such as FunSearch (Romera-Paredes et al., 2024) and EoH (Liu et al., 2024b) established the viability of this LLM+EC paradigm, demonstrating its capacity to discover novel and effective heuristics.

However, as the field progresses, two fundamental challenges have emerged in AHD: *the inability to steer the heuristic generation process based on population dynamics* and *the failure to distill and manage the core design principles of high-performance heuristics to guide the subsequent heuristic generation process*.

First, many approaches lack a mechanism for global adaptive guidance. They often rely on local or reactive signals; for instance, ReEvo (Ye et al., 2024) performs reflection on a single candidate, while methods such as MCTS-AHD (Zheng et al., 2025) passively embed the exploration-

---

*Equal contribution.
†Corresponding author.

exploitation trade-off within their search structure. This localized control does not respond to the macroscopical dynamics of the population and cannot proactively intervene when the search encounters systemic issues such as premature convergence or a decline in diversity. A more aggressive strategy involves in-weight adaptation (e.g., EvoTune (Šurina et al., 2025), CALM (Huang et al., 2025)), which uses numerical gradients to fine-tune the LLM. Although powerful, this approach incurs high computational costs from repeated fine-tuning and reduces the LLM to an opaque policy network. Consequently, learning is implicitly encoded in model weights, preventing the extraction of explicit design principles and obscuring the model's symbolic reasoning process.

Second, existing frameworks suffer from poor knowledge persistence, a phenomenon we term knowledge decay. Successful design strategies often remain entangled within specific code implementations; when parent candidates are discarded, the underlying logic is lost. Recent advances, such as evolving the optimizer in MoH (Shi et al., 2025) or automating problem reduction in RedAHD (Thach et al., 2025), operate in orthogonal dimensions. They do not explicitly decouple algorithmic knowledge from executable forms. Consequently, the system fails to achieve cumulative learning, perpetually rediscovering similar concepts instead of building on proven algorithmic strategies.

To overcome these fundamental limitations, we propose **HiFo-Prompt (Hindsight-Foresight Prompt)**, a framework that establishes a hierarchical control architecture for LLM-based AHD. HiFo-Prompt elevates the LLM from a mere code generator to a symbolic meta-optimizer by endowing it with two synergistic capabilities: First, the **Foresight** module addresses the control problem by serving as a meta-controller(**Evolutionary Navigator**) that observes population dynamics. Upon detecting states like performance plateaus, it explicitly modulates the generative process by switching evolutionary regimes via transparent verbal gradients injected at the prompt level, serving as a symbolic alternative to opaque and expensive numerical gradients. Second, the Hindsight module addresses knowledge decay by implementing the **Insight Pool**, an evolving repository of abstracted design principles. It distills core algorithmic patterns from specific code implementations, transforming them into reusable knowledge. This mechanism allows the system to build on validated design principles, effectively seeding subsequent generations with proven algorithmic strategies. In summary, the contributions of our approach are as follows:

- We introduce HiFo-Prompt, a novel framework consisting of Hindsight and Foresight modules. It dynamically generates prompts for LLMs by decoupling thoughts from code, thereby enabling independent updates and evaluations. This mechanism leads to a significant reduction in both training time for heuristics and evaluation costs for LLMs.

- To improve the Hindsight and Foresight abilities of our method, we introduce the Insight Pool and Evolutionary Navigator, respectively. The Insight Pool accumulates knowledge from high-performing codes through iterative updates. The Evolutionary Navigator controls population states by monitoring evolution and balancing exploration-exploitation dynamics.

- We evaluated the heuristics designed by HiFo-Prompt on complex optimization tasks, comparing them against advanced handcrafted heuristics and existing AHD approaches. Our results achieve state-of-the-art performance in the AHD domain, with substantial improvements over prior AHD methods, particularly excelling in the Traveling Salesman Problem (TSP) and Flow Shop Scheduling Problem (FSSP).

## 2 RELATED WORK

**LLM-driven Automatic Heuristic Design**  The integration of Large Language Models (LLMs) into Evolutionary Computation (EC) is a vibrant new direction for Automated Heuristic Design (AHD) (Liu et al., 2024a; Chauhan et al., 2025). Pioneered by works like FunSearch (Romera-Paredes et al., 2024) and EoH (Liu et al., 2024b), this paradigm leverages the LLM as a powerful semantic operator to generate heuristics as code. Recent efforts to advance this paradigm can be categorized along several axes. Some works focus on refining search control through sophisticated prompt engineering and guidance mechanisms (Ye et al., 2024; Dat et al., 2025), or by redesigning the population structure itself (Zheng et al., 2025). A distinct, more model-centric approach directly adapts the LLM's parameters via reinforcement learning-based fine-tuning (Šurina et al.,

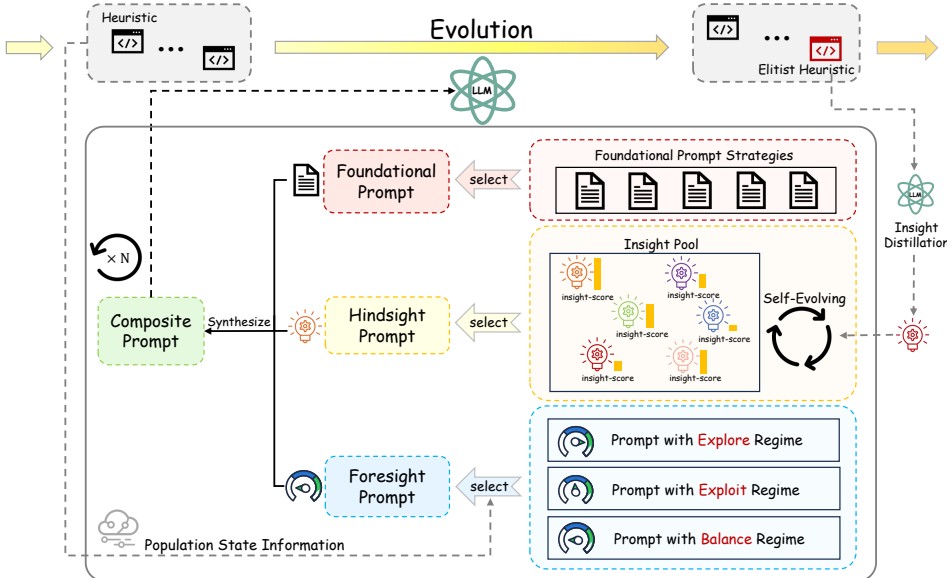

Figure 1: The framework of HiFo-Prompt, which comprises two core processes. (1) **Prompt Construction**: A foundational prompt is dynamically augmented with design directives from the *Foresight* and insights from the *Hindsight* to form the final composite prompt. (2) **Knowledge Evolution**: A self-evolving loop is established where elite heuristics from the evolutionary process are distilled into new insights, continuously enriching the Hindsight module's knowledge base.

2025; Huang et al., 2025). At the highest level of abstraction, research has also explored evolving other core components of the optimization process, such as the optimizer (Shi et al., 2025) or the problem representation (Thach et al., 2025).

**Knowledge Management in Generative Search**   Harnessing historical information is a cornerstone of efficient search. In classical EC, methods like Cultural Algorithms (Maheri et al., 2021) formalize this via a structured Belief Space that stores and evolves collective knowledge. Contemporary LLM-based approaches often rely on in-context reflection mechanisms (Shinn et al., 2023; Bo et al., 2024), where the model self-critiques failures to inform its next attempt. However, this knowledge is typically transient, unstructured, and instance-specific. Consequently, these methods lack a mechanism for accumulating and generalizing insights over time, preventing the formation of a persistent, structured knowledge base analogous to those in classical EC.

**Adaptive Control in Evolutionary Computation**   Dynamically adapting search strategies is a long-standing goal in EC. Historically, this has been addressed through low-level, reactive mechanisms like Adaptive Operator Selection (AOS) (Álvaro Fialho, 2010; Tian et al., 2023) and parameter control (Eiben & Smith, 2015; Aleti & Moser, 2016). These methods rely on numerical credit assignment, effectively voting for strategies that recently performed well, but they lack a semantic understanding of the search dynamics (e.g., identifying population stagnation). The reasoning capabilities of LLMs offer a paradigm shift towards higher-level, proactive control (Eiben et al., 1999; Papa, 2021). Instead of merely adjusting parameters, LLMs can interpret population-level statistics to suggest symbolic actions, such as *increase mutation rate to escape a local optimum*. This marks a transition from fine-grained numerical tuning to semantic-based strategic adjustment.

## 3   METHODOLOGY

In this section, we introduce our proposed HiFo-Prompt, a novel framework that equips the evolutionary process driven by LLM with mechanisms for learning and adaptation (shown in Figure 1). HiFo-Prompt integrates two synergistic components: a Foresight module for real-time adaptive control, which monitors evolutionary dynamics to steer the search strategy, and a Hindsight module

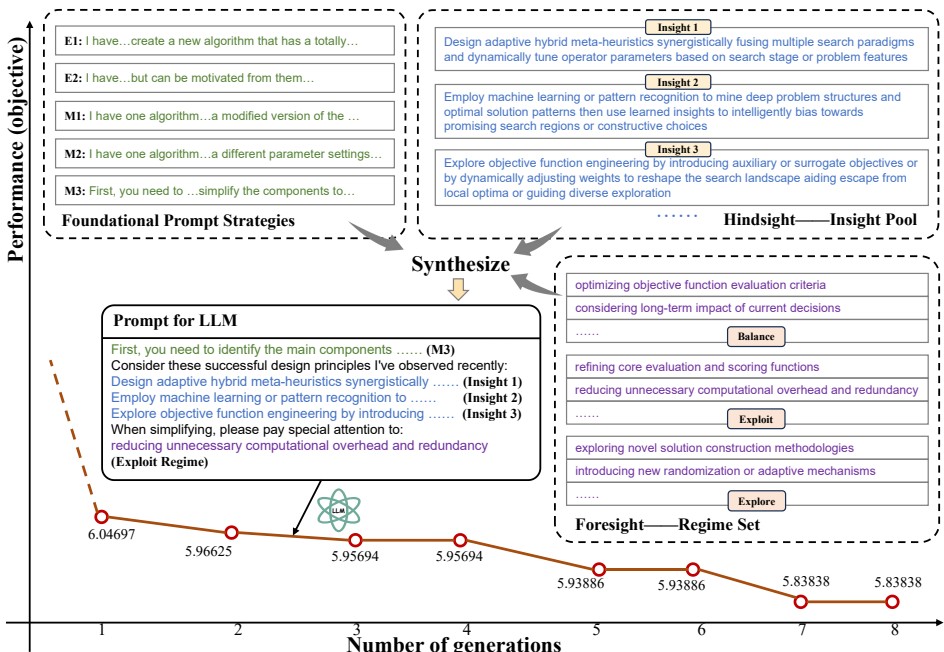

Figure 2: Dynamic prompt generation process of HiFo-Prompt.

for long-term knowledge accumulation, which manages a self-evolving repository of successful design principles that we term insights. By uniting foresight-driven strategy with hindsight-informed knowledge, our framework transforms the generative process into a robust, self-regulating system.

## 3.1 GUIDED PROMPT SYNTHESIS FOR AHD

Our HiFo-Prompt framework applies a Guided Prompt Synthesis mechanism that constructs each prompt as a context-aware composite instruction. This mechanism integrates three interlocking modules: Foundational prompt strategies, Hindsight, and Foresight. The resulting composite prompt provides precise, multifaceted guidance to the LLM. A complete example of prompt generation for TSP with step-by-step construction is provided in Figure 2, and the specific templates for all operators (e.g., I1, E1, M1) are detailed in Appendix E.

**Foundational prompt strategies.** Our framework's generative foundation is a set of Foundational Prompt Strategies, which function as the LLM-equivalent of genetic operators. We first generate heuristics from scratch using the initial prompt strategy **I1**, then evolve them with five foundational prompts adapted from EoH (Liu et al., 2024b). These prompts are organized into two primary strategies: 1) **Reorganization Strategies**, which include **E1**, synthesizing a new algorithm with a novel structure from multiple parents, and **E2**, abstracting shared core ideas to generate conceptually distinct variants; and 2) **Mutation Strategies**, which encompass **M1**, making structural modifications for functionally equivalent variants; **M2**, tuning critical parameters; and **M3**, simplifying components prone to overfitting. While this curated set provides the raw generative capability, its effectiveness depends on contextual guidance. This is the role of the Hindsight and Foresight modules, which inject insights and a design directive into the prompts to align each action with the search's current needs.

**Hindsight Module.** This module incorporates valid heuristic experience in the form of insights, which are abstract and generalizable design principles distilled from successful heuristics. These insights are managed in a dynamic Insight Pool, where each is assigned and continuously updated with a credibility score based on its empirical performance. Before generation, high-scoring insights are retrieved and embedded into the prompt. They serve as validated priors to steer the LLM toward

promising designs. While effective for historical guidance, this module cannot address real-time evolutionary needs.

**Foresight Module.** The Foresight module implements real-time evolutionary state control, orchestrated by its core component, the Evolutionary Navigator. The Navigator continuously monitors macroscopic evolutionary indicators, such as performance stagnation and population diversity, to assess the state of the search. Based on this analysis, it selects the governing evolutionary regime for the subsequent generation. This regime dictates the evolutionary search direction by choosing one of three explicit modes (Črepinšek et al., 2013): 1) **Explore**, to foster novelty when diversity is low or progress has stalled; 2) **Exploit**, to refine high-performing solutions when progress is consistent; 3) **Balance**, to maintain a synergistic application of all operators. This directive guides the prompt, ensuring the LLM adapts to the current search phase.

## 3.2 HINDSIGHT: MECHANISMS OF THE SELF-EVOLVING INSIGHT POOL

Initially, HiFo-Prompt initializes the Insight Pool with seed insights(See Section F.3) to bootstrap the generative process. Serving as an initial scaffold, these seeds allow the framework to immediately shift focus to autonomously evolving novel heuristics. This is achieved through a continuous lifecycle that systematically transforms transient evolutionary successes into reusable knowledge assets, governed by three integrated phases: insight extraction, utility-driven application, and adaptive pruning.

Initially, HiFo-Prompt initializes the Insight Pool with seed insights(See Section F.3) to bootstrap the generative process. Serving as an initial scaffold, these seeds allow the framework to immediately shift focus to autonomously evolving novel, problem-specific heuristics. This is achieved through a continuous lifecycle that systematically transforms transient evolutionary successes into explicit, reusable knowledge assets, governed by three integrated phases: insight extraction, utility-driven application, and adaptive pruning.

**Insight Extraction and Admission.** The lifecycle begins by expanding the knowledge base. At the end of each generation, we prompt an LLM to distill generalizable design principles (insights) from the elite individuals of the population (See Section F.1). To preserve informational diversity, a candidate insight $k_{\text{new}}$ is admitted to the insight pool $K_{\text{pool}}$ only if its Jaccard similarity to all existing pool members falls below a novelty threshold $\theta_{\text{novelty}}$. For this comparison, each insight's text is preprocessed by converting it to lowercase and tokenizing it based on whitespace. The insights themselves become active candidates for the guidance of future generations.

**Insight Retrieval and Credit Assignment.** To guide heuristic generation, we employ a utility-based retrieval mechanism. For each new generation attempt, the mechanism selects the top-$s$ insights with the highest adaptive utility score $U(k_i, t)$. This contributing set of insights, denoted as $K_c$, is then injected into the LLM's prompt. Following the evaluation of the offspring generated, we use credit assignment (Whitacre et al., 2006) to update the utility scores of all the insights in $K_c$. The utility function is formulated to balance exploitation and exploration:

$$U(k_i, t) = \underbrace{E_i(t)}_{\text{Effectiveness}} - \underbrace{w_u \log(N_i(t) + 1)}_{\text{Usage Penalty}} + \underbrace{B_r(t, t_i^{\text{last}})}_{\text{Recency Bonus}} \tag{1}$$

where $E_i(t)$ is the learned effectiveness of insight $i$. The penalty term, weighted by $w_u$, discourages overuse by penalizing an insight based on its total retrieval count $N_i(t)$, thus promoting exploration of less-used ideas. The recency bonus $B_r$ offers a temporary reward for insights used recently; specifically, it grants a fixed bonus if the insight was used within a small generation window $T_w$ (that is, if $t - t_i^{\text{last}} \leq T_w$), promoting strategic coherence.

The effectiveness score $E_i(t)$ is updated via credit assignment. This process first converts the raw fitness $g(h_{\text{new}})$ of an offspring into a normalized, problem-agnostic score $\tilde{\rho}$, scaling its performance relative to the best ($g(h_{\text{best}})$) and worst ($g(h_{\text{worst}})$) solutions of the current population:

$$\tilde{\rho} = \frac{g(h_{\text{worst}}) - g(h_{\text{new}})}{g(h_{\text{worst}}) - g(h_{\text{best}}) + \epsilon} \tag{2}$$

where $\epsilon$ is a small constant to prevent division by zero. Drawing upon techniques for handling sparse rewards in Hierarchical Reinforcement Learning, we posit that an offspring's evolutionary contribution is non-linear. Therefore, we map $\tilde{\rho}$ to a final credit signal, $g_{\text{eff}}$, using a tiered, piecewise function. This design creates distinct reward regimes for qualitatively different outcomes:

$$g_{\text{eff}}^{\text{raw}} = \begin{cases} 0.8 + 0.2 \cdot \tilde{\rho} & \text{if } g(h_{\text{new}}) \geq g(h_{\text{best}}) \\ 0.2 + 0.6 \cdot \tilde{\rho} & \text{if } g(h_{\text{best}}) > g(h_{\text{new}}) \geq g(h_{\text{avg}}) \\ -0.3 + 0.5 \cdot \tilde{\rho} & \text{if } g(h_{\text{new}}) < g(h_{\text{avg}}) \end{cases} \tag{3}$$

This structure provides strong positive signals for paradigm-changing improvements ($\tilde{\rho} \geq 1$), moderate rewards for incremental progress ($0 \leq \tilde{\rho} < 1$), and penalties for below-average performance. The allowance of negative credit is a deliberate design choice to accelerate the pruning of detrimental ideas.

This piecewise reward structure incorporates techniques for handling sparse rewards in Hierarchical Reinforcement Learning. It provides discrete and explicit signals for distinct performance levels to enhance interpretability. The coefficients follow a specific asymmetric philosophy where the reward for elite solutions is significantly larger than the penalty magnitude, which is in turn larger than the reward for incremental improvements. This prioritization provides a strong signal for paradigm-shifting breakthroughs while effectively pruning unproductive directions. Furthermore, the parameter sensitivity analysis detailed in Appendix C.7 demonstrates that the framework maintains robust performance across a range of coefficient settings. This confirms that the structural logic of the reward mechanism is more critical than precise tuning.

To ensure stable updates, the raw credit is then clipped to a final value $g_{\text{eff}} = \max(-1.0, \min(1.0, g_{\text{eff}}^{\text{raw}}))$. Finally, for each insight $j$ in the contributing set $K_c$, its effectiveness score is updated through an Exponential Moving Average (EMA):

$$E_j(t+1) = (1 - \alpha) \cdot E_j(t) + \alpha \cdot g_{\text{eff}} \tag{4}$$

where the learning rate $\alpha \in [0, 1]$ smooths the stochastic credit signals from individual evaluations. This allows an insight's long-term utility to emerge statistically, preventing its score from being skewed by outlier performances.

**Adaptive Pruning and Pool Maintenance.** To maintain quality within its finite capacity $C_{\text{pool}}$, the pool employs an adaptive pruning mechanism triggered when $|K_{\text{pool}}| > C_{\text{pool}}$. This process removes the insight with the minimum eviction score, $\mathcal{S}_{\text{evict}}$, which balances the proven performance of an insight against the risk of its obsolescence:

$$\mathcal{S}_{\text{evict}}(k_i, t) = E_i(t) - R_{\text{decay}} \cdot (t - t_{\text{last\_used}}(k_i)) \tag{5}$$

where $E_i(t)$ is the current effectiveness of the insight, $t_{\text{last\_used}}(k_i)$ is the generation of its last use, and the time decay rate $R_{\text{decay}}$ targets not only low-effectiveness insights, but also those that have become inactive. The decay rate is set conservatively to prevent the premature removal of valuable but temporarily dormant knowledge. Additionally, to prevent the premature elimination of under-explored principles, insights with usage counts below a threshold $T_{\text{usage}}$ are exempt from pruning, allowing sufficient iterations to establish their true utility.

### 3.3 FORESIGHT: THE EVOLUTIONARY NAVIGATOR FOR STATE-AWARE GUIDANCE

While the Hindsight module grounds the search in historically validated principles, the Foresight module acts as an **Evolutionary Navigator**, providing real-time, state-aware guidance. Its primary role is to modulate the exploration-exploitation trade-off via a control policy, $\pi : S_t \to \theta_t$. This policy maps the current evolutionary state $S_t$ to a high-level search orientation, denoted as the *Evolutionary Regime* $\theta_t$. Unlike traditional adaptive EAs that solely tune numerical parameters, Foresight establishes a semantic feedback loop that directly steers the conceptual strategy of the LLMs. We define $\pi$ as a rule-based system that selects the appropriate strategic modes based on evolutionary state :

$$\theta_t = \begin{cases} \theta_{\text{explore}} & \text{if } C_{\text{stag}}(t) \geq \tau_{\text{stag}} \text{ or } \Delta_p(t) < \delta_p \\ \theta_{\text{exploit}} & \text{if } C_{\text{prog}}(t) \geq \tau_{\text{prog}} \\ \theta_{\text{balance}} & \text{otherwise} \end{cases} \tag{6}$$

To implement this policy, the Navigator continuously monitors two primary aspects of the evolutionary search: its performance trajectory and its population diversity. The performance trajectory is tracked via two mutually exclusive counters, $C_{\text{prog}}$ and $C_{\text{stag}}$, based on the improvement in the best raw fitness $\Delta g$. If $\Delta g > 10^{-4}$, it is a generation of *progress* (increment $C_{\text{prog}}$, reset $C_{\text{stag}}$); otherwise, it is *stagnation* (increment $C_{\text{stag}}$, reset $C_{\text{prog}}$). Simultaneously, the Navigator assesses population health through a novel measure of phenotypic diversity, $\Delta_p(t)$. This metric is crucial because relying on fitness alone can be misleading; a population of high-fitness but structurally similar individuals often indicates entrapment in a local optimum. To counteract this, our metric quantifies the semantic variety within the population by measuring the dissimilarity of the generated algorithms' textual forms, rather than their fitness values. It is computed as the normalized fraction of unique pairs of algorithms in the population $P$ whose textual descriptions are non-identical:

$$\Delta_p(t) = \frac{1}{|P|(|P|-1)/2} \sum_{i=1}^{|P|} \sum_{j=i+1}^{|P|} \mathbb{I}(\text{alg}_i \neq \text{alg}_j) \tag{7}$$

where $\mathbb{I}(\cdot)$ is the indicator function and $\text{alg}_i$ denotes the textual description of the $i$-th algorithm. Unlike embedding-based metrics, which risk obscuring critical logical alterations (e.g., *dynamic* vs. *static*) through semantic smoothing (Agarwal et al., 2025), this metric relies on exact string matching of the generated algorithmic descriptions. By enforcing a standardized prompt template to eliminate trivial syntactic noise, any remaining lexical difference indicates a deliberate structural modification by the LLM. Consequently, this approach provides a robust and computationally efficient proxy for phenotypic diversity, ensuring that even subtle yet functional changes are preserved.

This hybrid state representation, which marries quantitative performance trends with a qualitative measure of semantic diversity, provides the Navigator with a much more holistic understanding of the search landscape than fitness-based metrics alone can. These state indicators are evaluated against empirically determined thresholds, with their specific values detailed in our experimental setup. The progress and stagnation counters are designed to track immediate performance trends. This focus on recent history is crucial for capturing the rapid dynamics inherent in LLM-based evolution, enabling swift strategic adjustments in response to even short periods of stagnation or consistent improvement. Similarly, the diversity threshold is calibrated to detect a significant collapse in semantic variety. It acts as an early warning mechanism against premature convergence, triggering stronger exploratory pressure when a substantial portion of the population becomes homogeneous. Once the regime $\theta_t$ is determined, it is translated into a natural language **Design Directive**(See Section F.2) that is injected into the LLM's prompt. For instance, $\theta_{\text{explore}}$ might yield a directive to *try a significantly different approach from conventional solutions*, whereas $\theta_{\text{exploit}}$ would instruct the model to *focus on refining and optimizing the most effective patterns*. This mechanism ensures the LLM's generative focus is explicitly aligned with the high-level strategy dictated by the current evolutionary state.

## 4 EXPERIMENTS

In this section, we present the results of heuristics designed by our proposed HiFo-Prompt on different complex tasks, including Traveling Salesman Problem (TSP) (Matai et al., 2010), Online Bin Packing Problem (Online BPP) (Seiden, 2002), Flow Shop Scheduling Problem (FSSP) (Emmons & Vairaktarakis, 2012), and Bayesian Optimization (BO) (Shahriari et al., 2016). Task definitions and details are given in Appendix B. Results on TSP, online BPP, and FSSP are presented in this section, while the results on BO are provided in Appendix C.4, where HiFo-Prompt demonstrates competitive and reliable performance.

**Experimental Settings.** Experiments were conducted on a workstation equipped with an *Intel Core i7-12700 CPU*, employing `Qwen2.5-Max` as the core LLM. The evolutionary process maintains a population size of 8, running for 8 generations on Combinatorial Optimization (CO) tasks and 4 generations on Bayesian Optimization (BO) tasks. Regarding specific module hyperparameters, the **Hindsight Module** maintains an Insight Pool of capacity $C_{\text{pool}} = 30$ (Jaccard threshold 0.7), with retrieval parameters set to selection count $s = 3$, usage penalty $w_u = 0.1$, recency bonus magnitude $\tau_r = 0.2$, EMA rate $\alpha = 0.3$, decay $R_{\text{decay}} = 0.01$, and probation $T_{\text{usage}} = 3$. The **Foresight Module** utilizes Navigator thresholds of $\tau_{\text{stag}} = 3$ (stagnation), $\tau_{\text{prog}} = 2$ (progress), and $\delta_p = 0.3$

Table 1: Results on TSP with step-by-step construction. Gap(%) denotes the performance gap compared to advanced heuristic algorithms. Time(s) represents the running time of the designed heuristics. This result of LLM-based AHD method is the average of three runs. The best-performing LLM-based AHD results are shown in bold.

| Method | TSP50 | | TSP100 | | TSP200 | |
|---|---|---|---|---|---|---|
| | Gap | Time(s) | Gap | Time(s) | Gap | Time(s) |
| LKH3 | 0.000% | 323.3 | 0.000% | 1450 | 0.000% | 6312 |
| POMO | 0.163% | - | 1.636% | - | 13.961% | - |
| LEHD | 0.117% | - | 0.452% | - | 0.367% | - |
| EoH | 12.820% | 1.4 | 15.361% | 9 | 16.658% | 78 |
| ReEvo | 10.239% | 21.5 | 12.577% | 224 | 14.890% | 3013 |
| HSEvo | 10.467% | 89.5 | 12.008% | 1286 | 13.578% | 24835 |
| MCTS-AHD | 10.642% | 91.5 | 12.521% | 1084 | 13.510% | 14521 |
| Ours | **6.625%** | 244.7 | **8.582%** | 1843 | **8.877%** | 16099 |

Table 2: Results on TSP with GLS. Comparison relative to the results of advanced heuristic on TSP100, TSP200 and TSP500. The result of the LLM-based AHD method is the average of three runs. The best results are shown in bold.

| Method | TSP100 | | TSP200 | | TSP500 | |
|---|---|---|---|---|---|---|
| | Gap | Time(s) | Gap | Time(s) | Gap | Time(s) |
| LKH3 | 0.000% | - | 0.000% | - | 0.000% | - |
| EoH | 0.026% | 210.3 | 0.453% | 368.1 | 2.037% | 1100.7 |
| ReEvo | 0.049% | 357.1 | 0.424% | 775.8 | 2.090% | 1103.4 |
| HSEvo | 0.087% | 543.4 | 0.886% | 792.9 | 2.507% | 1105.1 |
| Ours | **0.015%** | 217.0 | **0.382%** | 392.4 | **1.520%** | 1100.8 |

(diversity). To ensure fair comparison, all LLM-based AHD baselines utilize the same underlying LLM, while their hyperparameters follow the settings reported in their original publications.

**Baselines.** To demonstrate the effectiveness of our proposed method in designing heuristics, we introduce several approaches for solving these complex optimization tasks. (1) handcrafted heuristics, e.g., LKH3(Lin & Kernighan, 1973) for TSP, First Fit and Best Fit (Romera-Paredes et al., 2024) for Online BPP, NEH (Nawaz et al., 1983) and NEHFF (Fernandez-Viagas & Framinan, 2014) for FSSP. (2) Neural Combinatorial Optimization (NCO) methods, e.g., POMO (Kwon et al., 2020) and LEHD (Luo et al., 2023) for TSP, PFSPNet_NEH (Pan et al., 2022) for FSSP. (3) LLM-based AHD methods, e.g., Funsearch (Romera-Paredes et al., 2024), EoH (Liu et al., 2024b), ReEvo (Ye et al., 2024), HSEvo (Dat et al., 2025) and MCTS-AHD (Zheng et al., 2025). Most of our test datasets follow EoH. Notably, Funsearch, ReEvo, and HSEvo require a seed function to initialize their populations, while EoH, MCTS-AHD, and our method can run without it.

**Traveling Salesman Problem.** Step-by-Step Construction (Asani et al., 2023) and Guided Local Search (GLS) (Alsheddy et al., 2016) are two different strategies for solving TSP. The detailed procedure of the two strategies can be found in the Appendix B.1. We design key heuristics for these two strategies. Table 1 compares the results of our method with other baselines in step-by-step construction. We evaluated the performance on 100 instances at each of three sizes. To demonstrate performance on out-of-distribution instances, we also conducted experiments on the TSPLib dataset (Reinelt, 1991), and the results can be found in the Appendix C.1. We further evaluated our method on 100 instances of TSP100, TSP200, and TSP500 in GLS. The results are shown in Table 2. The heuristic designed by our method achieves a significant performance improvement compared to LLM-based AHD methods.

**Online Bin Packing Problem.** The key function of Online BPP is a scoring function that outputs a score for each bin, based on the current item's size and the remaining capacities of the bins. We

Table 3: Results on Online BPP. Gap(%) denotes the ratio of excess bins compared to the lower bound on Weibull instances. Obj. represents the value of the objective function. Results with * are from EoH (Liu et al., 2024b). This result of LLM-based AHD method is the average of three runs. The best results are highlighted in bold.

| Method | 5k_C100 | | 5k_C300 | | 5k_C500 | |
|---|---|---|---|---|---|---|
| | Gap | Obj | Gap | Obj | Gap | Obj |
| lower bound | 0.00% | 2006.2 | 0.00% | 1740.2 | 0.00% | 1687.0 |
| First Fit* | 4.40% | - | 4.18% | - | 4.27% | - |
| Best Fit* | 4.08% | - | 3.83% | - | 3.91% | - |
| Funsearch* | 0.80% | 2022.2 | 1.07% | 1758.8 | 1.47% | 1711.8 |
| EoH | 1.02% | 2026.6 | 1.00% | 1757.6 | 1.00% | 1703.9 |
| ReEvo | 0.78% | 2021.8 | 4.47% | 1818.0 | 3.24% | 1741.6 |
| HSEvo | 1.91% | 2044.6 | 5.47% | 1835.4 | 4.39% | 1761.1 |
| MCTS-AHD | 0.99% | 2026.0 | 0.95% | 1756.8 | 0.95% | 1703.0 |
| Ours | **0.69%** | 2020.1 | **0.66%** | 1751.7 | **0.66%** | 1698.1 |

Table 4: Results on FSSP. The results show the comparison of makespan relative to the baseline on Taillard instances. $n$ denotes the number of jobs and $m$ denotes the number of machines. Results with * are from EoH (Liu et al., 2024b). This result of LLM-based AHD method is the average of three runs. The best results are highlighted in bold.

| Method | n20,m10 | n50,m10 | n100,m10 | n20,m20 | n50,m20 | n100,m20 |
|---|---|---|---|---|---|---|
| NEH* | 4.05% | 3.47% | 2.07% | 3.06% | 5.48% | 3.58% |
| NEHFF* | 4.15% | 3.62% | 1.88% | 2.72% | 5.10% | 3.73% |
| PFSPNet_NEH* | 4.04% | 3.48% | 1.72% | 2.96% | 5.05% | 3.56% |
| EoH | 0.31% | 0.29% | 0.23% | 0.20% | 0.84% | 0.94% |
| Ours | **0.17%** | **0.17%** | **0.13%** | **0.10%** | **0.58%** | **0.51%** |

evaluate our method on Weibull BPP instances (Romera-Paredes et al., 2024) following the EoH setting. The results with three scales are shown in Table 3. Our method not only significantly outperforms handcrafted heuristics but also surpasses LLM-based AHD approaches. More results are provided in Appendix C.2.

**Flow Shop Scheduling Problem.** FSSP involves scheduling $n$ jobs on $m$ machines to minimize the makespan, where each job comprises $m$ operations processed in a fixed order. We evaluate the heuristic designed by our method on Taillard instances (Taillard, 1993). The dataset includes instances with 20 to 100 jobs ($n$) and 10 to 20 machines ($m$). Table 4 presents some of the results, with additional results provided in Appendix C.3. Our method performs well on all datasets, consistently delivering strong and reliable results across different scenarios.

**Ablation Study.** To evaluate the contribution of each component in our method, we conducted a series of ablation studies, as shown in Table 5. We separately removed the Hindsight obtained by the insight pool and the Foresight provided by the navigator, and then removed both Hindsight and Foresight entirely. These experiments are conducted on TSP and Online BPP. The design and parameters of the experiments are aligned with those used in the main experiments.

**Discussion.** We provide more experiments and discussion in the appendix. In Appendix C.5, we present comparative studies between our method and EoH in terms of convergence speed. We also evaluate our method across different LLMs to further demonstrate its effectiveness and generality. In Appendix C.6, we conduct a thorough sensitivity analysis of the key hyperparameters involved in our method, providing detailed explanations of their impact on performance. In Appendix C.7, we conduct additional ablation studies and statistical analyses to further investigate the necessity, underlying mechanisms, and stability of the key design components of HiFo-Prompt.

Table 5: Ablations on Insight Pool and Navigator in TSP and Online BPP. The best results are highlighted in bold.

| | TSP | | | | | |
|---|---|---|---|---|---|---|
| | TSP20 | | TSP50 | | TSP100 | |
| | Gap | Obj. | Gap | Obj. | Gap | Obj. |
| w/o Insight Pool | 11.07% | 4.29 | 13.76% | 6.50 | 16.26% | 9.06 |
| w/o Navigator | 5.82% | 4.09 | 10.31% | 6.30 | 11.48% | 8.69 |
| w/o Insight Pool and Navigator | 11.49% | 4.31 | 14.36% | 6.53 | 18.80% | 9.26 |
| HiFo-Prompt | **3.62%** | **4.00** | **6.63%** | **6.09** | **8.58%** | **8.46** |
| | Online BPP | | | | | |
| | 1k, 100 | 5k, 100 | 1k, 300 | 5k, 300 | 1k, 500 | 5k, 500 |
| w/o Insight Pool | 4.33% | 1.27% | 4.07% | 1.22% | 4.14% | 1.29% |
| w/o Navigator | 2.83% | 1.26% | 2.64% | 1.24% | 2.60% | 1.24% |
| w/o Insight Pool and Navigator | 4.53% | 2.19% | 4.30% | 2.01% | 4.26% | 2.05% |
| HiFo-Prompt | **2.19%** | **0.69%** | **2.08%** | **0.66%** | **2.07%** | **0.66%** |

## 5 CONCLUSION

We introduced HiFo-Prompt, a novel evolutionary framework that advances LLM-driven heuristic design through a hierarchical foresight-hindsight prompting mechanism. By synergizing an Evolutionary Navigator for adaptive control with a self-evolving Insight Pool for knowledge reuse, our framework transforms the search process into a closed-loop, self-regulating system. Across diverse optimization benchmarks, HiFo-Prompt consistently outperforms state-of-the-art methods with remarkable sample efficiency, often finding superior solutions using only 200 LLM requests. This work provides not only a powerful method for heuristic automated algorithm design but also a concrete step towards agents that can learn to invent their problem-solving methodologies.

## ACKNOWLEDGMENTS

This work was partially supported by National Natural Science Foundation of China (grants 12571590, 12426305, 62401473), National Key Research and Development Program of China (grant 2022YFA1004201), Tianyuan Fund for Mathematics of the National Natural Science Foundation of China (grant 12426105), Fundamental Research Funds for the Central Universities (grant xzd012024049), National Key Laboratory Fund Project for Space Microwave Communication (grant HTKJ2024KL504010), Shenzhen Science and Technology Program (grant JCYJ20240813150735045) and Key Research and Development Project in Shaanxi Province (grant 2025CY-YBXM-055).

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

# A  PRELIMINARY

## A.1  PROBLEM FORMULATION OF AUTOMATIC HEURISTIC DESIGN

Automatic Heuristic Design (AHD) (Burke et al., 2009; Voudouris et al., 2016) aims to identify an optimal heuristic $h^*$ from a vast search space $\mathcal{H}$ for a given computational task $\mathcal{P}$. This process can be formally expressed as the following optimization problem (Zheng et al., 2025):

$$h^* = \arg\max_{h \in \mathcal{H}} g(h), \tag{8}$$

where $g(h)$ is a fitness function (maximized) that maps a heuristic to a real number, estimating its quality based on its expected performance on a representative set of problem instances $D$. For a task with a minimization objective $f$ (minimized, e.g., minimizing cost or error), this performance metric is defined as:

$$g(h) = \mathbb{E}_{\text{ins} \in D}\left[-f(h(\text{ins}))\right]. \tag{9}$$

This formulation reframes the original task as a maximization problem over the heuristic space $\mathcal{H}$, thereby enabling the search for robust heuristics that yield high-quality solutions across diverse instances.

## A.2 LLM-DRIVEN EVOLUTIONARY COMPUTATION

The LLM-driven Evolutionary Computation (LLM+EC) framewor(Liu et al., 2024b; Romera-Paredes et al., 2024; Yao et al., 2025; Chauhan et al., 2025) casts the AHD problem as an iterative, population-based search process. Let $P^{(t)} = \{h_1^{(t)}, \ldots, h_M^{(t)}\}$ be the population of $M$ heuristics at generation $t$, where each heuristic $h_i^{(t)} \in \mathcal{H}$ is represented by its thought and code. The transition from $P^{(t)}$ to $P^{(t+1)}$ is governed by a stochastic evolutionary kernel $\mathcal{F}$, which is parameterized by a control vector $\theta^{(t)} \in \Theta$:

$$P^{(t+1)} \sim \mathcal{F}(P^{(t)} \mid \theta^{(t)}). \tag{10}$$

Here, the control vector $\theta^{(t)}$ encapsulates contextual information that guides heuristic generation, such as prompting strategies or performance feedback. In the absence of adaptive control, $\theta^{(t)}$ can be considered constant or null, i.e., $\theta^{(t)} = \theta_{\text{const}}$ or $\theta^{(t)} = \varnothing$. The kernel $\mathcal{F}$ comprises two phases:

1. **Generation Phase**: An LLM, acting as a conditional generator $\mathcal{L}$, creates new heuristics through prompted crossover and mutation. A set of parents $h_p \subseteq P^{(t)}$ is selected, and their symbolic representations $\rho(h_p)$, which include the `thought-and-code`, are used. A prompt function $\Pi$ constructs a conditional prompt from $\rho(h_p)$ and $\theta^{(t)}$ (e.g., exploration or modification strategies). The LLM then generates offspring:

$$h_c = \mathcal{L}(\Pi(\rho(h_p), \theta^{(t)})), \quad O^{(t)} = \{h_{c,1}, h_{c,2}, \ldots, h_{c,N}\}, \tag{11}$$

where $N$ is the number of offspring (e.g., $N = \lambda M$, where $\lambda$ is the reproduction rate).

2. **Selection Phase**: The parent and offspring populations are merged into a candidate pool, $U^{(t)} = P^{(t)} \cup O^{(t)}$. A selection operator $\mathcal{S}$ then chooses the top $M$ heuristics from this pool based on the fitness metric $g$ to form the next generation:

$$P^{(t+1)} = \mathcal{S}(U^{(t)}; g). \tag{12}$$

This framework, introduced within the heuristic evolution paradigm, leverages LLM-driven generation to explore the heuristic space and a selection mechanism to exploit high-performing solutions.

## A.3 KNOWLEDGE-AUGMENTED EVOLUTIONARY COMPUTATION

To extend the capabilities of evolutionary algorithms beyond simple adaptive control, a prominent research direction involves incorporating an explicit knowledge component. Cultural Algorithms (Coello & Becerra, 2010; Yan et al., 2017) provide a classic framework for this idea, decoupling the evolutionary system into two interacting spaces: a population space containing candidate solutions $P^{(t)}$ and a belief space $K_t$ serving as a repository of experiential knowledge (Maheri et al., 2021). These two spaces co-evolve through a dual-inheritance communication protocol.

Crucially, knowledge $k_s$ extracted from the belief space $K_t$ becomes part of the high-level control vector $\theta^{(t)}$. This knowledge then guides the generation of offspring via an influence mechanism:

$$h_c \sim \mathcal{L}\big(\Pi(\rho(h_p), \theta^{(t)})\big), \quad \text{where } k_s \subseteq \theta^{(t)}. \tag{13}$$

Currently, the successes of the population are fed back into the belief space. An acceptance function $\mathcal{A}$ identifies and extracts potentially valuable experiences, $\mathcal{A}(P^{(t)})$, from the current population. These are then used by a knowledge update function $\mathcal{U}_K$ to update the belief space:

$$K_{t+1} = \mathcal{U}_K(K_t, \mathcal{A}(P^{(t)})). \tag{14}$$

Knowledge-augmented evolution is thus a coupled dynamical system where the population and knowledge base co-evolve interdependently. This explicit mechanism for knowledge management allows the framework to achieve a more sophisticated form of learning based on abstracted experience.

## B OPTIMIZATION PROBLEM DETAILS

### B.1 TRAVELING SALESMAN PROBLEM

The Traveling Salesman Problem (TSP) (Matai et al., 2010; Voudouris & Tsang, 1999) is a canonical NP-hard combinatorial optimization problem. Given a set of $N$ cities and the distances between

each pair, the objective is to find the shortest possible tour that visits each city exactly once before returning to the starting city.

Formally, let $V = \{v_1, v_2, \ldots, v_N\}$ be the set of cities. We model the problem on a complete, undirected graph $G = (V, E)$, where a non-negative distance $d_{ij}$ is associated with each edge $(v_i, v_j) \in E$. A tour is a permutation $\pi$ of the indices $\{1, 2, \ldots, N\}$. The goal is to find a permutation $\pi^*$ that minimizes the total tour length (Zheng et al., 2024):

$$\min_{\pi} \left( \sum_{i=1}^{N-1} d_{\pi_i, \pi_{i+1}} + d_{\pi_N, \pi_1} \right)$$

where $v_{\pi_i}$ denotes the $i$-th city in the tour.

**Step-by-step Construction.** Construction heuristics build a feasible solution from an empty set by making a sequence of decisions (Glover et al., 2001). At each step, a component is added to the partial solution based on a specific greedy criterion until a complete tour is formed (Martí & Reinelt, 2011). A fundamental example is the Nearest Neighbor (NN) heuristic. Starting from a node $v_{\pi_1}$, it constructs a tour by iteratively selecting the closest unvisited node. At step $t$, with a partial tour $S_t = (v_{\pi_1}, \ldots, v_{\pi_t})$ and the set of unvisited nodes $U_t = V \setminus \{v_{\pi_1}, \ldots, v_{\pi_t}\}$, the next node $v_{\pi_{t+1}}$ is chosen according to the rule (Marinakis, 2024):

$$v_{\pi_{t+1}} = \arg \min_{v_j \in U_t} d_{\pi_t, j}$$

Regarding the integration of our method with the framework, we will discuss this in more detail. While fast, the myopic nature of such simple rules often leads to suboptimal solutions. Our approach, HiFo-Prompt, addresses this limitation by automating the discovery of more sophisticated construction heuristics. We retain the foundational step-by-step framework where a solution is built incrementally.

During training phase, we replace the fixed, handcrafted decision logic with a heuristic function synthesized by an LLM within our evolutionary framework. Crucially, rather than employing the LLM for online, per-step inference during the solving process, our method's core contribution lies in the offline synthesis of a complete, executable heuristic function.

During inference phase, this generated function—not the LLM itself—that is invoked at each step $t$. This function receives the full problem state, including the current partial tour $S_t$, the set of unvisited candidate nodes $U_t$, and the global distance matrix $C$ (Liu et al., 2024b; 2023), and computes the next node to add to the tour locally and efficiently.

**Guided Local Search** Improvement heuristics, such as local search, start with a complete tour and iteratively refine it. Guided Local Search (GLS) is an advanced metaheuristic that enhances local search by introducing a guidance mechanism to escape local optima (Voudouris & Tsang, 1999; Voudouris et al., 2016). GLS achieves this by modifying the objective function with penalties on certain solution features that appear in locally optimal solutions (Wu et al., 2015). For the Traveling Salesperson Problem (TSP), the most natural features to penalize are the edges of the tour (Tairan & Zhang, 2010).

The standard GLS cost function is augmented with a penalty term:

$$L_{\text{aug}}(s) = L(s) + \lambda \sum_{(u,v) \in s} p_{uv} \tag{15}$$

where $L(s)$ is the original tour length, the sum is over all edges $(u, v)$ in tour $s$, $p_{uv}$ is a penalty counter for using the edge between cities $u$ and $v$, and $\lambda$ is a regularization parameter. An efficient implementation involves creating a penalized distance matrix $D'$ for the local search:

$$D'_{uv} = d_{uv} + \lambda \cdot p_{uv} \quad . \tag{16}$$

Minimizing the tour length using $D'$ is equivalent to minimizing $L_{\text{aug}}(s)$. The critical challenge lies in designing the rule for updating the penalty matrix $P = \{p_{uv}\}$. When the local search becomes

trapped in a local optimum $s^*$, a state-dependent update heuristic, $H_{\text{update}}$, is invoked to determine which edges in $s^*$ should be penalized:

$$P_{\text{new}} = H_{\text{update}}(P_{\text{old}}, s^*, D, N) \quad , \tag{17}$$

where $N$ is a matrix of edge usage frequencies. Typically, this heuristic is a static, handcrafted rule that increments the penalties for a subset of edges in $s^*$. This update reshapes the search landscape to guide the search away from the current basin of attraction.

HiFo-Prompt automates the discovery of the update heuristic $H_{\text{update}}$. Instead of relying on a static, human-designed rule, we task an LLM with synthesizing a complete, executable Python function to serve as $H_{\text{update}}$.

At each GLS iteration, after converging to a local optimum $s^*$, this LLM-generated function is invoked. It receives the full state necessary for intelligent penalization—the current penalty matrix ($P_{\text{old}}$), the local optimum tour ($s^*$), the original distance matrix ($D$), and the edge frequency matrix ($N$)—and outputs a new penalty matrix, $P_{\text{new}}$. This updated matrix then modifies the search costs via Eq. 16 for the next major iteration. The local search itself is driven by fundamental prompt strategies like **Relocate** (Tuononen, 2022) and **2-opt** (Sengupta et al., 2019). Our contribution lies not in these prompt strategies but in the automated design of the sophisticated $H_{\text{update}}$ function through HiFo-Prompt, which provides the intelligence to guide these prompt strategies effectively.

We further elevate this concept by embedding heuristic generation within an evolutionary framework. We treat the distinct $H_{\text{update}}$ functions as a population of individuals. The LLM itself serves as the primary genetic operator. Through structured prompts for crossover and mutation, the LLM intelligently combines or modifies existing high-performing strategies to produce novel offspring (Liu et al., 2024b).

## B.2 ONLINE BIN PACKING PROBLEM

The Online Bin Packing Problem (OBP) (Seiden, 2002) is a classic sequential decision-making problem. We are presented with a sequence of items, $A = (a_1, a_2, \ldots, a_T)$, arriving one at a time. Each item $a_t$ has a size $s_t \in (0, 1]$. We have an unlimited supply of bins, each with a unit capacity of 1. The core constraints of the OBP are:

- **Online Constraint:** When item $a_t$ arrives, a decision must be made to place it into a bin without any knowledge of future items ($a_{t+1}, \ldots, a_T$).
- **Irrevocable Placement:** Once an item is placed in a bin, it cannot be moved.
- **Capacity Constraint:** For any bin $B_j$, the sum of the sizes of all items placed within it must not exceed 1.

The objective is to minimize the total number of bins used after placing all $T$ items (Epstein et al., 2012; Yarimcam et al., 2014). If we let $y_j = 1$ if bin $j$ is used and $y_j = 0$ otherwise, the goal is to minimize $\sum_j y_j$ (Sgall, 2014).

Given the online nature of the problem, optimal solutions are generally not achievable. Instead, high-performance algorithms rely on sophisticated greedy placement policies or heuristics. Following the approach of (Romera-Paredes et al., 2024), we frame the task as learning a superior placement heuristic. Specifically, we use the HiFo-Prompt framework to design a scoring function, $H_{\text{score}}$, that determines the most suitable bin for an incoming item.

When an item $a_t$ with size $s_t$ arrives, the HiFo-Prompt-generated heuristic is invoked. It takes as input the item's size and the state of all currently open bins. The state of a bin $B_j$ is captured by its residual capacity, $c_j = 1 - \sum_{a_k \in B_j} s_k$. The scoring function produces a scalar value for each candidate bin:

$$\sigma_j = H_{\text{score}}(s_t, c_j) \tag{18}$$

where $\sigma_j$ represents the desirability of placing item $a_t$ into bin $B_j$. The placement policy is then to assign the item to the valid bin with the highest score:

$$j^* = \underset{\{j \mid c_j \geq s_t\}}{\arg\max} \sigma_j \tag{19}$$

If no existing bin can accommodate the item (i.e., the set of valid bins is empty), a new bin is opened. The intelligence of our method lies in HiFo-Prompt's ability to discover a non-trivial scoring function $H_{\text{score}}$ that implicitly balances competing objectives, such as leaving space for potentially larger future items versus consolidating small items efficiently. This contrasts with classic heuristics like Best Fit (which is equivalent to $H_{\text{score}}(s_t, c_j) = -c_j$) or First Fit, by allowing for a much richer and more adaptive decision-making process.

### B.3 FLOW SHOP SCHEDULING PROBLEM

The Flow Shop Scheduling Problem (FSSP) (Emmons & Vairaktarakis, 2012; Komaki et al., 2019) is a canonical NP-hard scheduling challenge. The task is to schedule a set of $n$ jobs, $J = \{1, \ldots, n\}$, on a series of $m$ machines, $M = \{1, \ldots, m\}$. Each job $i \in J$ requires $m$ operations, with the $j$-th operation occurring on machine $j$. The processing time for job $i$ on machine $j$ is given by $T_{ij}$ from a processing time matrix $T$ (Vaessens et al., 1996; Ribas et al., 2010). A solution (Juan et al., 2014) is a permutation of jobs, $\pi = (\pi_1, \ldots, \pi_n)$, which dictates the processing order on all machines. Key constraints include that no machine can process multiple jobs at once, and no job can be on multiple machines simultaneously.

The objective is to find a permutation $\pi^*$ that minimizes the makespan, $C_{\max}$. This is the total time elapsed until the last job completes its final operation. The completion time $C(\pi_i, j)$ for job $\pi_i$ on machine $j$ is calculated recursively:

$$C(\pi_i, j) = \max\left(C(\pi_{i-1}, j), C(\pi_i, j-1)\right) + T_{\pi_i, j} \tag{20}$$

with base cases $C(\pi_0, j) = 0$ and $C(\pi_i, 0) = 0$. The makespan for a sequence $\pi$ is therefore $C_{\max}(\pi) = C(\pi_n, m)$.

Due to the problem's complexity, we employ a local search metaheuristic (Liu et al., 2024b). To prevent the search from being trapped in local optima, we use the HiFo-Prompt framework to design a sophisticated guidance strategy automatically. When the search converges to a locally optimal sequence $\pi^*$, HiFo-Prompt-generated heuristic, $H_{\text{guide}}$, is invoked. It takes the current sequence, the original processing time matrix, and problem dimensions to produce both a new time matrix $T'$ and a designated list of jobs to perturb, $J_{\text{perturb}}$:

$$(T', J_{\text{perturb}}) = H_{\text{guide}}(\pi^*, T, n, m) \tag{21}$$

This dual output provides a powerful guidance mechanism. The new matrix $T'$ reshapes the search landscape by penalizing attributes of the local optimum, effectively steering the search toward unexplored regions. Concurrently, the list $J_{\text{perturb}}$ directs subsequent local search operators, such as insertions or swaps, to focus their computational effort on a specific subset of critical jobs. This combined strategy of altering the problem's cost structure while focusing on the search operators constitutes a complete and intelligent guidance component, designed automatically by our framework.

### B.4 BAYESIAN OPTIMIZATION

Bayesian Optimization (BO) (Shahriari et al., 2016) and its ongoing development are of paramount importance, as it provides the leading framework for sample-efficiently navigating the complex, high-cost search spaces prevalent in modern science and engineering. Bayesian Optimization (BO) has emerged as a principal framework for this task, excelling in applications like hyperparameter tuning and automated scientific discovery. The power of BO lies in its sample efficiency. It builds a probabilistic surrogate model of the objective function. It then uses an acquisition function (Lam et al., 2016) to intelligently decide where to sample next, thereby minimizing the number of costly evaluations.

Our work addresses a particularly demanding variant: cost-aware BO (Snoek et al., 2012; Yao et al., 2024; Zheng et al., 2025). In this setting, each function evaluation $f(\mathbf{x})$ has a heterogeneous and unknown cost, denoted by $c(\mathbf{x})$. The goal is to find the global maximum of $f(\mathbf{x})$ within a fixed total budget $B_{\text{total}}$. This requires sequentially choosing evaluation points $\{\mathbf{x}_1, \ldots, \mathbf{x}_N\}$ to maximize the outcome, subject to the constraint that $\sum_{i=1}^{N} c(\mathbf{x}_i) \leq B_{\text{total}}$ (Lee et al., 2020). To manage this, the BO

agent maintains two surrogate models, typically Gaussian Processes (GPs) (Rasmussen & Williams, 2006). One GP models the objective function, predicting its posterior mean $\mu_f(\mathbf{x})$ and standard deviation $\sigma_f(\mathbf{x})$. A second GP models the evaluation cost, providing a cost prediction $\mu_c(\mathbf{x})$.

The core intelligence of the agent is encoded in its acquisition function, $\alpha(\mathbf{x})$. This function must navigate a complex trade-off between seeking high rewards (exploitation), reducing model uncertainty (exploration), and managing evaluation costs. At each iteration $t$, the next evaluation point $\mathbf{x}_{t+1}$ is selected by maximizing this utility:

$$\mathbf{x}_{t+1} = \arg\max_{\mathbf{x} \in \mathcal{X}} \alpha(\mathbf{x}|\mathcal{D}_t, B_{\text{rem}}) \tag{22}$$

where $\mathcal{D}_t = \{(\mathbf{x}_i, y_i, c_i)\}_{i=1}^{t}$ is the set of previously evaluated points and $B_{\text{rem}}$ is the remaining budget. The design of $\alpha(\mathbf{x})$ is the single most critical factor for achieving high performance.

We propose to automate the discovery of superior acquisition functions using the HiFo-Prompt framework. Rather than relying on static, human-designed heuristics, HiFo-Prompt generates a novel utility function, $H_{\text{utility}}$, from scratch. This generated function is highly context-aware, synthesizing all critical information available at each decision step. It explicitly considers the surrogate models' predictions, the best-found solution so far ($y_t^* = \max_i y_i$), and the dynamic state of the budget:

$$\alpha(\mathbf{x}) = H_{\text{utility}}\big(\mu_f(\mathbf{x}), \sigma_f(\mathbf{x}), \mu_c(\mathbf{x}), y_t^*, B_{\text{used}}, B_{\text{total}}\big) \tag{23}$$

By generating a holistic function that reasons about the interplay between potential gain, uncertainty, cost, and remaining resources, HiFo-Prompt creates powerful and adaptive sampling strategies. This approach moves beyond hand-crafted designs, enabling superior performance in complex, budget-constrained optimization scenarios.

## C MORE RESULTS

### C.1 TRAVELING SALESMAN PROBLEM

To demonstrate the generality and robustness of the heuristic designed by our proposed method, we conduct a comprehensive evaluation using the real-world benchmark dataset TSPLib (Reinelt, 1991). TSPLib is a well-established collection of TSP instances, widely used in the research community for benchmarking optimization algorithms.

For our experiments, we select a diverse subset of TSP instances from TSPLib, specifically focusing on those containing no more than 500 nodes. This selection criterion ensures a manageable problem scale while still retaining the complexity necessary to assess algorithmic performance effectively.

In our evaluation framework, we adopt a step-by-step construction approach to solve the TSP, which incrementally builds a tour by selecting the next node based on a learned heuristic. This paradigm allows us to evaluate the quality of decisions made at each step and better observe the contribution of the designed heuristic to the final solution quality.

To rigorously assess the effectiveness and competitiveness of our proposed heuristic, we compare its performance against state-of-the-art LLM-based AHD baselines. These methods represent recent advances in leveraging large language models for combinatorial optimization tasks and serve as strong comparative baselines in our study. The experimental results, which include performance metrics, are summarized in Table 6. These results provide empirical evidence that our method not only performs competitively but also generalizes well across a variety of TSP instances in real-world scenarios. In addition, we further conduct experiments on small-scale instances (TSP10 and TSP20) under the step-by-step construction framework for TSP. The results are presented in Table 7.

### C.2 ONLINE BIN PACKING PROBLEM

To provide a more comprehensive empirical validation and to assess the robustness of our framework, we conducted further evaluations of the online Bin Packing Problem (BPP) on a broader and more challenging set of Weibull-distributed instances. These instances, which more closely mimic real-world scenarios than uniform distributions, were generated following the protocol established in prior work on LLM-based heuristic discovery (Romera-Paredes et al., 2024). Table 8 presents a

Table 6: Results on TSPLib. The best results are highlighted in bold.

| name | EoH | MCTS-AHD | ReEvo | HSEvo | Ours |
|------|------|----------|-------|-------|------|
| eil51 | 7.665% | 17.133% | 4.409% | **4.295%** | 6.691% |
| berlin52 | 16.080% | 17.481% | 15.678% | 19.814% | **10.973%** |
| eil76 | 10.566% | 15.193% | 8.450% | 10.034% | **8.439%** |
| pr76 | 25.222% | 29.371% | 21.089% | 21.718% | **20.505%** |
| kroA100 | 24.347% | 24.315% | 17.509% | 18.758% | **16.616%** |
| kroB100 | 30.274% | 22.593% | **9.059%** | 13.266% | 20.502% |
| kroC100 | 25.542% | 10.212% | 27.343% | 36.176% | **10.000%** |
| kroD100 | 31.497% | 25.765% | 25.797% | **13.258%** | 32.315% |
| kroE100 | 24.618% | 21.299% | 23.905% | 22.218% | **14.925%** |
| rd100 | 12.654% | 11.470% | 8.660% | 12.995% | **7.921%** |
| eil101 | 14.421% | 22.947% | 12.163% | 12.372% | **7.229%** |
| lin105 | 40.297% | 16.797% | 27.774% | 42.519% | **13.974%** |
| pr107 | 9.900% | 5.004% | 7.894% | 7.537% | **3.811%** |
| ch130 | 21.406% | **6.930%** | 7.352% | 13.107% | 7.090% |
| ch150 | 19.604% | 9.874% | 12.644% | 10.185% | **4.275%** |
| kroA150 | 27.891% | 20.517% | 25.887% | 21.869% | **20.070%** |
| kroB150 | 22.403% | 27.716% | 29.499% | 32.729% | **13.677%** |
| pr152 | 12.302% | 10.457% | 12.473% | 11.960% | **6.545%** |
| u159 | 13.444% | **7.074%** | 7.951% | 13.644% | 7.828% |
| kroA200 | 26.395% | 26.219% | 27.366% | 28.537% | **22.702%** |
| kroB200 | 20.980% | 20.083% | 23.861% | 21.590% | **14.314%** |
| tsp225 | 32.938% | 20.193% | 21.703% | 22.503% | **18.683%** |
| a280 | 34.081% | 24.095% | 27.921% | 31.736% | **15.936%** |
| rd400 | 14.612% | 14.745% | 13.865% | 15.213% | **8.075%** |
| d493 | 18.833% | 13.248% | 14.776% | 21.179% | **7.976%** |
| avg. | 21.519% | 17.629% | 17.401% | 19.168% | **12.843%** |

Table 7: Results on TSP with step-by-step construction in 10 and 20.

| Method | TSP10 | | TSP20 | |
|--------|-------|---------|-------|---------|
| | Gap | Time(s) | Gap | Time(s) |
| LKH3 | 0.000% | 6.492 | 0.000% | 24.948 |
| POMO | 0.246% | - | 0.248% | - |
| LEHD | 0.183% | - | 0.010% | - |
| EoH | 7.148% | 0.042 | 10.064% | 0.168 |
| ReEvo | 5.227% | 0.228 | 6.811% | 1.215 |
| HSEvo | 5.461% | 0.689 | 7.950% | 3.273 |
| MCTS-AHD | 4.829% | 0.440 | 8.045% | 4.087 |
| Ours | **1.654%** | 0.709 | **3.619%** | 12.961 |

detailed comparative analysis of our method against a wide spectrum of competitors, including both classical, widely-adopted handcrafted heuristics (First Fit, Best Fit) and a suite of contemporary approaches based on Large Language Models.

For each combination of bin capacity and problem size, the dataset includes five unique instances. Performance is quantified by the average percentage gap to the known theoretical lower bound across these instances, where a smaller value signifies a more efficient and effective packing solution.

The empirical results demonstrate the superior performance of the heuristic we have discovered. As shown in the table, our method consistently outperforms all baseline methods in nearly every setting, securing the smallest average gap in 8 out of the 9 distinct configurations tested. This highlights a remarkable level of consistency and dominance. The only exception is the (Capacity=100, Size=10k) case, where Funsearch achieves a marginally lower gap.

Table 8: Results on Online BPP in Weibull instances with varied capacities and problem sizes. Results marked with * denote values taken from (Liu et al., 2024b).

| Capacity | Size | First Fit | Best Fit | EoH | ReEvo | HSEvo | MCTS-AHD | Ours |
|---|---|---|---|---|---|---|---|---|
| 100 | 1k | 5.32% | 4.87% | 3.10% | 3.63% | 3.51% | 3.38% | **2.19%** |
| | 5k | 4.40% | 4.08% | 1.02% | 0.78% | 1.91% | 0.99% | **0.69%** |
| | 10k | 4.44% | 4.09% | 0.80% | **0.35%** | 1.68% | 0.84% | 0.42% |
| 300 | 1k | 4.93% | 4.48% | 3.04% | 7.34% | 6.88% | 3.21% | **2.08%** |
| | 5k | 4.18% | 3.83% | 1.00% | 4.47% | 5.47% | 0.95% | **0.66%** |
| | 10k | 4.20% | 3.87% | 0.78% | 4.05% | 5.26% | 0.85% | **0.39%** |
| 500 | 1k | 4.97% | 4.50% | 3.04% | 5.92% | 5.80% | 3.20% | **2.07%** |
| | 5k | 4.27% | 3.91% | 1.00% | 3.24% | 4.39% | 0.95% | **0.66%** |
| | 10k | 4.28% | 3.95% | 0.78% | 2.79% | 4.14% | 0.85% | **0.40%** |

However, our method's strong performance across the entire parameter space underscores its greater reliability and generalizability compared to methods that may excel only in specific, narrow scenarios. Notably, our approach scales gracefully, achieving extremely low gap values (e.g., 0.41%, 0.42%) on the largest and most complex problems, significantly surpassing both traditional algorithms and other state-of-the-art LLM-driven frameworks. This comprehensive evaluation on challenging instances further validates the efficacy of our evolutionary framework, showcasing its capacity to discover sophisticated and high-performance heuristics that are robust across varied problem characteristics.

## C.3 FLOW SHOP SCHEDULING PROBLEM

Table 9 presents a comparative analysis of the performance of EoH and our proposed method on FSSP of varying scales, defined by the number of jobs $n$ and machines $m$. Performance is evaluated using two key metrics: the objective function value and the relative gap, where the gap is measured with respect to the solution quality of an advanced handcrafted heuristic. A smaller gap reflects a solution that is closer to the heuristic baseline and, therefore, indicates higher solution quality. The results demonstrate that our method consistently achieves smaller gaps across all tested problem configurations, regardless of scale. Moreover, in many cases, it outperforms the advanced heuristic itself in terms of the raw objective value. This consistent superiority suggests that our approach not only generalizes well across diverse FSSP instances (Liu et al., 2024b) but also offers a competitive alternative to domain-specific heuristics, exhibiting both strong effectiveness and robustness.

Table 9: Results on FSSP. The best results are highlighted in bold.

| n | m | EoH Gap | EoH Time(s) | Ours Gap | Ours Time(s) |
|---|---|---|---|---|---|
| 20 | 5 | 0.25% | 7.5 | **-0.01%** | 5.5 |
| | 10 | 0.31% | 13.0 | **0.17%** | 8.8 |
| | 20 | 0.20% | 23.8 | **0.10%** | 18.5 |
| 50 | 5 | 0.01% | 45.1 | **0.00%** | 29.7 |
| | 10 | 0.29% | 83.9 | **0.17%** | 50.6 |
| | 20 | 0.84% | 168.4 | **0.58%** | 101.1 |
| 100 | 5 | -0.02% | 230.2 | **-0.04%** | 146.7 |
| | 10 | 0.23% | 299.6 | **0.13%** | 243.5 |
| | 20 | 0.94% | 305.2 | **0.51%** | 303.0 |
| 200 | 10 | 0.37% | 337.1 | **0.12%** | 317.2 |
| | 20 | 1.23% | 334.6 | **0.71%** | 321.4 |

Table 10: Results on BO. The results denote the absolute error relative to the optimal solution. Results marked with * are from (Yao et al., 2024). The best results are highlighted in bold.

| | Ackley | Rastrigin | Griewank | Rosenbrock | Levy | Three HumpCamel |
|---|---|---|---|---|---|---|
| EI* | 2.66 | 4.74 | 0.49 | **1.26** | 0.01 | 0.05 |
| EIpu* | 2.33 | 5.62 | **0.34** | 2.36 | 0.01 | 0.12 |
| EI-cool* | 2.74 | 5.78 | **0.34** | 2.29 | 0.01 | 0.07 |
| EoH | 3.11 | 3.48 | 0.72 | 2.57 | 0.04 | 0.18 |
| MCTS-AHD | 3.23 | 0.87 | 0.43 | 1.30 | 0.01 | 0.05 |
| Ours | **1.78** | **0.45** | 0.41 | 1.50 | **0.00** | **0.01** |

| | Styblinski Tang | Hartmann | Powell | Shekel | Hartmann | Cosine8 |
|---|---|---|---|---|---|---|
| EI* | 0.03 | **0.00** | 18.89 | 7.91 | **0.03** | 0.47 |
| EIpu* | **0.02** | **0.00** | 19.83 | 7.92 | **0.03** | 0.47 |
| EI-cool* | 0.03 | **0.00** | 14.95 | 8.21 | **0.03** | 0.54 |
| EoH | 2.89 | 0.01 | 13.71 | 8.71 | 0.47 | 1.04 |
| MCTS-AHD | **0.02** | **0.00** | **1.91** | 5.14 | 0.08 | 0.29 |
| Ours | **0.02** | 0.01 | 2.65 | **4.08** | 0.57 | **0.28** |

## C.4 BAYESIAN OPTIMIZATION

Table 10 presents the experimental results on the design of cost-aware acquisition functions (CAFs) (Yao et al., 2024) in Bayesian Optimization. We compare our method against both manually crafted CAFs and those generated by LLM-based AHD methods. To ensure a fair comparison, all LLM-based AHD approaches utilize the same underlying language model. Our method achieves superior performance on the majority of benchmark functions, outperforming traditional CAFs (e.g., EI, EIpu, EI-cool) (Yao et al., 2024) as well as recent LLM-based AHD baselines (e.g., EoH (Liu et al., 2024b), MCTS (Zheng et al., 2025)). The performance advantage is particularly pronounced on challenging functions such as Rastrigin, ThreeHumpCamel, and Shekel. These results underscore the robustness and strong generalization ability of our approach across a wide range of optimization landscapes.

## C.5 COMPARATIVE RESULTS

**Convergence Analysis.** We compared the progression of objective function values during the evolutionary process of our method and EoH on both the TSP and Online BPP. Figure 3 presents the convergence analysis curves. The figure demonstrates that our method converges more rapidly while requiring fewer individuals in the population, indicating higher efficiency. For a detailed breakdown of computational costs, including runtime and token consumption comparisons, please refer to Appendix C.8.

**Results with Different LLMs.** Furthermore, we evaluated it across a diverse set of LLMs. To ensure fairness and consistency, we adopted the evaluation protocol established in MCTS-AHD (Zheng et al., 2025). Specifically, we selected two representative problem scales for each task to validate scalability, and each experiment was conducted across three independent runs, with average values reported to mitigate stochasticity. The detailed comparative results are presented in Table 11.

The results indicate that our framework exhibits strong adaptability, consistently discovering effective heuristics across diverse LLMs. While Qwen-2.5-max yields the leading metrics , we observe that performance does not follow the generic reasoning benchmarks of these models. Instead, success appears to rely on the synergy between the framework's guidance mechanisms and the specific model's characteristics (e.g., instruction following). Even smaller models like GPT-4o-mini show competitive results, highlighting that optimizing the coordination between the LLM-based AHD and the underlying model remains a promising and open direction for future research.

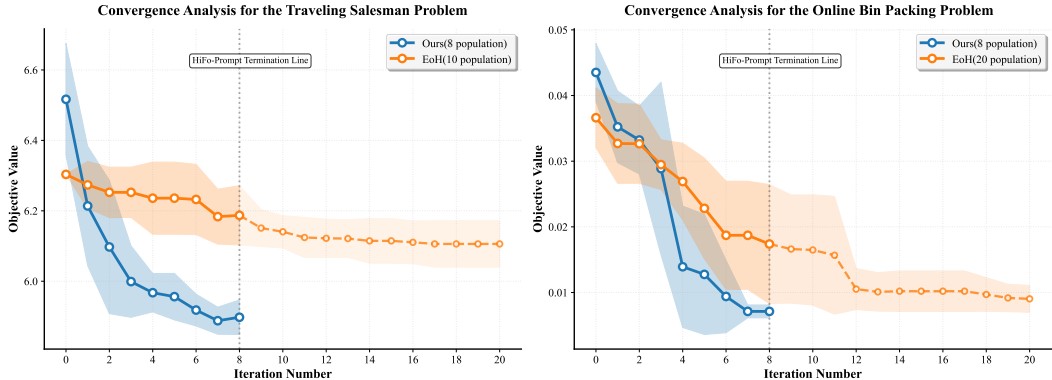

Figure 3: A Key Example of Convergence Analysis for TSP and Online BPP.

Table 11: Results with different LLMs.

| Problems | TSP-construction | | TSP-GLS | |
| --- | --- | --- | --- | --- |
| Scale | N=50 | N=100 | N=100 | N=200 |
| Qwen-2.5-max | 6.625% | 8.582% | 0.015% | 0.382% |
| GPT-4o-mini | 7.058% | 9.579% | 0.021% | 0.433% |
| DeepSeek-v3 | 7.717% | 10.455% | 0.041% | 0.480% |
| DeepSeek-r1 | 10.412% | 12.638% | 0.074% | 0.589% |
| Claude3.5-sonnet | 8.276% | 11.087% | 0.061% | 0.454% |
| Problems | Online BPP | | FSSP-GLS | |
| Scale | N=1k, C=100 | N=10k, C=300 | n=100, m=10 | n=200, m=20 |
| Qwen-2.5-max | 2.188% | 0.391% | 0.128% | 0.710% |
| GPT-4o-mini | 2.287% | 0.492% | 0.137% | 0.787% |
| DeepSeek-v3 | 2.238% | 0.978% | 0.185% | 0.916% |
| DeepSeek-r1 | 2.984% | 1.487% | 0.313% | 1.485% |
| Claude3.5-sonnet | 2.089% | 1.487% | 0.179% | 1.256% |

## C.6 PARAMETER SENSITIVITY ANALYSIS

To rigorously quantify the impact of key hyperparameters on the performance of HiFo-Prompt, we conducted a comprehensive ablation study involving eight core parameters. All sensitivity experiments were conducted on the TSP-50 construction task using the training dataset. To ensure statistical reliability, each configuration was executed across 3 independent runs, with results reported as the average objective function value. Unless otherwise specified, the framework operated under a default configuration of population size $N_{\text{pop}} = 8$ and maximum generations $G_{\text{max}} = 8$. The results confirm that the framework consistently converges to high-quality solutions across a broad spectrum of configurations, demonstrating that our settings are both effective and robust without relying on instance-specific fine-tuning. The results, visualized in Figure 4 and detailed in Table 12, not only justify our default configurations but also reveal the underlying mechanics of the framework. Collectively, these experiments demonstrate that HiFo-Prompt maintains consistent performance across a reasonable range of parameter settings, confirming the robustness and reproducibility of the proposed method.

**Recency Bonus Magnitude** $(\tau_r)$ The role of this parameter is grounded in the formulation of the recency bonus function $B_r$, designed to capture the temporal dynamics of the search process. Formally defined as:

$$B_r(t, t_i^{\text{last}}) = \begin{cases} \tau_r & \text{if } t - t_i^{\text{last}} \leq T_w \\ 0 & \text{otherwise} \end{cases} \tag{24}$$

Table 12: Ablation analysis of **recency bonus magnitude** ($\tau_r$), **usage penalty weight** ($w_u$) and **decay rate** ($R_{\text{decay}}$).

| $w_u$ | run1 | run2 | run3 | avg. |
|---|---|---|---|---|
| 0 | 5.958 | 5.989 | 5.978 | 5.975 |
| 0.05 | 5.894 | 5.901 | 5.907 | 5.901 |
| 0.1 (default) | 5.885 | 5.838 | 5.873 | **5.866** |
| 0.3 | 5.921 | 5.907 | 5.922 | 5.917 |

| $\tau_r$ | run1 | run2 | run3 | avg. |
|---|---|---|---|---|
| 0 | 5.954 | 5.972 | 5.979 | 5.968 |
| 0.1 | 5.920 | 5.925 | 5.901 | 5.915 |
| 0.2 (default) | 5.885 | 5.838 | 5.873 | **5.866** |
| 0.4 | 5.937 | 5.945 | 5.933 | 5.938 |

| $R_{\text{decay}}$ | run1 | run2 | run3 | avg. |
|---|---|---|---|---|
| 0 | 5.998 | 6.097 | 6.075 | 6.057 |
| 0.005 | 5.899 | 5.908 | 5.906 | 5.904 |
| 0.01 (default) | 5.885 | 5.838 | 5.873 | **5.866** |
| 0.04 | 5.985 | 5.956 | 5.967 | 5.969 |

where $t$ denotes the current generation index, $t_i^{\text{last}}$ records the last generation in which insight $i$ was successfully applied, and $T_w$ represents the sliding window size (set to $T_w = 2$). Here, $\tau_r$ controls the magnitude of the reward assigned to recently validated insights. Theoretically, this mechanism addresses the non-stationary nature of evolutionary search—an insight highly effective during early exploration may become obsolete during later exploitation. This design aligns with the principles of Sliding-Window Upper Confidence Bound (SW-UCB) (Garivier & Moulines, 2011) algorithms used in non-stationary Multi-Armed Bandits. By restricting the bonus to a narrow temporal window $T_w$, the framework implicitly discounts outdated information, ensuring that the selection process prioritizes insights that are not just historically successful, but *contextually relevant* to the current state of the population.

The sensitivity analysis presented in Table 12 validates this design choice. Specifically, when this short-term memory is deactivated ($\tau_r = 0$), the algorithm inherently lacks the incentive to maintain search continuity. Consequently, the search explores widely but inefficiently, resulting in degraded performance. However, it is crucial to note that simply increasing the reward is not monotonically beneficial. As observed in the table, setting the reward magnitude too high ($\tau_r = 0.4$) leads to a deterioration in algorithm quality. This outcome suggests that excessive recency bias induces myopic behavior, wherein the algorithm becomes overly focused on the specific traits of the most recent iteration. Such short-sighted focus ultimately overshadows the robust, long-term effectiveness signals ($E_i$) accumulated over time. In contrast, introducing a moderate reward ($\tau_r = 0.2$) achieves consistently strong performance. This finding confirms that explicitly rewarding recent successes allows the LLM to effectively build upon immediate gains, thus guiding the population through the shifting fitness landscape without becoming distracted by transient fluctuations.

**Usage Penalty Weight** ($w_u$). This parameter regulates the exploration pressure within the Insight Pool by modulating the usage cost term, defined as $-w_u \log(N_i(t) + 1)$, where $N_i(t)$ represents the retrieval count of insight $i$. The choice of a logarithmic growth function is deliberate: it ensures that the penalty accumulates rapidly for the first few uses to prevent immediate overfitting, but saturates quickly thereafter. This design enables the score of a frequently used insight to stabilize rather than suffer a precipitous decline, thereby avoiding sharp priority reversals or oscillatory behavior while granting other high-potential candidates a fair chance. The sensitivity analysis in Table 12 confirms the necessity of this balanced regulation. When the penalty is removed ($w_u = 0$), the framework exhibits a significant performance drop. This indicates a phenomenon of *knowledge collapse*, where the algorithm over-exploits a few early dominant insights, effectively depriving the search of diversity. Conversely, an aggressive penalty ($w_u = 0.3$) yields suboptimal results by discarding valid insights prematurely before their full potential is realized. The default value of

$w_u = 0.1$ demonstrates robust efficacy, showing that a gentle, saturating penalty is sufficient to maintain a healthy turnover of valid ideas without suppressing fundamental design principles.

**Decay Rate ($R_{\mathbf{decay}}$).**  This parameter governs the temporal lifecycle of insights, controlling the rate at which older knowledge becomes obsolete. By applying a linear decay to the eviction score, $R_{\mathrm{decay}}$ enforces a mechanism of knowledge turnover, preventing the Insight Pool from becoming a static archive of legacy strategies that, while successful in early generations, may no longer be competitive. The sensitivity analysis in Table 12 reveals that this dynamic maintenance is critical for sustained performance. When the decay mechanism is disabled ($R_{\mathrm{decay}} = 0$), the algorithm yields the worst results among all tested configurations. This confirms that a static pool eventually becomes saturated with stale information, effectively locking out superior novel insights due to capacity constraints. Conversely, setting the rate too high ($R_{\mathrm{decay}} = 0.04$) causes the system to rapidly forget valid principles, disrupting the accumulation of algorithmic knowledge. The default rate of $R_{\mathrm{decay}} = 0.01$ demonstrates robust performance, continuously purging outdated or inactive insights to ensure the pool remains relevant and valid, thereby staying aligned with the evolving needs of the population.

**Diversity Threshold ($\delta_p$).**  This parameter establishes the minimum acceptable level of phenotypic diversity, acting as a sensitivity trigger for the Evolutionary Navigator. As defined in Eq. 7, diversity is computed via exact string matching of the generated algorithmic descriptions. This rigorous design choice is motivated by the fact that our standardized prompt template eliminates trivial syntactic noise (e.g., variable renaming), ensuring that any remaining lexical difference reflects a deliberate structural modification by the LLM. Moreover, exact matching captures subtle but critical logical alterations—such as *dynamic* vs. *static*—that embedding-based metrics might obscure. The threshold $\delta_p$ dictates when the system perceives the population as stagnated; dropping below it triggers a shift to an *Exploration* regime. The sensitivity analysis in Figure 4 reveals the delicate balance required here. A low threshold ($\delta_p = 0.1$) proves overly tolerant of redundancy, allowing the LLM to repeatedly output identical text without intervention. This failure to trigger exploration traps the search in local optima, leading to premature convergence. Conversely, an excessively high threshold ($\delta_p = 0.5$) imposes a hypersensitive constraint that misinterprets healthy convergence as stagnation. As evidenced by the significant instability in the green curve, this causes frequent, unnecessary disruptions that abort the fine-tuning of high-potential heuristics. Therefore, the default setting ($\delta_p = 0.3$) represents a robust operating point. By triggering intervention only when over 70% of the population consists of exact duplicates, it functions as an indicator that effectively anticipates local-optimum collapse, enforcing proactive exploration before the search fully stalls while still permitting sufficient iterations for the local refinement of elite patterns.

**Population Size ($N_{\mathbf{pop}}$).**  This parameter determines the breadth of the heuristic search space covered per generation. The sensitivity analysis in Figure 4 reveals a distinct trade-off: excessively small populations (e.g., $N_{\mathrm{pop}} = 4$) lead to premature convergence, as the limited gene pool lacks sufficient diversity to sustain effective recombination. Conversely, overly large populations (e.g., $N_{\mathrm{pop}} = 12$) significantly increase the computational burden (token consumption) without yielding proportional performance gains. The results confirm that a size of 8 serves as an effective balance point, providing sufficient genetic material for the LLM to synthesize high-quality heuristics while maintaining a manageable query budget.

**Maximum Generations ($G_{\mathbf{max}}$).**  This parameter defines the temporal budget of the evolutionary process. The convergence trajectories in Figure 4 demonstrate that the algorithm typically achieves near-optimal solutions within the first 6–8 generations. While extending the search to 12 or more generations might offer slight refinements, it yields diminishing returns. The marginal performance improvements observed in these later stages do not justify the significant additional inference costs. Consequently, we selected $G_{\mathrm{max}} = 8$ as the standard configuration, representing a trade-off that secures high solution quality while maximizing computational efficiency.

**Insight Pool Size ($C_{\mathbf{pool}}$).**  The capacity of the Insight Pool regulates the system's long-term memory. As indicated by the variance and trends in Figure 4, a small pool (e.g., $< 10$) leads to *catastrophic forgetting*, where valuable high-level principles are evicted before they can be effectively reused. On the other hand, an unconstrained or overly large pool acts as a noise accumulator, dilut-

ing high-quality insights with mediocre ones and confusing the LLM with irrelevant context. Our experiments indicate that a capacity of 30 effectively balances the retention of diverse, high-utility insights with the need to filter out noise, maintaining both stability and agility.

**Stagnation Threshold ($\tau_{\mathbf{stag}}$).** This parameter determines the sensitivity of the Evolutionary Navigator in detecting performance plateaus. It sets the number of consecutive non-improving generations required to trigger a shift from *Exploitation* to *Exploration*. The sensitivity analysis in Figure 4 highlights the critical balance involved. A low threshold (e.g., $\tau_{\text{stag}} = 1$) makes the system hypersensitive, frequently interrupting the refinement of promising solutions due to minor stochastic fluctuations. This prevents the algorithm from fully exploiting local basins of attraction. Conversely, a high threshold (e.g., $\tau_{\text{stag}} = 5$) renders the framework sluggish. Given the constrained evolutionary window (typically 12 generations) and the high cost of LLM inference, waiting for 5 generations represents a significant waste of the computational budget on unproductive search. The default value of $\tau_{\text{stag}} = 3$ provides a trade-off, allowing for rapid detection of diminishing returns while filtering out transient noise in the optimization trajectory.

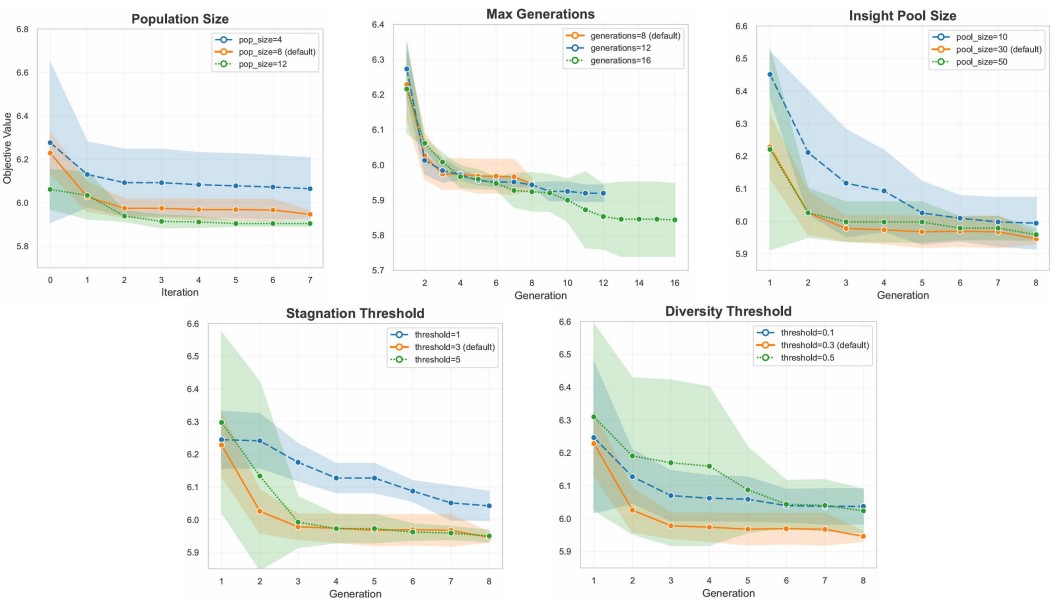

Figure 4: Parameter sensitivity analysis of **population size, maximum generations, insight pool size, stagnation threshold and diversity threshold**.

## C.7 ADDITIONAL ABLATION STUDIES AND ANALYSIS

To provide a comprehensive understanding of the internal mechanisms of HiFo-Prompt and to dissociate the contribution of specific components from the overall framework, we conducted a series of additional ablation studies. These experiments investigate the architectural necessity of the Insight Pool, the efficacy of the adaptive Evolutionary Navigator, the behavioral dynamics of regime switching, and the fine-grained design of the utility function.

**Architectural Necessity of the Insight Pool.** To determine whether the performance gains of HiFo-Prompt stem principally from the initialization with high-quality seed insights or from the framework's capability to manage them, we injected the identical set of seed insights into the initial populations or prompts of four state-of-the-art baselines: EoH, ReEvo, HSEvo, and MCTS-AHD. As detailed in Table 13, the incorporation of seed insights into these baselines yields negligible performance improvements, and in certain cases (e.g., MCTS-AHD), results in slight performance degradation due to interference with their native search logic. This stands in sharp contrast to the significant gains achieved by HiFo-Prompt. These results substantiate that standard LLM-based

evolutionary methods lack the necessary architectural mechanisms—specifically, the *Insight Pool* and the *Hindsight* reflection process—to effectively retain, retrieve, and utilize abstract knowledge. Consequently, the superior performance of HiFo-Prompt is attributed to its unique ability to leverage high-level guidance rather than merely the quality of initialization.

Table 13: Impact of **seed insights** on baselines, using TSP construction as an example. The results show the objective function values. Lower is better.

| Method | TSP10 | TSP20 | TSP50 | TSP100 | TSP200 |
|---|---|---|---|---|---|
| EoH (w/ seed insight) | 3.024 | 4.233 | 6.512 | 8.998 | 12.470 |
| EoH (w/o seed insight) | 3.049 | 4.252 | 6.441 | 8.990 | 12.514 |
| ReEvo (w/ seed insight) | 2.993 | 4.132 | 6.344 | 8.759 | 12.198 |
| ReEvo (w/o seed insight) | 2.994 | 4.126 | 6.294 | 8.773 | 12.324 |
| HSEvo (w/ seed insight) | 3.064 | 4.324 | 6.452 | 8.842 | 12.270 |
| HSEvo (w/o seed insight) | 3.001 | 4.170 | 6.307 | 8.729 | 12.183 |
| MCTS-AHD (w/ seed insight) | 2.938 | 4.142 | 6.294 | 8.724 | 12.076 |
| MCTS-AHD (w/o seed insight) | 2.983 | 4.174 | 6.317 | 8.769 | 12.176 |

**Efficacy of Adaptive Regime Switching.** The Evolutionary Navigator is designed to dynamically switch between *Explore*, *Exploit*, and *Balance* regimes based on real-time population states. To validate the necessity of this adaptive control, we compared the full HiFo-Prompt against a variant fixed permanently in the *Balance* mode, which attempts to combine exploration and exploitation in every prompt without explicit directional guidance. Table 14 demonstrates that the adaptive mechanism significantly outperforms the fixed strategy. This performance gap widens as the problem scale increases (e.g., reducing the gap on TSP200 from 22.115% to 8.877%). This indicates that a static prompt strategy is insufficient for navigating complex search landscapes. The Navigator's ability to explicitly enforce exploration during stagnation and exploitation during progress is critical for escaping local optima and refining solutions efficiently.

Table 14: Ablation study of the Evolutionary Navigator's **adaptive mechanism** vs. a fixed strategy.

| Method | TSP10 | TSP20 | TSP50 | TSP100 | TSP200 |
|---|---|---|---|---|---|
| HiFo-Prompt (adaptive) | **1.654%** | **3.619%** | **6.625%** | **8.582%** | **8.877%** |
| HiFo-Prompt (fixed Balance) | 5.812% | 9.676% | 15.599% | 18.924% | 22.115% |

**Behavioral Analysis of the Evolutionary Navigator.** To further elucidate the operational behavior of the Navigator, we analyzed the frequency of each regime triggered during the evolutionary process across three independent runs. As shown in Table 15, the system exhibits a balanced distribution of states. While *Balance* is the most frequent state (providing stability), the system frequently triggers *Explore* (avg. 102.3 times) and *Exploit* (avg. 83.7 times). This confirms that the diversity ($\delta_p$) and stagnation ($\tau_{stag}$) thresholds are actively functioning, ensuring the search dynamically oscillates between structural diversification and local refinement rather than converging prematurely.

Table 15: Frequency statistics of Evolutionary Regimes triggered by the Navigator.

| Regime | run 1 | run 2 | run 3 | avg. |
|---|---|---|---|---|
| Explore | 110 | 95 | 102 | 102.3 |
| Exploit | 81 | 88 | 82 | 83.7 |
| Balance | 129 | 137 | 136 | 134.0 |

**Synergistic Effect of Utility Function Components.** The utility function $U(k_i, t)$ is not merely a sum of terms but establishes a dynamic interplay among three constituents to ensure stable, convergent selection. The learned effectiveness $E_i(t)$ serves as the dominant, low-variance signal reflecting

long-term value via Exponential Moving Average (EMA). The usage penalty and recency bonus act as critical modulating factors. Specifically, the usage penalty employs a logarithmic growth design $(\log(N_i(t) + 1))$; this ensures the penalty saturates quickly, meaning an insight's priority is gently down-weighted rather than suffering a precipitous decline. This prevents sharp priority reversals and oscillatory behavior while preserving diversity. Conversely, the recency bonus injects a fixed, time-dependent reward to encourage search continuity along recently validated trajectories.

The ablation study in Table 16 empirically confirms this synergy. Removing the penalty ($w_u = 0$) leads to inferior results, indicating that without exploration pressure, the system over-exploits early dominant insights, leading to premature *knowledge collapse*. Removing the reward ($\tau_r = 0$) similarly harms performance, as the search loses its momentum and explores widely but inefficiently. The poorest performance occurs when both modulators are removed, validating that the simultaneous application of penalizing overuse and rewarding recency is essential for robust credit assignment.

Table 16: Ablation analysis of **usage penalty** ($w_u$) and **recency bonus** ($\tau_r$) in the Utility Function.

| Setting | run 1 | run 2 | run 3 | avg. |
|---|---|---|---|---|
| $w_u > 0, \tau_r > 0$ (default) | 5.885 | 5.838 | 5.873 | **5.866** |
| $w_u = 0, \tau_r > 0$ (w/o penalty) | 5.958 | 5.989 | 5.978 | 5.975 |
| $w_u > 0, \tau_r = 0$ (w/o reward) | 5.954 | 5.972 | 5.979 | 5.968 |
| $w_u = 0, \tau_r = 0$ (w/o both) | 5.984 | 6.026 | 5.982 | 5.997 |

**Calibration of Base Credit Assignment.** Finally, we analyzed the sensitivity of the base credit assignment coefficients defined in Eq. 3 ($g_{\text{eff}}^{\text{raw}}$). In Table 17, each configuration is presented as a triplet $[\beta_{\text{best}}, \beta_{\text{inc}}, \beta_{\text{pen}}]$, representing the base intercept values for three distinct performance tiers: the first value ($\beta_{\text{best}}$) rewards offspring that surpass the current population best (paradigm-shifting); the second ($\beta_{\text{inc}}$) rewards those improving upon the average but not the best (incremental); and the third ($\beta_{\text{pen}}$) penalizes individuals performing below the average (sub-optimal).

Table 17: Sensitivity analysis of **base reward** configurations.

| Configuration | run 1 | run 2 | run 3 | avg. |
|---|---|---|---|---|
| $[0.8, 0.2, 0.0]$ | 6.046 | 5.976 | 5.998 | 6.007 |
| $[1.0, 0.1, -0.5]$ | 5.921 | 5.932 | 5.902 | 5.918 |
| $[0.6, 0.3, -0.1]$ | 5.916 | 5.917 | 5.915 | 5.916 |
| $[0.8, 0.2, -0.3]$ (default) | 5.885 | 5.838 | 5.873 | **5.866** |

Our default configuration $[0.8, 0.2, -0.3]$ is not arbitrary but follows a specific asymmetric design philosophy: Elite Reward $\gg$ |Penalty| $\gtrsim$ Incremental Reward. The substantial elite reward (0.8) provides a strong signal for breakthroughs, prioritizing paradigm discovery. The incremental reward (0.2) acknowledges smaller gains, while the penalty (-0.3)—larger in magnitude than the incremental reward—effectively prunes unproductive directions. This asymmetry improves search efficiency by aggressively filtering noise while amplifying true innovations. Additionally, the coefficients follow a $\wedge$-*shaped* sensitivity design, where the intermediate tier receives the largest gradient to encourage rapid upward transitions, while the elite tier is assigned lower sensitivity to maintain stability. The results in Table 17 confirm that this theoretically grounded configuration yields the lowest objective value. Deviating towards more aggressive rewards (e.g., $[1.0, 0.1, -0.5]$) or different penalty structures disrupts this balance, resulting in inferior performance.

**Standard Deviation Analysis.** To comprehensively evaluate the stability and consistency of the models, we conducted three independent runs of all LLM-based AHD methods on several representative scales. We then summarized the objective function values of each run and reported their average (avg.) and standard deviation (std.), as shown in Table 18. These statistics provide a clear indication of the variability across runs, offering a more reliable basis for subsequent performance comparison and robustness analysis.

Table 18: Objective function values of three run and their average(avg.) and standard deviation(std.).

| Task | Method | run1 | run2 | run3 | avg. | std. |
|------|--------|------|------|------|------|------|
| TSP50-construction | EoH | 6.369 | 6.554 | 6.401 | 6.441 | 0.09887534 |
| | ReEvo | 6.264 | 6.298 | 6.320 | 6.294 | 0.02821347 |
| | HSEvo | 6.348 | 6.259 | 6.314 | 6.307 | 0.04491102 |
| | MCTS-AHD | 6.359 | 6.288 | 6.303 | 6.317 | 0.03742103 |
| | Ours | 6.135 | 6.087 | 6.041 | 6.088 | 0.04700355 |
| TSP100-GLS | EoH | 7.738 | 7.738 | 7.739 | 7.738 | 0.00034451 |
| | ReEvo | 7.742 | 7.741 | 7.738 | 7.740 | 0.00173957 |
| | HSEvo | 7.739 | 7.737 | 7.753 | 7.743 | 0.00894281 |
| | Ours | 7.737 | 7.737 | 7.738 | 7.737 | 0.00007013 |
| Online BPP 5k_C100 | EoH | 2028 | 2027 | 2025 | 2027 | 1.11355287 |
| | ReEvo | 2021 | 2023 | 2022 | 2022 | 0.60880932 |
| | HSEvo | 2022 | 2024 | 2088 | 2045 | 37.59627641 |
| | MCTS-AHD | 2027 | 2026 | 2026 | 2026 | 0.66901381 |
| | Ours | 2019 | 2020 | 2021 | 2020 | 0.64291005 |
| FSSP-GLS n100_m10 | EoH | 5640 | 5645 | 5644 | 5643 | 2.60832002 |
| | Ours | 5635 | 5638 | 5638 | 5637 | 1.67470225 |

## C.8    CONSUMPTION OF TIME AND TOKEN

Compared to other LLM-based AHD methods, our approach demonstrates advantages in both time and token consumption, with particularly significant reductions in computational time. Based on `qwen2.5-max` model, we calculated the runtime and token consumption of various methods across three problem domains: TSP-construction, Online BPP, and BO-CAF in Table 19.

Table 19: Time and token consumption with different methods.

| Methods | Consumption | TSP-construction | Online BPP | BO-CAF |
|---------|-------------|------------------|------------|--------|
| EoH | Time | 1.2h | 1h | 2h |
| | Input Token | 0.8M | 0.5M | 1.2M |
| | Output Token | 0.2M | 0.2M | 0.5M |
| ReEvo | Time | 2h | - | - |
| | Input Token | 1.1M | - | - |
| | Output Token | 0.4M | - | - |
| MCTS-AHD | Time | 4h | 3h | 14h |
| | Input Token | 1M | 1M | 1.3M |
| | Output Token | 0.3M | 0.2M | 0.6M |
| Ours | Time | 40 min | 36min | 1h |
| | Input Token | 0.5M | 0.28M | 0.8M |
| | Output Token | 0.2M | 0.12M | 0.2M |

## D    LIMITATION AND FUTURE WORK

### D.1    LIMITATION

While HiFo-Prompt demonstrates significant advancements in automated heuristic design, its current architecture possesses inherent limitations that define the boundaries of its present capabilities.

First, the core decision-making mechanism of the Evolutionary Navigator is predicated on a static, handcrafted control logic. This rule-based system, while interpretable, lacks the capacity for self-adaptation. Its fixed thresholds for stagnation, progress, and diversity are calibrated for the evaluated

problems but may not be universally optimal, potentially constraining its performance on novel problem landscapes or over different evolutionary timescales. The Navigator can react to predefined states but cannot learn or refine its control strategy from experience.

Second, the knowledge evolution within the Insight Pool is fundamentally intra-task. The framework excels at capturing, refining, and reusing design principles within the context of a single optimization problem. However, the true generalizability of these learned insights across different problem domains remains an unevaluated and open question. We have not yet systematically validated whether insights distilled from solving one problem class can effectively bootstrap the learning process on a structurally different, unseen one.

## D.2 FUTURE WORK

The limitations mentioned above naturally chart a course for several high-impact avenues for future research, aimed at enhancing the framework's autonomy, generality, and strategic depth.

A primary research thrust will be to transcend the Evolutionary Navigator's current heuristic-driven design by developing it into a learned metacontroller. We propose parameterizing its control function and leveraging techniques from the domain of Meta-Reinforcement Learning (Meta-RL). In this paradigm, the Navigator would operate as a high-level agent, learning a control policy that dynamically maps observational data from the evolutionary process—such as population diversity metrics, fitness landscape topology, and rates of convergence—to a nuanced control vector. This vector would modulate critical evolutionary parameters in real-time, including operator probabilities, selection pressure, and even strategic alterations to the LLM prompts themselves. The reward signal for this meta-agent would be carefully engineered to reflect overarching goals of evolutionary efficiency, such as maximizing the rate of fitness improvement or minimizing the computational cost to reach a performance threshold. Successfully implementing this would elevate the framework from a system guided by static rules to one capable of learning and deploying its own adaptive control strategies online, thereby achieving a superior order of autonomy and problem-specificity.

Another critical, complementary direction is the systematic investigation of inter-task knowledge transfer, intending to evolve the framework from a single-task solver into a more general algorithmic discovery platform. This research will proceed along two parallel vectors. First, we will conduct a rigorous empirical evaluation of the framework's zero-shot and few-shot transfer capabilities. By applying a mature Insight Pool—developed for a source task—to novel target domains, we can precisely quantify the generality of the learned principles and measure the extent to which discovery costs can be amortized across problems. Second, we will pioneer the exploration of more abstract and powerful knowledge representations that transcend the inherent ambiguities of natural language strings. This includes investigating structured canonical forms, such as predicate logic or probabilistic program sketches, and rich semantic representations like knowledge graphs. The central hypothesis is that such formalisms can more effectively decouple a core algorithmic invariant from its domain-specific instantiation, thereby creating a more robust foundation for seamless and compositional cross-domain knowledge transfer.

## E   PROMPTS WITH FORESIGHT AND HINDSIGHT

This section provides the specific, concrete templates of the prompts used to guide the LLM, offering a transparent and reproducible view of the core interaction engine at the heart of our HiFo-Prompt framework. The HiFo (Hindsight-Foresight) name is not arbitrary; it directly reflects our core methodology: the strategic deployment of distinct, context-aware prompts at different stages of the evolutionary process. Foresight prompts are strategically employed during the initial generation and creative mutation stages, encouraging the LLM to explore a diverse space of novel heuristic possibilities. In stark contrast, Hindsight prompts play a critical role in reflection and data-driven refinement, leveraging empirical performance feedback from past evaluations to methodically prune the search space and systematically improve promising solutions.

To ground these abstract concepts and make our prompts tangible, we anchor our examples in the canonical Online Bin Packing (OBP) problem. The objective in OBP is to assign an incoming sequence of items of varying sizes into a minimum number of fixed-capacity bins, with the crucial constraint that each item must be placed without knowledge of future items. Within this problem

context, our framework's specific task is to evolve a high-performance Python scoring function, score(item, bin), which acts as the core decision-making logic. This function evaluates the suitability of a specific candidate bin for an incoming item. A higher returned score signifies a more desirable placement, and the overarching control logic of the framework places the item in the bin that yields the maximum score. The following subsections provide verbatim templates for each distinct phase of the evolutionary process, clearly demonstrating how the system's guidance dynamically shifts its focus—from broad exploration to focused exploitation—as the search progresses.

## E.1 INITIAL PROMPT STRATEGY I1

The i1 operator uses the following prompt template to generate the initial population of heuristics. The guidance provided to the LLM at this stage comes from the initial state of the framework's components, including a set of high-quality seed insights.

---

### Prompt for Operator i1

Given a sequence of items and a set of identical bins with a fixed capacity, you need to assign each item to a bin to minimize the total number of bins used. The task can be solved step-by-step by taking the next item and deciding which bin to place it in based on a score.

First, describe your new algorithm and main steps in one sentence. The description must be inside a brace. Next, implement it in Python as a function named **score**.

This function should accept 2 input(s): `'item'`, `'bins'`.

The function should return 1 output(s): `'scores'`.

The **score** function is designed to evaluate the placement options for a given item. It takes the item to be placed and the current list of bins as input. It returns a list of numerical scores, with each score corresponding to a bin in the input list. This list of scores guides the heuristic in selecting the most suitable bin for the item according to the generated logic.

**Consider these successful design principles I've observed recently:**

- *<A successful design principle from Insights pool>*

- *<Another successful design principle...>*

For the evolutionary regime, please pay special attention to: *<A specific instruction from Design Directive>*

*Depending on the regime, try significantly different parameter values (focus_exploration), or fine-tune existing ones (focus_exploitation), or combine both strategies (balanced_search).*

Do not give additional explanations.

---

## E.2 RECOMBINATION PROMPT STRATEGY E1

The following template for the e1 operator is a representative example of a prompt used during the main evolutionary loop. It demonstrates how parent heuristics and the full, dynamic guidance from the meta-cognitive components are integrated.

---

### Prompt for Operator e1

Given a sequence of items and a set of identical bins with a fixed capacity, you need to assign each item to a bin to minimize the total number of bins used. The task can be solved step-by-step by taking the next item and deciding which bin to place it in based on a score.

I have k existing algorithms with their codes as follows:

**No.1 algorithm and the corresponding code are:**

*<Description of the first algorithm>*

*<The Python code implementation of the first algorithm>*

...

**No.k algorithm and the corresponding code are:**

*<Description of the last algorithm>*

*<The Python code implementation of the last algorithm>*

Please help me create a new algorithm that has a totally different form from the given ones.

---

First, describe your new algorithm and main steps in one sentence, enclosed in braces {}. Next, implement it in Python as a function named **score**. This function should accept 2 input(s): `<'item',` `'bins'>`. The function should return 1 output(s): `<'scores'>`. `<Additional info on inputs & outputs> <Other constraints or requirements>`
**Consider these successful design principles I've observed recently:**

- `<A successful design principle from Insights pool>`

- `<Another successful design principle...>`

For the evolutionary regime, please pay special attention to: `<A specific Design Directive for promoting structural novelty, preserving diversity, and avoiding premature convergence>`
*Depending on the regime, try significantly different parameter values (focus_exploration), or fine-tune existing ones (focus_exploitation), or combine both strategies (balanced_search).*
Do not give additional explanations.

## E.3 RECOMBINATION PROMPT STRATEGY E2

The e2 operator focuses on "motivated recombination." It prompts the LLM to first identify a common "backbone" or core principle shared by the parent heuristics and then to create a new, improved algorithm based on that shared foundation.

### Prompt for Operator e2

Given a sequence of items and a set of identical bins with a fixed capacity, you need to assign each item to a bin to minimize the total number of bins used. The task can be solved step-by-step by taking the next item and deciding which bin to place it in based on a score.
I have k existing algorithms with their codes as follows:
**No.1 algorithm and the corresponding code are:**
`<Description of the first algorithm>`
`<The Python code implementation of the first algorithm>`
…
**No.k algorithm and the corresponding code are:**
`<Description of the last algorithm>`
`<The Python code implementation of the last algorithm>`
Please help me create a new algorithm that has a totally different form from the given ones but can be motivated from them.
Firstly, identify the common backbone idea in the provided algorithms. Secondly, based on the backbone idea, describe your new algorithm and main steps in one sentence, enclosed in braces {}. Thirdly, implement it in Python as a function named **score**. This function should accept 2 input(s): `<'item',` `'bins'>`. The function should return 1 output(s): `<'scores'>`. `<Additional info on inputs & outputs> <Other constraints or requirements>`
**Consider these successful design principles I've observed recently:**

- `<A successful design principle from Insights pool>`

- `<Another successful design principle>`

**For this recombination, please pay special attention to:** `<A specific Design Directive for simplification prune low-impact features>`
*Depending on the regime, try significantly different parameter values (focus_exploration), or fine-tune existing ones (focus_exploitation), or combine both strategies (balanced_search).*
Do not give additional explanations.

## E.4 MUTATION PROMPT STRATEGY M1

The M1 operator implements a form of targeted mutation, generating a single-parent descendant by refining its most critical components. It performs surgical modifications to elements like scoring rules or parameter weights, while meticulously safeguarding the parent's established, high-performing logic. This process is not random; it is guided by a synthesis of recent, high-utility insights and a high-level strategic directive. By injecting controlled, purposeful diversity, M1 en-

ables the search to escape local optima without dismantling the parent's effective architecture. This deliberate balance between exploitation (refining what works) and exploration (seeking novelty) is crucial for accelerating convergence towards superior heuristics.

---

**Prompt for Operator m1**

Given a sequence of items and a set of identical bins with a fixed capacity, you need to assign each item to a bin to minimize the total number of bins used. The task can be solved step-by-step by taking the next item and deciding which bin to place it in based on a score.

I have one algorithm with its code as follows:

**Algorithm description:** `<Description of the parent algorithm>`

**Code:**

`<The Python code implementation of the parent algorithm>`

Please assist me in creating a new algorithm that has a different form but can be a modified version of the algorithm provided.

First, describe your new algorithm and main steps in one sentence, enclosed in braces {}. Next, implement it in Python as a function named **score**. This function should accept 2 input(s): `<'item', 'bins'>`. The function should return 1 output(s): `<'scores'>`. `<Additional info on inputs & outputs>` `<Other constraints or requirements>`

**Consider these successful design principles I've observed recently:**

- `<A successful design principle from Insights pool>`

- `<Another successful design principle>`

**For this mutation, please pay special attention to:** `<A specific Design Directive for mutation>`

*Depending on the regime, try significantly different parameter values (focus_exploration), or fine-tune existing ones (focus_exploitation), or combine both strategies (balanced_search).*

Do not give additional explanations.

---

### E.5  MUTATION PROMPT STRATEGY M2

The M2 operator implements *parameter mutation*, a targeted process to modify a heuristic's numerical behavior. It instructs the LLM to first deconstruct the parent's score function to identify its key hyperparameters. Following this analysis, the model is prompted to generate a new set of parameter values. This generation can either explore novel configurations through significant changes or fine-tune existing ones via subtle adjustments. This surgical approach methodically injects parametric diversity into the population while preserving the integrity of the core algorithmic logic.

---

**Prompt for Operator m2**

Given a sequence of items and a set of identical bins with a fixed capacity, you need to assign each item to a bin to minimize the total number of bins used. The task can be solved step-by-step by taking the next item and deciding which bin to place it in based on a score.

I have one algorithm with its code as follows:

**Algorithm description:** `<Description of the parent algorithm>`

**Code:**

`<The Python code implementation of the parent algorithm>`

Please identify the main algorithm parameters and assist me in creating a new algorithm that has different parameter settings of the score function provided.

First, describe your new algorithm and main steps in one sentence, enclosed in braces {}. Next, implement it in Python as a function named **score**. This function should accept 2 input(s): `<'item', 'bins'>`. The function should return 1 output(s): `<'scores'>`. `<Additional info on inputs & outputs>` `<Other constraints or requirements>`

**Consider these successful design principles I've observed recently:**

- `<A successful design principle from Insights pool>`

- `<Another successful design principle>`

**When adjusting parameters, please pay special attention to:** `<A specific Design Directive for parameter tuning>`

---

> *Depending on the regime, try significantly different parameter values (focus_exploration), or fine-tune existing ones (focus_exploitation), or combine both strategies (balanced_search).*
> Do not give additional explanations.

## E.6 MUTATION PROMPT STRATEGY M3

The M3 operator is designed for *structural simplification* to enhance heuristic robustness and combat overfitting. It instructs the LLM to perform a critical analysis of the parent score function, specifically targeting components suspected of being over-specialized to in-distribution data. These potentially brittle or overly complex segments are then strategically pruned or streamlined. The outcome is a more parsimonious and computationally lean implementation that is theorized to exhibit superior generalization to out-of-distribution scenarios, all while preserving the original function signature to ensure architectural compatibility.

---

**Prompt for Operator m3**

Given a sequence of items and a set of identical bins with a fixed capacity, you need to assign each item to a bin to minimize the total number of bins used. The task can be solved step-by-step by taking the next item and deciding which bin to place it in based on a score.
I have one algorithm with its code as follows:
**Code:**
`<The Python code implementation of the parent algorithm>`
First, identify the main components in the function above. Next, analyze which of these components may be overfitting to the in-distribution instances. Then, simplify those components to enhance generalization to out-of-distribution cases. Finally, provide the revised code, keeping the function name, inputs, and outputs unchanged.
Provide the complete revised function implementation, preserving its original signature. `<The function name, inputs & outputs specification from your prompt>`
**Consider these successful design principles I've observed recently:**

- `<A successful design principle from Insights pool>`

- `<Another successful design principle>`

**When simplifying, please pay special attention to:** `<A specific Design Directive for simplification>`
*Depending on the regime, try significantly different parameter values (focus_exploration), or fine-tune existing ones (focus_exploitation), or combine both strategies (balanced_search).*
Do not give additional explanations.

---

## F   MANAGING FORESIGHT AND HINDSIGHT KNOWLEDGE

### F.1   HINDSIGHT EVOLUTION VIA INSIGHT DISTILLATION

To continually enrich our system's understanding of effective optimization strategies, we employ a Large Language Model (LLM) to distill high-level design principles from high-performing algorithms. This process is guided by a structured prompt, shown below, which provides the LLM with the descriptions and/or code of elite solutions discovered during an evolutionary run. The core instruction tasks the model with synthesizing concise, generalizable, and performance-positive patterns, drawing insights from both the conceptual descriptions and the code implementations. This automated extraction mechanism allows our system to learn from its own successes and progressively build a more sophisticated knowledge base.

---

**Prompt Template for Insight Extraction**

**The following are core descriptions and/or code of high-performance optimization algorithms evolved recently:**

**Algorithm 1:** *<Natural language description and/or code of elite individual 1>*
**Algorithm 2:** *<Natural language description and/or code of elite individual 2>*
**Algorithm n: ...** *(and so on for the top 30% of the population)*

Please extract 1-2 concise, generic, and performance-positive **[design principles]** or **[effective patterns]** from the above algorithms. These principles should be applicable to various combinatorial optimization problems, not just the specific problem domain. When formulating these principles, it is essential to draw insights from *both* the conceptual natural language descriptions *and* their corresponding code implementations. Focus on identifying the underlying strategic design choices and algorithmic methodologies rather than superficial characteristics or specific implementation minutiae.

**Each principle/pattern must be expressed as an independent sentence in the following format:**

- *Balance local optimization with global solution structure when making decisions.*

- *Prioritize choices that maintain flexibility for future decision-making steps.*

- *Implement adaptive mechanisms that respond to problem instance characteristics.*

**Provide only the list of principles**, without any preamble or other explanatory text.

---

## F.2 THE DIRECTIVE POOL FOR FORESIGHT

To operationalize the Evolutionary Navigator's high-level strategy, we map the chosen regime—Exploration, Exploitation, or Balance—to a specific Design Directive. Each directive is a fine-grained textual instruction, uniformly sampled from a predefined pool corresponding to the active regime. This sampled directive is then integrated into the generation prompt. This two-tiered mechanism facilitates fine-grained control over the generation process, ensuring model outputs are precisely aligned with the overarching evolutionary objective.

---

**Design Directive**

- **Balance:**
  - *Optimizing objective function evaluation criteria.*
  - *Considering the long-term impact of current decisions.*
  - *Balancing local optimality with global search strategies.*
  - *Improving algorithm robustness across different problem instances.*
  - *Managing computational complexity and time efficiency.*

- **Exploitation:**
  - *Refining core evaluation and scoring functions.*
  - *Fine-tuning critical algorithm parameters and thresholds.*
  - *Improving the precision of existing heuristics and rules.*
  - *Reducing unnecessary computational overhead.*

- **Exploration:**
  - *Exploring novel solution construction methodologies.*
  - *Investigating alternative problem decomposition approaches.*
  - *Introducing new randomization or adaptive mechanisms.*
  - *Experimenting with hybrid strategy combinations.*

---

### F.3 INSIGHT SEED POOL

At the inception of the framework's execution, the LLM bootstraps the heuristic generation process by drawing from a curated repository of Seed Insights, a mechanism designed to mitigate the classic cold start problem and channel the model's creativity. These insights, which encapsulate established principles and canonical rules-of-thumb distilled from decades of human expertise in heuristic design—such as the *Shortest Processing Time* principle in scheduling or the *Nearest Neighbor* concept in routing—are not rigid constraints but rather high-level conceptual guidelines. They are strategically injected into the foundational prompts, often framed within a dedicated *human knowledge* block, to act as an intellectual scaffold. This initial infusion of well-vetted knowledge serves to ground the search, preventing the generation of naive or logically flawed heuristics and providing a potent directional bias that immediately steers the early stages of algorithmic evolution away from vast, unproductive regions of the design space. By ensuring the process begins not from a tabula rasa but from a high-quality, well-founded starting point, these seed insights exert a persistent influence that accelerates convergence and significantly enhances the quality and novelty of all subsequent automated discovery.

---

**Seed Insights**

- *Design adaptive hybrid meta-heuristics synergistically fusing multiple search paradigms and dynamically tune operator parameters based on search stage or problem features.*

- *Employ machine learning to mine problem structures and use learned insights to intelligently bias towards promising search regions.*

- *Explore objective function engineering by introducing auxiliary objectives or dynamically adjusting weights to reshape the search landscape.*

- *Construct problem-specialized solution representations and co-design dedicated operators to fully leverage the representation's structure.*

- *Implement intelligent diversification based on solution feature space analysis to systematically target uncovered regions and escape local optima.*

---

## G THE USE OF LARGE LANGUAGE MODELS (LLMS)

Throughout the drafting and revision process of this manuscript, we employed Google's Gemini large language model as an advanced writing and editing tool. Its use was strictly limited to refining the language and presenting our existing ideas and research. The model's contributions include: (1) correcting grammatical, spelling, and punctuation errors; (2) rephrasing complex sentences to improve readability and precision; and (3) suggesting adjustments to tone and style to ensure consistency with academic standards.

Critically, the LLM was not used for any substantive intellectual contribution. All aspects of the research, including the formulation of research questions, literature review, methodology, data analysis, and the drawing of conclusions, were conducted exclusively by the human authors. The authors critically evaluated each suggestion provided by the LLM for accuracy and appropriateness, and we retain full responsibility for all claims, arguments, and the final articulation of the work.

## H GENERATED HEURISTICS

In this section, we compile the most successful heuristics produced by HiFo-Prompt, spanning the entire suite of experimental settings.

### H.1 THE BEST TSP-CONSTRUCTIVE HEURISTIC

Listing 1: Python Implementation of the Node Selection Strategy

```python
import numpy as np

def select_next_node(current_node, destination_node, unvisited_nodes,
    distance_matrix):
    """
    Selects the next node to visit based on a hybrid scoring mechanism
    involving
    MST lookahead and cluster-aware simulation.
    """

    def calculate_pruned_mst_cost(nodes):
        """Calculates the MST cost for a subset of nodes."""
        if len(nodes) <= 1:
            return 0
        # Create edges only between relevant nodes
        edges = [(distance_matrix[i][j], i, j)
                for i in nodes for j in nodes if i < j]
        edges.sort(key=lambda x: x[0])

        parent = {node: node for node in nodes}
        def find(u):
            while parent[u] != u:
                parent[u] = parent[parent[u]]
                u = parent[u]
            return u

        mst_cost = 0
        for cost, u, v in edges:
            root_u, root_v = find(u), find(v)
            if root_u != root_v:
                parent[root_v] = root_u
                mst_cost += cost
        return mst_cost

    def perform_cluster_aware_simulation(node):
        """Simulates a greedy path from the candidate node."""
        future_nodes = [n for n in unvisited_nodes if n != node]
        total_cost = distance_matrix[current_node][node]
        current = node

        while future_nodes:
            # Greedy selection based on normalized distance
            next_candidate = min(
                future_nodes,
                key=lambda x: distance_matrix[current][x] /
                            (np.median(distance_matrix[x]) + 1e-9)
            )
            total_cost += distance_matrix[current][next_candidate]
            current = next_candidate
            future_nodes.remove(current)

        total_cost += distance_matrix[current][destination_node]
        return total_cost

    def evaluate_candidate(node):
        """Computes the weighted score for a candidate node."""
        future_nodes = [n for n in unvisited_nodes if n != node]

        # 1. MST Cost (Lookahead)
        pruned_mst_cost = calculate_pruned_mst_cost(future_nodes + [
    destination_node])

        # 2. Normalized Immediate Cost
```

```python
        normalized_cost = distance_matrix[current_node][node] /
            (np.max(distance_matrix[current_node]) + 1e-9)

        # 3. Simulation Cost
        simulation_cost = perform_cluster_aware_simulation(node)

        # Dynamic Weighting based on search progress
        progress_ratio = len(unvisited_nodes) / len(distance_matrix)
        weight_mst = 0.4 - 0.15 * progress_ratio
        weight_normalized = 0.3 + 0.1 * progress_ratio
        weight_simulation = 0.3

        return (weight_mst * pruned_mst_cost +
                weight_normalized * normalized_cost +
                weight_simulation * simulation_cost)

    # Beam search filtering to reduce computation
    beam_width = max(2, min(6, len(unvisited_nodes) // 3))
    # Pre-filter candidates using a simple heuristic
    candidates = sorted(
        unvisited_nodes,
        key=lambda x: distance_matrix[current_node][x] /
                      (np.mean(distance_matrix[current_node]) + 1e-9)
    )[:beam_width * 3]

    scored_candidates = []
    for candidate in candidates:
        eval_score = evaluate_candidate(candidate)
        scored_candidates.append((eval_score, candidate))

    # Select the node with the minimum weighted score
    next_node = min(scored_candidates, key=lambda x: x[0])[1]
    return next_node
```

## H.2 THE BEST ONLINE BIN PACKING HEURISTIC

Listing 2: Evolved Scoring Function for Online BPP

```python
import numpy as np

def score(item, bins):
    """
    Calculates the placement scores for candidate bins given an item.
    Higher scores indicate better placement suitability.
    """

    # Compute residuals (remaining space) after placing the item
    residuals = bins - item

    # Stochastic perturbation for enhanced exploration
    # Helps break ties and avoid deterministic local optima
    perturbation = np.random.normal(1.0, 0.2, size=len(bins))

    # Cubic residual scoring to emphasize finer space usage precision
    # Smaller residuals yield significantly higher base scores
    residual_scores = 1 / (1 + residuals ** 3) * perturbation

    # Delayed penalty adjustment based on long-term impact of
    underutilization
    # Adaptive threshold with increased sensitivity
    threshold = np.mean(bins) * 0.4
    delayed_penalty = np.where(residuals > threshold,
                               1 / (1 + residuals ** 2),
                               0)
```

```python
    # Reward function for bins maintaining balanced utilization over time
    # Gaussian-like reward centered around the threshold
    balanced_utilization_reward = np.exp(
        -np.abs(residuals - threshold) ** 2 / (threshold ** 2)
    )

    # Dynamic scaling factor based on cumulative fit ratio and trends
    fit_ratios = item / bins
    scaling_factor = 0.6 + 0.4 * np.tanh(fit_ratios * 5)  # Sharper
    scaling

    # Combine components:
    # Emphasizes long-term efficiency via scaling and delayed penalties
    scores = (scaling_factor * (residual_scores +
    balanced_utilization_reward)
              - delayed_penalty)

    return scores
```

## H.3 THE BEST TSP-GLS HEURISTIC

Listing 3: Hybrid Edge Update Function for TSP

```python
import numpy as np

def update_edge_distance(edge_distance, local_opt_tour, edge_n_used):
    """
    Updates the edge distance matrix to penalize edges used in local
    optima,
    facilitating escape from local basins using a hybrid penalty strategy
    .
    """

    # Parameters for the novel hybrid strategy
    learning_rate = 0.3
    diversity_factor = 0.4
    penalty_decay = 0.85
    min_penalty = 0.01
    smoothing_term = 1e-6

    # Compute normalized edge usage probabilities
    total_usage = np.sum(edge_n_used) + smoothing_term
    edge_probabilities = edge_n_used / total_usage

    # Measure global asymmetry to gauge search bias
    asymmetry = np.abs(edge_probabilities - edge_probabilities.T).mean()
    # Normalize asymmetry to [0, 1] range
    normalized_asymmetry = asymmetry / (1 + smoothing_term)

    # Identify edges in the local optimal tour
    tour_edges = [
        (local_opt_tour[i], local_opt_tour[i+1])
        for i in range(len(local_opt_tour)-1)
    ]
    tour_edges.append((local_opt_tour[-1], local_opt_tour[0])) # Close
    loop

    updated_edge_distance = edge_distance.copy()

    for (i, j) in tour_edges:
        if i > j:  # Ensure symmetry handling
            i, j = j, i
```

```python
        # Phase 1: Adaptive learning penalty
        learning_penalty = learning_rate * (
            1 - np.exp(-edge_probabilities[i, j])
        )

        # Phase 2: Diversity-driven modulation
        diversity_penalty = (diversity_factor * normalized_asymmetry *
                             np.log(1 + edge_probabilities[i, j]))

        # Dynamic weight modulation based on edge usage frequency
        modulation_factor = penalty_decay ** (
            edge_n_used[i, j] + smoothing_term
        )

        combined_penalty = (learning_penalty + diversity_penalty) * \
                           modulation_factor

        # Add stochastic noise to encourage exploration
        noise = np.random.uniform(0, 1.5)
        exploration_noise = min_penalty + noise * modulation_factor

        # Combine penalties and update edge distance (symmetric)
        total_penalty = combined_penalty + exploration_noise
        updated_edge_distance[i, j] += total_penalty
        updated_edge_distance[j, i] += total_penalty

    return updated_edge_distance
```

## H.4 THE BEST FSSP-GLS HEURISTIC

Listing 4: Chaotic Matrix Adjustment and Job Selection

```python
import numpy as np
from sklearn.cluster import KMeans

def get_matrix_and_jobs(current_sequence, time_matrix, m, n):
    """
    Generates a perturbed time matrix using chaotic dynamics and
    identifies
    critical jobs to disturb based on clustering dispersion patterns.
    """

    def calculate_makespan(sequence, matrix, m):
        """Helper to compute the makespan of a sequence."""
        machine_times = [0] * m
        for job in sequence:
            machine_times[0] += matrix[job, 0]
            for i in range(1, m):
                machine_times[i] = (max(machine_times[i], machine_times[i
-1])
                                    + matrix[job, i])
        return max(machine_times)

    # Step 1: Compute current makespan baseline
    current_makespan = calculate_makespan(current_sequence, time_matrix,
m)

    # Step 2: Chaotic adjustment using Logistic Map dynamics
    # Introduces non-linear variance to escape local optima
    new_matrix = time_matrix.copy()
    chaos_factor = 3.99  # Parameter near the edge of chaos
    chaotic_values = np.random.rand(n, m)

    for _ in range(10):  # Iterative chaotic updates
```

```python
        chaotic_values = chaos_factor * chaotic_values * (1 -
chaotic_values)

        # Scale chaotic values to a perturbation range [0.9, 1.1]
        chaotic_perturbation = 0.9 + 0.2 * chaotic_values
        new_matrix *= chaotic_perturbation
        new_matrix = np.clip(new_matrix, 1, None)  # Ensure valid processing
times

        # Step 3: Cluster jobs based on execution time patterns
        # Fit KMeans on the perturbed matrix regarding the current sequence
        kmeans = KMeans(
            n_clusters=max(2, n // 10),
            random_state=42,
            n_init='auto'
        ).fit(new_matrix[current_sequence])

        job_labels = kmeans.labels_

        # Calculate dispersion: average std deviation within each cluster
        cluster_dispersion = np.array([
            np.std(new_matrix[current_sequence][job_labels == c], axis=0).
mean()
            for c in range(kmeans.n_clusters)
        ])

        # Assign scores: jobs in highly dispersed clusters get higher
priority
        job_scores = np.array([cluster_dispersion[label] for label in
job_labels])

        # Select top diverse jobs for structural perturbation
        num_perturb = max(1, n // 6)
        perturb_jobs = np.argsort(-job_scores)[:num_perturb]

        return new_matrix, perturb_jobs
```

