# OpenReview forum: "HiFo-Prompt: Prompting with Hindsight and Foresight for LLM-based Automatic Heuristic Design"
_ICLR.cc/2026/Conference — ICLR 2026 Poster_

### Official Review · Reviewer_9fSr · 2025-10-28

**Soundness:** 3
**Presentation:** 3
**Contribution:** 3
**Rating:** 6
**Confidence:** 4

**Summary:**

This paper introduces HiFo-Prompt, a new method for LLM-based AHD. It integrates a Foresight module, featuring an evolutionary navigator that monitors population dynamics and steers the search using interpretable "verbal gradients," and a Hindsight module, which maintains a self-evolving insight pool that distills successful design principles from high-performing code into knowledge. Evaluated on different heuristic design tasks like TSP, BPP, and FSSP, HiFo-Prompt demonstrates competetive performance, achieving superior results with greater computational efficiency than existing LLM-based AHD methods.

**Strengths:**

The self-evolving insight pool (Hindsight) and foresight instructions effectively prevent knowledge decay while enabling more strategic exploration of the heuristic space.

It achieves superior performance with fewer LLM calls and lower runtime compared to other state-of-the-art AHD methods.

**Weaknesses:**

The evolutionary navigator uses a fixed, rule-based policy with hand-tuned thresholds, which may lack generalization.

The paper would benefit from additional illustrations and a more extensive set of results to further support its claims

**Questions:**

The stagnation is measured by raw fitness, delta g, which is a fixed value and may suffer from poor generalization to different heuristic design tasks.

The semantic variety is calculated based on the textual descriptions of algorithms (eq. 7). Is it the thought or the code text? It seems that the indicator only counts when the two algorithms are exactly the same. Will it be too greedy?

For the Foresight module, how was the specific set of Design Directives in the pool (Appendix G.3) designed? Was there an ablation study on the impact of different directive wordings on the LLM's output quality?

The framework's knowledge management is confined to a single task; can the learned insights be generalized or transferred to new, unseen problem domains?

What does the final algorithm look like? and how the insights and foresight prompts contribute to the generation of better heuristics, could you provide example illustrations?

A discussion and comparison with related works on prompt evolution and hierarchical search is suggested [1-3].

[1] MeLA: A Metacognitive LLM-Driven Architecture for Automatic Heuristic Design, arXiv

[2] Large Language Model-driven Large Neighborhood Search for Large-Scale MILP Problems, ICML

[3] Experience-guided reflective co-evolution of prompts and heuristics for automatic algorithm design, arXiv


There are typos and inadequate descriptions: e.g.,

line 811 ?

line 854, 788

Figure 1, The Left and Right can be misleading

---

> ### Author Response · Authors · 2025-11-26
> **Response (1/3)**
>
> > **W1: Concerns about evolutionary navigator.** The evolutionary navigator uses a fixed, rule-based policy with hand-tuned thresholds, which may lack generalization.
>
> Thank you for raising this very promising question. We acknowledge that the current version of the Evolutionary Navigator uses rule-based strategies with hand-tuned thresholds. This is a deliberate trade-off to simplify the model while ensuring interpretability and computational efficiency. We will explain as follows:
>
> 1.Our core design is to implement an explicit control strategy rather than passively embedding the exploration–exploitation trade-off within a specific framework. In LLM-driven evolution, each algorithm generation and evaluation is far costlier than in traditional EC. Therefore, we rely on a Navigator to actively monitor the population and instruct the LLM on strategy changes, maximizing the value of each expensive query.
>
> 2.The effectiveness of explicit control indeed depends on well-chosen thresholds. In Foresight, $τ_{stag} = 3$ ensures that three consecutive generations of stagnation in an 8-generation window trigger quick intervention, avoiding wasted LLM evaluations. Similarly, $\delta_p = 0.3$ flags when over 70\% of the population becomes ideologically homogeneous, preventing premature convergence.
>
> Figure 4 shows these defaults are robust across different problems. We also note their limitations and discuss in Sections E.1 and E.2 that developing an adaptive, learning-based meta-controller is a key future direction.
>
> > **W2: Concerns about illustrations.** The paper would benefit from additional illustrations and a more extensive set of results to further support its claims
>
> This is a very valuable suggestion. We agree that including specific figures and more comprehensive results would greatly enhance the paper's clarity and persuasiveness. Due to space limitations in the main paper, we will include additional results and illustrations in the appendix to aid understanding.
>
> > **Q1: Concerns about stagnation.** The stagnation is measured by raw fitness, delta g, which is a fixed value and may suffer from poor generalization to different heuristic design tasks.
>
> 1.We would like to clarify that in our framework, $\Delta g$ threshold ($10^{-4}$) is not intended to measure the "magnitude" of fitness changes. It serves as a problem-independent and numerically robust "zero-improvement" detector. Fitness values can vary greatly across optimization tasks, but $10^{-4}$ here is not a scale-based measure. Its main role is to determine whether the best solution improves over the previous one within floating-point precision. So, it can avoid the precision issues that arise when directly using $\Delta g = 0$.
>
> 2.Our stagnation detection mechanism is insensitive to the absolute fitness scale. Therefore, the concern about "poor generalization to different heuristic-design tasks" does not apply. As described in Section 3.3, the Evolutionary Navigator uses a stagnation counter $C_{stag}$ that tracks consecutive "zero-improvement" generations ($\Delta g \le 10^{-4}$). Strategy changes, such as switching to exploration, are triggered by the duration of stagnation ($C_{stag} \ge \tau_{stag}$), not the absolute fitness change. This ensures robustness across tasks with different fitness ranges.

---

> ### Author Response · Authors · 2025-11-26
> **Response (2/3)**
>
> > **Q2: Concerns about semantic variety.** The semantic variety is calculated based on the textual descriptions of algorithms (eq. 7). Is it the thought or the code text? It seems that the indicator only counts when the two algorithms are exactly the same. Will it be too greedy?
>
> The diversity metric in Eq. 7 is computed from the textual descriptions of algorithms, reflecting their design thought rather than code text. We would like to clarify that this choice is intentional in LLM-based AHD:
>
> 1.Our standardized prompt template imposes strict syntactic constraints, fixing the output/input format and required terminology. Without such constraints, LLMs often produce semantically identical content with different wording. Under this controlled syntax, differences in character sequences reliably correspond to meaningful changes in algorithmic logic. Thus, exact matching effectively filters out trivial variations while capturing substantive design updates.
>
> 2.Semantic embeddings cannot reliably detect fine-grained functional differences. Although they capture general similarity, they often fail to distinguish sequences with small edit distances but conflicting behaviors (for example, "dynamic weighting" vs. "static weighting"). Prior work [1] shows that embedding models struggle with such distinctions. Embedding-based metrics would therefore risk conflating genuinely different strategies.
>
> By using exact string matching, HiFo-Prompt ensures that even minor but meaningful logical modifications are preserved as unique individuals, avoiding the loss of valid variation that semantic metrics might cause. The diversity threshold (0.3) serves as an early warning signal: a score below this indicates that over 70% of the population is identical, suggesting growing redundancy. Triggering "Exploration" at this point prevents the search from becoming trapped in a local optimum.
>
> [1] Agarwal D, Arivazhagan M G, Das R, et al. Searching for optimal solutions with LLMs via bayesian optimization. ICLR, 2025.
>
> > **Q3: Concerns about Design Directives.** For the Foresight module, how was the specific set of Design Directives in the pool (Appendix G.3) designed? Was there an ablation study on the impact of different directive wordings on the LLM's output quality?
>
> We will address this question from the following aspects.
>
> 1.All directives in the pool are derived from common principles in evolutionary computation (EC) and heuristic design. We summarize these high-level principles into three core evolutionary strategy categories: Exploration, Exploitation, and Balance, aligned with the population's state.
>
> For example, directives under "Exploitation" (e.g., "fine-tuning critical parameters") reflect the principle of reinforcing local search in EC, while those under "Exploration" (e.g., "exploring novel solution construction methodologies") embody the idea of introducing diversity to escape local optima. We translate these high-level strategy concepts into natural language instructions understandable by the LLM.
>
> 2.We would like to emphasize that these directives are not to specify exact operational steps, but to provide the LLM with a high-level strategic guidance. We believe that the directives only need to clearly convey the intent of "Exploration" or "Exploitation". Minor variations in wording have little impact on the final performance.
>
> 3.Meanwhile, our method randomly samples a directive from the pool. It inherently smooths out any minor differences caused by variations in wording.
>
> It is worth noting that ablating different wordings would involve a vast experimental space. It may be vary challenging both in terms of experimental design and computational cost. From what we know, other LLM-based AHD methods have not conducted ablation studies on wordings either.
>
> > **Q4: Concerns about transferred knowledge.** The framework's knowledge management is confined to a single task; can the learned insights be generalized or transferred to new, unseen problem domains?
>
> This is an insightful question. Achieving cross-domain generalization of learned knowledge is indeed one of the long-term goals of LLM-based AHD. Our current study intentionally focuses on validating the effectiveness of the knowledge-management loop within a single task (intra-task). We view this as a crucial first step before establishing the feasibility of broader cross-domain generalization.
>
> More importantly, we have strong reasons to believe that our framework forms a solid basis for achieving this long-term goal. The "insights" stored in the Insight Pool are highly abstract, focusing on "how to optimize" rather than on any specific task. Because they are decoupled from task-specific code, these strategic insights hold strong potential for cross-domain transfer. As a result, HiFo-Prompt's knowledge management mechanism provides a robust foundation for future work on zero-shot and few-shot transfer of algorithmic knowledge across domains.

---

> ### Author Response · Authors · 2025-11-26
> **Response (3/3)**
>
> > **Q5: Concerns about illustrations.** What does the final algorithm look like? and how the insights and foresight prompts contribute to the generation of better heuristics, could you provide example illustrations?
>
> Thank you for this suggestion. As noted in our response to Weakness 2, we will add a dedicated section in the appendix to address this point.
>
> In our response to Reviewer GjDu's Q2, we present portions of the generated algorithm. Due to space limitations, the results for the remaining problems are included in the appendix.
>
> > **Q6: Concerns about comparison with related works**. A discussion and comparison with related works on prompt evolution and hierarchical search is suggested [1-3].
>
> Thank you for your question. We provide detailed answers below and will incorporate these discussions into the main text.
>
> 1.We would first like to differentiate HiFo-Prompt from MeLA [1], LLM-LNS [2], and EvoPH [3] based on the control mechanism. Existing methods rely on reactive strategies using lagging indicators. We specifically point out that LLM-LNS uses a passive trigger, updating the prompt strategy only after the population performance has stagnated for several generations, which inevitably delays the response to local optima. Furthermore, MeLA and EvoPH rely on retrospective feedback, adjusting the strategy based solely on the success rates or fitness scores of the previous generation. We emphasize that relying exclusively on performance metrics fails to diagnose the state of the search. In contrast, HiFo-Prompt establishes a proactive control strategy via the Foresight module. We introduce a metric for semantic diversity to monitor population health. By tracking this metric and fitness trends in real-time, our Evolutionary Navigator actively intervenes before stagnation occurs, issuing explicit directives to switch between "Exploration" and "Exploitation" regimes.
>
> 2.We then emphasize the difference in knowledge management between our method and the baselines. MeLA [1] and EvoPH [3] primarily rely on iterative rewriting or context accumulation. In these frameworks, historical experience is encoded in a continuously updated context or a single mutable prompt. We further point out that this approach risks "Knowledge Decay," where effective principles found early are easily overwritten or forgotten as the conversation grows. HiFo-Prompt overcomes this by decoupling implementation from design principles via the Hindsight module. Instead of rewriting the entire prompt, we use a structured Insight Pool to extract and store specific, successful design rules. For each generation, we retrieve only the most relevant insights based on utility scores. This ensures high-value strategies are persistently reused without being overwritten by recent updates, realizing cumulative learning.
>
> 3.Finally, we highlight the distinct advantages of HiFo-Prompt in terms of architectural stability and efficiency. We point out that MeLA [1] and EvoPH [3] rely on iterative prompt rewriting or context accumulation to evolve strategies. This unstructured approach often leads to high volatility in instruction quality and excessive token consumption due to lengthy context requirements. Similarly, LLM-LNS [2] employs a dual-population co-evolution framework, which results in high computational overhead and implicit "black-box" control. In contrast, HiFo-Prompt adopts Guided Prompt Synthesis. We dynamically assemble prompts using fixed operators, verified insights from Hindsight, and explicit directives from Foresight. This modular structure guarantees the stability of the instruction format while allowing for flexible content. Additionally, our approach significantly reduces token consumption compared to the context-heavy methods of MeLA and EvoPH, and avoids the double-population computational expense of LLM-LNS.manuscript.
>
> [1] Qiu Z, Chen X, Chen L, et al. Mela: A metacognitive llm-driven architecture for automatic heuristic design. arXiv preprint arXiv:2507.20541, 2025.
>
> [2] Ye H, Xu H, Yan A, et al. Large Language Model-driven Large Neighborhood Search for Large-Scale MILP Problems. ICML, 2025.
>
> [3] Liu Y, Li J, Zhao W X, et al. Experience-Guided Reflective Co-Evolution of Prompts and Heuristics for Automatic Algorithm Design. arXiv preprint arXiv:2509.24509, 2025.
>
> > **Q7: Concerns about descriptions.** There are typos and inadequate descriptions. Figure 1, The Left and Right can be misleading
>
> Thank you very much for pointing out these issues in the manuscript. We sincerely apologize for the oversight. We will carefully review the entire paper in the revised version to prevent similar problems from occurring again. We also apologize for the confusion caused by the figure annotations, and we will revise them to ensure that the descriptions are clearer and more intuitive.

---

### Official Review · Reviewer_99Cj · 2025-10-30

**Soundness:** 2
**Presentation:** 2
**Contribution:** 2
**Rating:** 4
**Confidence:** 4

**Summary:**

The paper proposes HiFo-Prompt, a prompting framework for LLM-based automated heuristic design that marries two modules: Foresight (an Evolutionary Navigator that steers exploration vs. exploitation from population signals) and Hindsight (an Insight Pool that distills and reuses design principles from successful code across generations). By decoupling “thoughts” from code, HiFo-Prompt supplies state-aware guidance and persistent knowledge. Experiments on TSP, FSSP, online bin packing, and black-box functions show state-of-the-art quality, faster convergence, and lower token/time cost than prior AHD systems; ablations confirm both modules matter.

**Strengths:**

- Dual Foresight/Hindsight design elevates the LLM from code generator to meta-optimizer.
- Evaluation sees evident performance gain.

**Weaknesses:**

- It’s unclear how you ensured a fair comparison under “the same query budget.” Does distilling insights consume additional queries? How many times did you run your method and the baselines? Did you use the same number of heuristic evaluations? Standard deviations are not reported, so the performance gains are not fully convincing.

- The approach involves many hyperparameters. It’s unclear how they were chosen and how robust the method is to their settings.

- The method relies heavily on pre-engineered prompts.

- Similar ideas appear in EoH and ReEvo, where thoughts and reflections are distilled (both) and accumulated (the latter). Please clarify the novelty relative to these.

**Questions:**

See weaknesses.

---

> ### Author Response · Authors · 2025-11-26
> **Response (1/3)**
>
> > **W1: Concerns about experimental fairness.**
>
> We are grateful for your consideration of experimental fairness. We would like to clarify the following aspects.
>
> 1.The expression "the same query budget" could potentially lead to misunderstanding. We would like to clarify its exact meaning. To fairly reproduce the baselines, we strictly followed the settings reported in their original papers. By "the same" we mean that our reproduction preserves consistency with the original experimental conditions and all methods use the same LLM during training. We apologize for the lack of clarity in this statement and will correct it in the revised manuscript.
>
> 2.Knowledge distillation requires additional queries. These queries are used to extract reusable design principles (insights) from the elite individuals in the current population. The cost of these queries is very low and fully controllable. It is worth noting that although additional queries are introduced, our method converges in far fewer generations than others. So the total number of API calls is lower, which is an overall resource advantage. As shown in Table 10 of Appendix C.7, HiFo-Prompt incurs significantly lower total token consumption and runtime compared to EoH and MCTS-AHD.
>
> 3.To ensure experimental reproducibility, all methods (HiFo-Prompt and baselines) were independently run three times, with reported results averaged over these runs. We acknowledge that omitting standard deviations in the initial draft reduces the conclusiveness of the results. We present some complete results below. The avg. (average) column reports the average value of the objective function achieved by each method across multiple independent runs. The revised version will include standard deviations to demonstrate the stability and statistical significance of our method's improvements.
>
> | Task| Method| avg.| std.|
> |-|-|-|-|
> | TSP50-construction      | EoH | 6.441 | 0.09887534    |
> | | ReEvo | 6.294 | 0.02821347    |
> | | HsEvo | 6.307 | 0.04491102    |
> | | MCTS-AHD | 6.317 | 0.03742103    |
> | | Ours | 6.088 | 0.04700355    |
> | TSP100-GLS | EoH | 7.738 | 0.00034451    |
> | | ReEvo | 7.740 | 0.00173957    |
> | | HSEvo | 7.743 | 0.00894281    |
> | | Ours | 7.737 | 0.00007013    |
> | Online BPP 5k_C100 | EoH | 2027  | 1.11355287    |
> | | ReEvo | 2022  | 0.60880932    |
> | | HsEvo | 2045  | 37.59627641   |
> | | MCTS-AHD   | 2026  | 0.66901381    |
> | | Ours | 2020  | 0.64291005    |
> | FSSP-GLS n100_m10 | EoH | 5643  | 2.60832002    |
> | | Ours | 5637  | 1.67470225    |
>
> 4. We are not entirely sure in which specific context your comment refers to "heuristic evaluations". In general, heuristic evaluations occur in both the training and inference phases.
> + During the training phase, every generated heuristic individual should be evaluated. In that case, our method uses fewer heuristic evaluations than baselines. Specifically, our evaluation count is:
> 8 (iterations) × 5 (operators) × 8 (population size) = 320 evaluations.
> In contrast, a typical baseline setting (using EoH as an example) is:
> 20 ((iterations) × 5 (operators) × 10 (population size) = 1000 evaluations. For tasks other than TSP construction, EoH uses a population size of 20.
> Thus, our method achieves superior performance while using only about one-third of the evaluation cost.
> + During inference, to ensure fairness, we use the same number of heuristic evaluations for all methods. Since every method runs three times, there will produce three heuristic individuals per method. In the inference stage, we evaluate three heuristics on the same test instances and report the average performance in the paper.

---

> ### Author Response · Authors · 2025-11-26
> **Response (2/3)**
>
> > **W2: Concerns about hyperparameters.**
>
> Thank you for your consideration to the hyperparameter settings. We offer a detailed clarification below.
>
> In the Hindsight module, there are several parameters:
> - We set $\alpha = 0.3$ to balance history and new feedback. It allows the insight score to update smoothly and avoids instability under the short 8-generation evolution.
> - We set $w_u = 0.1$ and $\tau_r = 0.2$ in $B_r$ to balance exploration and exploitation: the mild penalty from $w_u$ prevents premature convergence without suppressing valuable insights, while $B_r$ gives recently successful insights a short-lived boost to encourage rapid short-term exploitation.
> - The time-decay rate $R_{decay} = 0.01$ ensures a steady turnover of insights in the pruning mechanism. It gradually removes outdated or long-ineffective knowledge, preventing the pool from becoming rigid in later stages of evolution.
> - The threshold $T_{usage}=3$ provides a minimal but reliable observation window. Fewer uses make the evaluation too noisy, while more would waste valuable evaluations in our fast-converging setting.
>
> In the Hindsight module, there are several parameters:
> - We set $\tau_{stag} = 3$, because three stagnant rounds already indicate diminishing returns within only 8 generations. Given the high cost of LLM queries, this threshold allows rapid detection of plateaus and prevents wasted exploration.
> - We set the diversity threshold $\delta_p = 0.3$. If over 70\% of the population becomes similar, the threshold triggers early injection of new ideas, maintaining evolutionary diversity and preventing premature convergence.
>
> In the general configuration, we set the following parameters:
> - We set the population size to 8 to balance necessary diversity and efficiency. Smaller populations (e.g., 4) limit the Hindsight module’s ability to extract high-quality insights, while larger populations (e.g., 12) increase LLM query and evaluation costs with limited performance gains. Importantly, in our framework, even a small population maintains diversity because the LLM generates new individuals creatively from high-level insights, and slight prompt variations produce fundamentally different algorithmic ideas.
> - We set the maximum generations to 8 to balance rapid convergence and efficiency. Fewer generations risk underutilizing insights, while more offer minimal gains with higher costs. Eight generations suffice for high-quality convergence.
> - We set the insight pool size to 30 to balance stability and agility. Smaller pools risk losing valuable insights too early, while larger pools dilute core knowledge with outdated ones. Thirty provides a good trade-off, retaining effective strategies while staying responsive to new insights.
>
> Finally, **Appendix C.6** (Figure 4) presents a sensitivity analysis, showing that these parameter choices yield stable and strong performance within a nearby range. Besides that, some additional ablation studies are shown below. This confirms that our settings are both theoretically justified and practically effective.
>
> | Setting             | run1  | run2  | run3  | avg.  |
> |---------------------|-------|-------|-------|-------|
> | $w_u$ = 0             | 5.958 | 5.989 | 5.978 | 5.975 |
> | $w_u$ = 0.05          | 5.894 | 5.901 | 5.907 | 5.901 |
> | $w_u$ = 0.1 (default)    | 5.885 | 5.838 | 5.873 | 5.866 |
> | $w_u$ = 0.3           | 5.921 | 5.907 | 5.922 | 5.917 |
> | $\tau_r$ = 0             | 5.954 | 5.972 | 5.979 | 5.968 |
> | $\tau_r$ = 0.1           | 5.920 | 5.925 | 5.901 | 5.915 |
> | $\tau_r$ = 0.2 (default)    | 5.885 | 5.838 | 5.873 | 5.866 |
> | $\tau_r$ = 0.4           | 5.937 | 5.945 | 5.933 | 5.938 |
> | $R_{decay}$ = 0         | 5.998 | 6.097 | 6.075 | 6.057 |
> | $R_{decay}$ = 0.005     | 5.899 | 5.908 | 5.906 | 5.904 |
> | $R_{decay}$ = 0.01 (default) | 5.885 | 5.838 | 5.873 | 5.866 |
> | $R_{decay}$ = 0.04      | 5.985 | 5.956 | 5.967 | 5.969 |

---

> ### Author Response · Authors · 2025-11-26
> **Response (3/3)**
>
> > **W3: Concerns about prompts design.**
>
> We clarify that careful prompt design is standard practice in recent LLM-based AHD methods [1,2,3] to ensure framework functionality, yet our method minimizes reliance through dynamic mechanism design.
>
> First, we adopted standardized templates from EoH [1] as a structural skeleton to minimize the influence of manual prompt engineering, thereby isolating our contribution to the dynamic mechanism rather than the static templates.
>
> Second, components like Seed Insights and Design Directives serve as dynamic scaffolding rather than fixed solutions. Specifically, Seed Insights are used strictly to mitigate cold starts, where all subsequent insights are autonomously distilled, while Directives act as a generic pool for real-time sampling.
>
> Furthermore, we applied identical templates across disparate domains (TSP, Bin Packing, FSSP) without manual tuning, demonstrating that our design does not rely on problem-specific engineering.
>
> Finally, the ablation study in Table 5 confirms that relying solely on pre-engineered templates without our dynamic mechanism degrades performance, proving that efficacy stems from autonomous adaptation.
>
>
> [1] Liu F, Tong X, Yuan M, et al. Evolution of heuristics: Towards efficient automatic algorithm design using large language model. ICML, 2024.
>
> [2] Ye H, Wang J, Cao Z, et al. Reevo: Large language models as hyper-heuristics with reflective evolution. NIPS, 2024.
>
> [3] Zheng Z, Xie Z, Wang Z, et al. Monte carlo tree search for comprehensive exploration in llm-based automatic heuristic design. ICML, 2025.
>
> > **W4: Concerns about comparison with other methods.**
>
> We thank the reviewer for the opportunity to clarify the distinctions relative to EoH and ReEvo. Our contributions lie in the architectural transition from passive or local operations to active, global management of both knowledge and evolution strategy.
>
> 1.Regarding knowledge management (Hindsight), we first address the limitations found in prior works. In EoH, algorithmic reasoning is coupled with specific code implementations; consequently, when an individual is discarded during selection, the associated reasoning is permanently lost—a phenomenon we term Knowledge Decay. Furthermore, regarding ReEvo, we point out distinct issues in its reflection mechanism[1]: its short-term reflection relies on local pairwise comparisons, which often fail to extract valid improvement directions when performance differences are subtle or stochastic; meanwhile, its long-term reflection relies on a linear summarization of recent experiences. This leads to Knowledge Dilution, where early, high-value insights are often overshadowed by later, mediocre reflections. In contrast, HiFo-Prompt decouples insights from specific code implementations. By applying a utility-based credit assignment mechanism to reinforce proven principles and prune ineffective ones, we ensure the knowledge base actively evolves and improves, rather than merely accumulating static information.
>
> 2.Regarding evolutionary control (Foresight), we wish to emphasize that HiFo-Prompt introduces a global, efficient control mechanism compared to the passive or local approaches of baselines. Specifically, EoH relies on random evolutionary operators, which necessitate large population sizes and extensive iterations to maintain diversity, resulting in frequent LLM queries and high computational costs. Additionally, while ReEvo attempts to guide the search using "verbal gradients" from local comparisons, our empirical observations (particularly on Online Bin Packing) indicate that this mechanism frequently fails to escape local optima. Conversely, our Foresight module functions as a global regulator. It monitors population-level dynamics—specifically diversity collapse and performance stagnation—rather than focusing solely on individual fitness. A decline in diversity signals that the LLM is producing redundant variations within a local region; in response, the system proactively switches to an exploration regime via explicit natural language instructions. This intervention prevents search stagnation, achieving superior sample efficiency and reduced computational costs.
>
> [1] Wu X, Wang D, Wu C, et al. Efficient heuristics generation for solving combinatorial optimization problems using large language models. KDD, 2025.

---

### Official Review · Reviewer_xien · 2025-10-31

**Soundness:** 2
**Presentation:** 2
**Contribution:** 2
**Rating:** 4
**Confidence:** 3

**Summary:**

HiFo-Prompt tackles two common gaps in LLM-based Automatic Heuristic Design (AHD): lack of global search control and poor knowledge persistence. It adds  a rule-based Foresight meta-controller that watches population progress/diversity and switches prompts among explore/exploit/balance regimes, and a Hindsight Insight Pool that distills reusable design principles from elites with utility-based credit assignment, then injects top-scoring insights into subsequent prompts. The method obtains the best results among various LLM-AHD baselines.

**Strengths:**

- The idea of tracking both local and global evolution dynamics via specialized modules is interesting and well executed

- Useful ablation studies

- Strong performance with few function evaluations

**Weaknesses:**

1. Seed insights are required by the method. Importantly, these insights could significantly improve generation quality: “Design adaptive hybrid meta-heuristics synergistically fusing multiple search paradigms and dynamically tune operator parameters based on search stage or problem features.” particularly is a high-quality handcrafted prompt that can have a substantial effect on the generation. For fairness of comparison, one should provide the same information in the prompt of other baselines, say EoH.

2. The novelty regarding global control and historical information aggregation is overstated, e.g., ReEvo already implements a short and long-term reflection that could be seen as a simpler version of hindsight. Discussions would be appreciated.

3. I am not convinced about the population size being chosen as 4. How can diversity be maintained in such as small population and avoid inbreeding?

4. I found the methodology section quite confusing, with many quite complicated implementations. For example, a decay rate is introduced, but there is no ablation or sensitivity analysis on it.  Eq. 3, which describes the evolutionary contribution, is full of hardcoded parameters, which are hard to parse, and the rationale for choosing them is not explained.

    1. On this point, please clarify whether $g$ is a minimization or maximization objective. In EoH, this is maximization, but Fig. 2 and equations suggest otherwise. However, section A.1 again takes $g$ as argmax. This is confusing.

5. No code is provided

**Questions:**

1. About dissimilarity (Eq 7): how are the textual descriptions calculated, and how do you ensure these are the same? (e.g.: will changing a single word make two descriptions different?)

2. It appears that there is a massive degradation if Qwen 2.5 max is not used in Table 9. How do you explain this?

3. What would happen if baseline methods also have the seed insights as part of the generator prompt?

---

> ### Author Response · Authors · 2025-11-26
> **Response (1/3)**
>
> > **W1: Concerns about seed insight.**
>
> Thank you so much for your time and effort in reviewing our work. We address the concerns as follows.
>
> 1.The seed insights merely mitigate cold start and initialize the insight pool, as noted in Section 3.2. One of the key innovations of HiFo-Prompt is that its insight pool is dynamically self-evolving, meaning that the initial seed insights are quickly superseded by insights learned autonomously during evolution. Specifically, the insight pool has a limited capacity and is equipped with a utility and time-decay–based pruning mechanism. As a result, any insight that is ineffective, contributes little, or remains unused over time (including the initial seed insights) is automatically replaced during the process, ensuring that the pool consistently retains only the most valuable knowledge.
>
> 2.As suggested by the reviewer, we conducted additional experiments to eliminate the effect of the seed insights. Using TSP-construction as an example, we put the seed insights into EoH and MCTS-AHD by adding them to the initialization operator of i1, and into ReEvo and HSEvo by placing them in their external knowledge files, while keeping all other settings unchanged. The resulting performance is shown below.
>
> | Method| tsp10 | tsp20 | tsp50 | tsp100 | tsp200 |
> |-|-|-|-|-|-|
> | EoH (w/ seed insight)| 3.024 | 4.233 | 6.512 | 8.998  | 12.470 |
> | EoH (w/o seed insight)| 3.049 | 4.252 | 6.441 | 8.990  | 12.514 |
> | ReEvo (w/ seed insight)| 2.993 | 4.132 | 6.344 | 8.759  | 12.198 |
> | ReEvo (w/o seed insight)| 2.994 | 4.126 | 6.294 | 8.773  | 12.324 |
> | HsEvo (w/ seed insight)| 3.064 | 4.324 | 6.452 | 8.842  | 12.270 |
> | HsEvo (w/o seed insight)| 3.001 | 4.170 | 6.307 | 8.729  | 12.183 |
> | MCTS-AHD (w/ seed insight)| 2.938 | 4.142 | 6.294 | 8.724  | 12.076 |
> | MCTS-AHD (w/o seed insight)| 2.983 | 4.174 | 6.317 | 8.769  | 12.176 |
>
> We observe that injecting our seed insights into these baselines does not lead to any meaningful improvement. It indicates that the seed insights contribute only marginally to performance. The superior performance of our method primarily comes from the automatic insight updating and management mechanism in our insight pool, rather than from the initial seed insights.
>
> > **W2: Concerns about global control and historical information aggregation.**
>
> Thanks for your questions. We acknowledge that both our method and ReEvo aim to learn from historical experience, but there are fundamental differences between HiFo-Prompt and ReEvo.
>
> 1.ReEvo’s "reflection" operates in a local and reactive manner. It generates temporary reflective experiences based on pairwise comparisons of individuals, and its effectiveness heavily depends on the LLM's ability to attribute improvements from a single comparison. This leads to two core issues:
>
> (1) Reflection failure[1]. When simple success/failure comparisons provide insufficient information, the LLM struggles to extract genuinely valuable improvement directions, especially in tasks like Online BPP. So, the reflection often degenerates into blind recombination of the original ideas.
>
> (2) Knowledge dilution. Its "long-term reflection" is formed by linearly summarizing recent experiences, making early critical reflections easily diluted by subsequent mediocre or noisy reflections.
>
> In contrast, the Hindsight module constructs a persistent and structured knowledge base aimed at cumulative knowledge evolution.
>
> (1) First, our "insights" are not derived from arbitrary comparisons, but are distilled from the design experiences of the elite population.
>
> (2) Second, the Insight Pool employs quantitative management to continuously evaluate and amplify design principles that have been proven effective over the long term, thereby effectively mitigating knowledge dilution. The core function of Hindsight is to iteratively evolve superior insights to guide subsequent generations, which fundamentally differs from ReEvo's reliance on single-shot reflections.
>
> 2.ReEvo attempts to guide the search via so-called "language gradients" derived from its reflections. These gradient signals, generated from local pairwise comparisons, are inherently unstable. In contrast, the Foresight module does not rely on local gradient signals; instead, it globally diagnoses the evolutionary state by monitoring dynamic population-level indicators, such as stagnation and diversity. This system-state-based assessment enables proactive switching between exploration and exploitation, providing the framework with robust control to mitigate risks such as premature convergence and diversity collapse.
>
> [1] Wu X, Wang D, Wu C, et al. Efficient heuristics generation for solving combinatorial optimization problems using large language models. KDD, 2025.

---

> ### Author Response · Authors · 2025-11-26
> **Response (2/3)**
>
> > **W3: Concerns about population size.**
>
> We acknowledge that concerns regarding small population sizes, specifically regarding inbreeding and premature convergence, are well-founded in traditional evolutionary computation. However, we would like to clarify two critical points.
>
> 1.First, the population size of 4 mentioned in the paper refers to early exploratory experiments aimed at probing efficiency limits. In our subsequent extensive experiments, we standardized the population size to 8, which effectively balances performance and cost. This decision is also supported by our parameter sensitivity analysis in Figure 4. We sincerely apologize for the delay in updating this section.
>
> 2.Second, and more importantly, our method leverages the LLM as a semantic operator rather than relying on the statistical distribution of large populations. Since LLMs are highly sensitive to prompts, our Foresight module utilizes the Evolutionary Navigator to actively monitor the semantic diversity of the algorithms within the population via the phenotypic distance metric $\Delta_p$ (Eq. 7). Upon detecting homogenization, it triggers an Explore regime (Eq. 6) that injects explicit constraints into the context. This approach forces the generation of offspring with distinct logic and code structures even from identical parent individuals. This semantic creativity surpasses the local perturbations of traditional operators, thereby algorithmically preventing inbreeding.
>
> Furthermore, the Hindsight module decouples knowledge accumulation from immediate population capacity via the Insight Pool, ensuring generation relies on a diverse history rather than just immediate parents. This synergy ensures that the generative process is supplied with exogenous informational entropy and strategic guidance, preventing the recursive narrowing of the search space characteristic of standard small-population setups.
>
> > **W4: Concerns about methodology.**
>
> Thank you very much for your question. We will address your concerns as follows.
>
> 1.To investigate the effect of hyperparameters in our method, we performed an additional ablation study on the decay rate. Using TSP50-construction as an example, the results are presented in the table below.
>
> | decay rate     | run1     | run2     | run3     | avg.     |
> |----------------|----------|----------|----------|----------|
> | 0              | 5.998    | 6.097    | 6.075    | 6.057    |
> | 0.005          | 5.899    | 5.908    | 5.906    | 5.904    |
> | 0.01 (default) | 5.885    | 5.838    | 5.873    | 5.866    |
> | 0.04           | 5.985    | 5.956    | 5.967    | 5.969    |
>
> 2.We would like to clarify the motivation for Eq. 3 and provide a detailed explanation. We chose the piecewise function in Eq. 3 rather than a continuous function[1], achieve a clearer incentive guidance. Evolutionary contributions are nonlinear: breakthroughs are far more valuable than incremental improvements. The parameters (0.8 for breakthroughs, 0.2 for improvements, –0.3 for harmful attempts) define explicit reward boundaries, simplifying credit assignment and providing effective learning signals for our framework.
>
> 3.We thank the reviewer for pointing out the inconsistency in the description of the objective function $g$ in our paper. This was a typographical oversight. We sincerely apologize for this oversight. All optimization problems in our experiments are minimization problems, with the objective function denoted as $f$. However, within the evolutionary computation framework, fitness evaluations are typically framed as maximization. Therefore, we defined the fitness function as $g(h) = \mathbb{E}[-f(h(ins))]$, as shown in Equation 9 in the Appendix. Accordingly, $g(h)$ is a maximization objective, and the notation $h^* = \text{argmax} g(h)$ is correct.
>
> The inconsistency in Figure 2 and Equation 3 arises from this oversight: the Y-axis of Figure 2 actually shows the minimization objective values (lower is better), whereas Equation 3 incorrectly used the $\leq$ operator when comparing $g(h_{new})$. To address this, we will make the following corrections in the final version:
>
> + clarify that the Y-axis of Figure 2 represents the objective value (lower is better).
> + correct the comparison logic in Equations 2 and 3 to reflect that $g$ is a maximization target (e.g., using $g(h_{new}) ≥ g(h_{best})$ to indicate improvement).
> + thoroughly check the manuscript to ensure all related symbols and descriptions are consistent. We thank the reviewer again for this careful examination, which has greatly improved the clarity and rigor of our paper.
>
> [1] Tokic, M. Adaptive $\epsilon$-greedy Exploration in Reinforcement Learning Based on Value Differences. Annual conference on artificial intelligence, 2010.
>
> > **W5: Concerns about code.**
>
> Thank you for your attention about our method's code. We commit to making all code and experimental data publicly available upon acceptance of the paper, to ensure the reproducibility of our research.

---

> ### Author Response · Authors · 2025-11-26
> **Response (3/3)**
>
> > **Q1: Concerns about dissimilarity.**
>
> We clarify that the dissimilarity in Equation 7 is computed through exact string matching of generated algorithmic descriptions, meaning that even a one-word difference is treated as a distinct output. This design choice is motivated by two considerations:
>
> 1.Our standardized prompt template enforces a fixed syntactic format, eliminating trivial variations such as variable renaming or synonym replacement. Thus, any remaining lexical difference reflects an intentional modification by the LLM and reliably corresponds to a true structural change in the algorithm.[1]
>
> 2.Exact matching captures every logical alteration. In LLM-based AHD, a single keyword often signals a fundamentally different strategy—for example, "dynamic weighting" vs. "static weighting". Embedding-based similarity would cluster such cases due to lexical overlap[2], obscuring distinctions that may be crucial for escaping local optima.
>
> When the LLM becomes genuinely trapped in a local optimum, it stops producing even minor variations and repeatedly outputs identical text. We set the diversity threshold at 0.3 as a conservative indicator of this stagnation: a score below this level means that over 70% of descriptions are exact duplicates. This mechanism anticipates local-optimum collapse and triggers early exploration before the search fully stalls.
>
> [1] Gan C, Mori T. Sensitivity and robustness of large language models to prompt template in Japanese text classification tasks. ACL, 2023.
>
> [2] Truong T H, Baldwin T, Verspoor K, et al. Language models are not naysayers: an analysis of language models on negation benchmarks. ACL, 2023.
>
> > **Q2: Concerns about multiple LLMs comparison.**
>
> We clarify the reasons behind the performance differences in Table 9:
>
> 1.Table 9 reports the search performance on the training set, reflecting the model's upper-bound ability to discover new heuristics. The "massive degradation" shows that without Qwen 2.5-max, other models stagnate early and fail to reach peak performance during discovery. In contrast, the testing results (shown below) evaluate generalization on unseen data. Although Qwen 2.5-max still performs best, the gap is smaller, indicating that the largest difference lies in discovering solutions close to the optimal LKH3 baseline.
>
> | Method| TSP50-construction | TSP100-construction | TSP100-GLS | TSP200-GLS | Online BPP 1k,100 | Online BPP 10k,300 | FSSP-GLS j100 m10 | FSSP-GLS j200 m20 |
> |-|-|-|-|-|-|-|-|-|
> | Qwen-2.5-max| 6.625%  | 8.582%| 0.015%| 0.382%| 2.188%| 0.391%| 0.128%| 0.710%|
> | GPT-4o-mini| 7.058%  | 9.579%| 0.021%| 0.433%| 2.287%| 0.492%| 0.137%| 0.787%|
> | DeepSeek-v3| 7.717% | 10.455%| 0.041%| 0.480%| 2.238%| 0.978%| 0.185%| 0.916%|
> | Deepseek-r1| 10.412% | 12.638%| 0.074%| 0.589%| 2.984%| 1.487%| 0.313%| 1.485%|
> | claude3.5-sonnet | 8.276%| 11.087%| 0.061%| 0.454%| 2.089%| 1.487%| 0.179%| 1.256%|
>
> 2.Our framework requires creativity and diverse code mutations. Models like DeepSeek-R1 are optimized to produce a single correct answer and tend to revert changes back to standard, safe algorithms. Because they do not fully explore the requested variations, they struggle to improve the algorithm during evolutionary search, leading to degraded training performance.
>
> 3.Recent work shows that higher reasoning ability does not guarantee better heuristic discovery. The MOH paper [1] found GPT-4o-mini outperforming the stronger o1-mini due to better instruction following. The MCTS-AHD paper [2] reported Claude-3.5-sonnet performing worse than GPT-4o-mini, with a larger gap to the optimum. These cases suggest that success depends on alignment between model behavior and the search strategy, not just general capability.
>
> 4.These findings show that identifying an LLM that matches the evolutionary framework is key for strong AHD performance. No single model currently excels across all heuristic discovery tasks. Our results highlight that models like Qwen 2.5-max, which align well with evolutionary search, are essential for achieving competitive performance with baselines like LKH3. Thus, the observed "degradation" reflects the importance of model choice, not a limitation of our method.
>
> [1] Shi Y, Zhou J, Song W, et al. Generalizable Heuristic Generation Through Large Language Models with Meta-Optimization. arXiv preprint arXiv:2505.20881, 2025.
>
> [2] Zheng Z, Xie Z, Wang Z, et al. Monte carlo tree search for comprehensive exploration in llm-based automatic heuristic design. ICML, 2025.
>
> > **Q3: Concerns about additional experiments.**
>
> We showed the results when answering W1.

---

### Official Review · Reviewer_GjDu · 2025-10-31

**Soundness:** 2
**Presentation:** 3
**Contribution:** 3
**Rating:** 6
**Confidence:** 4

**Summary:**

The paper proposes HiFo-Prompt with (i) a Hindsight module that distills reusable principles from successful candidates, and (ii) a Foresight module that adaptively switches explore/exploit/balance based on population state to guide LLM-based AHD. The proposed method is evaluated on TSP, Online BPP, FSSP, and BO.

**Strengths:**

1. The proposed method is well motivated and outperforms recent LLM-based AHD baselines across several tasks.
2. The design details are well presented.
3. The limitations and future directions are clearly analyzed.

**Weaknesses:**

1. For TSP step-by-step construction (i.e., Table 1), Appendix B.1 states that HiFo-Prompt involves LLM calls at inference time, however, it is unclear to me that whether such strategy also applies to the baselines. Please disambiguate: (a) If baselines also call the LLM at inference, please explain why HiFo-Prompt’s runtime is longer; (b) If they do not, please also report HiFo-Prompt under the same inference protocol for fair comparisons.
2. The main text claims TSPLIB results are in Appendix C.1, but C.1 contains only descriptive text and a placeholder “Table ??”, with no actual results. Please add the promised table/metrics or revise the pointer.

**Questions:**

1. Line 387 says “100 instances at each of five sizes,” but Table 1 shows three, please fix the mismatch. Also, there are several misplaced “?” characters around lines 811, 854, 946, 967 that need cleanup.
2. Can you present some of the actual heuristics generated and used to produce the reported results?
3. In Table 5, removing the Insight Pool would make the method perform worse than EoH, which is surprising to me since the setup still retains the Foundational Prompts adapted from EoH and the Navigator module. Can you analyze the concrete differences between EoH and HiFo-Prompt w/o Insight Pool & Navigator that can explain this gap? Will the Navigator module improve baselines like EoH as a drop-in controller?
4. How frequently does the Navigator select explore or exploit across runs? Have you tried an ablation that fixes the state to “balance” throughout to isolate the benefit of adaptive switching?

---

> ### Author Response · Authors · 2025-11-26
> **Response (1/3)**
>
> > **W1: Concerns about LLM calls.**
>
> Thanks for your question. We would like to clarify that all baselines and our method don't involve LLM calls at inference time. Our method follows the same inference protocol as the baselines.
>
> All LLM-based AHD methods(including ours) invoke the LLM only during the training phase to generate high-quality heuristic code. During the inference phase, the results (i.e., Table 1) are obtained by executing the generated heuristics that contain no LLM calls. Therefore, both our method and the baselines don't call the LLM during inference phase, ensuring a fair comparison.
>
> The statements in Appendix B.1 are intended to illustrate the role of the LLM in the decision-making logic during the training phase for TSP step-by-step construction, rather than implying that the LLM is called to solve TSP instances during the inference phase.
>
> We apologize for the confusion caused by our unclear descriptions. In the revised manuscript, we will explicitly distinguish between the training and inference phases, and clearly indicate that no method invokes the LLM during inference.
>
> > **W2:  Concerns about TSPLIB results.**
>
> We thank the reviewer for pointing out this oversight. This is entirely an unintentional error during manuscript preparation, and we apologize for it.
> The results on TSPLib are as follows. This table provides a comparison of our method with the baselines across various instances on TSPLib. We commit to including these experimental results in the revised manuscript.
>
> | name| EoH| MCTS-AHD| ReEvo| Hsevo| Ours|
> |-|-|-|-|-|-|
> | eil51| 7.665%   | 17.133%  | 4.409%   | **4.295%** | 6.691%   |
> | berlin52| 16.080%  | 17.481%  | 15.678%  | 19.814%  | **10.973%** |
> | eil76| 10.566%  | 15.193%  | 8.450%   | 10.034%  | **8.439%** |
> | pr76| 25.222%  | 29.371%  | 21.089%  | 21.718%  | **20.505%** |
> | kroA100| 24.347%  | 24.315%  | 17.509%  | 18.758%  | **16.616%** |
> | kroB100| 30.274%  | 22.593%  | **9.059%** | 13.266%  | 20.502%  |
> | kroC100| 25.542%  | 10.212%  | 27.343%  | 36.176%  | **10.000%** |
> | kroD100| 31.497%  | 25.765%  | 25.797%  | **13.258%** | 32.315%  |
> | kroE100| 24.618%  | 21.299%  | 23.905%  | 22.218%  | **14.925%** |
> | rd100| 12.654%  | 11.470%  | 8.660%   | 12.995%  | **7.921%** |
> | eil101| 14.421%  | 22.947%  | 12.163%  | 12.372%  | **7.229%** |
> | lin105| 40.297%  | 16.797%  | 27.774%  | 42.519%  | **13.974%** |
> | pr107| 9.900%   | 5.004%   | 7.894%   | 7.537%   | **3.811%** |
> | ch130| 21.406%  | **6.930%** | 7.352%   | 13.107%  | 7.090%   |
> | ch150| 19.604%  | 9.874%   | 12.644%  | 10.185%  | **4.275%** |
> | kroA150  | 27.891%  | 20.517%  | 25.887%  | 21.869%  | **20.070%** |
> | kroB150  | 22.403%  | 27.716%  | 29.499%  | 32.729%  | **13.677%** |
> | pr152| 12.302%  | 10.457%  | 12.473%  | 11.960%  | **6.545%** |
> | u159| 13.444%  | **7.074%** | 7.951%   | 13.644%  | 7.828%   |
> | kroA200| 26.395%  | 26.219%  | 27.366%  | 28.537%  | **22.702%** |
> | kroB200| 20.980%  | 20.083%  | 23.861%  | 21.590%  | **14.314%** |
> | tsp225| 32.938%  | 20.193%  | 21.703%  | 22.503%  | **18.683%** |
> | a280| 34.081%  | 24.095%  | 27.921%  | 31.736%  | **15.936%** |
> | rd400| 14.612%  | 14.745%  | 13.865%  | 15.213%  | **8.075%** |
> | d493| 18.833%  | 13.248%  | 14.776%  | 21.179%  | **7.976%** |
> | avg. | 21.519%  | 17.629%  | 17.401%  | 19.168%  | **12.843%** |
>
> > **Q1: Concerns about TSP-construction results.**
>
> We thank the reviewer for the careful review and for pointing out these unintentional errors.
>
> During our experimental process, we actually tested across five different scales. However, due to space limitations in the main text, only results for three scales were presented. Unfortunately, the descriptions in the manuscript were not updated accordingly, and we sincerely apologize for this oversight. In the revised version, we will correct the descriptions in the main text and present the additional results for the other two scales in the appendix. We will also carefully review the manuscript for any mismatched or inconsistent statements and correct them accordingly.
>
> |method|TSP10 Gap|time(s)|TSP20 Gap|time(s)|TSP50 Gap|time(s)|TSP100 Gap|time(s)|TSP200 Gap|time(s)|
> |-|-|--|-|-|-|-|-|-|-|-|
> |LKH3|0.000%|6.492|0.000%|24.9|0.000%|323.3|0.000%|1450|0.000%|6312|
> |POMO|0.246%|—|0.248%|—|0.163%|—|1.636%|—|13.961%|—|
> |LEHD|0.183%|—|0.010%|—|0.117%|—|0.452%|—|0.367%|—|
> |EoH|7.148%|0.042|10.064%|0.1|12.820%|1.3|15.361%|9|16.658%|78|
> |ReEvo|5.227%|0.228|6.811%|1.2|10.239%|21.5|12.577%|224|14.890%|3013|
> |HSEvo|5.461%|0.689|7.950%|3.2|10.467%|89.4|12.008%|1286|13.578%|24835|
> |MCTS-AHD|4.829%|0.440|8.045%|4.1|10.642%|91.4|12.521%|1084|13.510%|14521|
> |Ours|**1.654%**|0.709|**3.619%**|12.9|**6.625%**|244.7|**8.582%**|1843|**8.877%**|16099|

---

> ### Author Response · Authors · 2025-11-26
> **Response (2/3)**
>
> > **Q2: Concerns about heuristic demonstration.** Can you present some of the actual heuristics generated and used to produce the reported results?
>
> This is a great suggestion. Providing code examples can enhance the transparency of our method. We commit to adding a subsection in the appendix of the revised manuscript to show heuristic algorithms generated by HiFo-Prompt. Here are some examples generated by our method on TSP-construction and Online BPP. Due to space limitations, the heuristic results for the remaining problems are included in the appendix.
>
> - One of examples on TSP-construction:
> ```
> import numpy as np
> def select_next_node(current_node, destination_node, unvisited_nodes, distance_matrix):
>     def calculate_pruned_mst_cost(nodes):
>         if len(nodes) <= 1:
>             return 0
>         edges = [(distance_matrix[i][j], i, j) for i in nodes for j in nodes if i < j]
>         edges.sort(key=lambda x: x[0])
>         parent = {node: node for node in nodes}
>
>         def find(u):
>             while parent[u] != u:
>                 parent[u] = parent[parent[u]]
>                 u = parent[u]
>             return u
>
>         mst_cost = 0
>         for cost, u, v in edges:
>             root_u, root_v = find(u), find(v)
>             if root_u != root_v:
>                 parent[root_v] = root_u
>                 mst_cost += cost
>         return mst_cost
>
>     def perform_cluster_aware_simulation(node):
>         future_nodes = [n for n in unvisited_nodes if n != node]
>         total_cost = distance_matrix[current_node][node]
>         current = node
>         while future_nodes:
>             next_candidate = min(future_nodes, key=lambda x: distance_matrix[current][x] / (np.median(distance_matrix[x]) + 1e-9))
>             total_cost += distance_matrix[current][next_candidate]
>             current = next_candidate
>             future_nodes.remove(current)
>         total_cost += distance_matrix[current][destination_node]
>         return total_cost
>
>     def evaluate_candidate(node):
>         future_nodes = [n for n in unvisited_nodes if n != node]
>         pruned_mst_cost = calculate_pruned_mst_cost(future_nodes + [destination_node])
>         normalized_cost = distance_matrix[current_node][node] / (np.max(distance_matrix[current_node]) + 1e-9)
>         simulation_cost = perform_cluster_aware_simulation(node)
>         weight_mst = 0.4 - 0.15 * (len(unvisited_nodes) / len(distance_matrix))
>         weight_normalized = 0.3 + 0.1 * (len(unvisited_nodes) / len(distance_matrix))
>         weight_simulation = 0.3
>         return weight_mst * pruned_mst_cost + weight_normalized * normalized_cost + weight_simulation * simulation_cost
>
>     beam_width = max(2, min(6, len(unvisited_nodes) // 3))
>     candidates = sorted(unvisited_nodes, key=lambda x: distance_matrix[current_node][x] / (np.mean(distance_matrix[current_node]) + 1e-9))[:beam_width * 3]
>
>     scored_candidates = []
>     for candidate in candidates:
>         eval_score = evaluate_candidate(candidate)
>         scored_candidates.append((eval_score, candidate))
>
>     next_node = min(scored_candidates, key=lambda x: x[0])[1]
>     return next_node
> ```
>
> - One of examples on Online BPP:
> ```
> import numpy as np
> def score(item, bins):
>     # Compute residuals after placing the item
>     residuals = bins - item
>
>     # Stochastic perturbation for enhanced exploration
>     perturbation = np.random.normal(1.0, 0.2, size=len(bins))
>
>     # Cubic residual scoring to emphasize finer space usage precision
>     residual_scores = 1 / (1 + residuals ** 3) * perturbation
>
>     # Delayed penalty adjustment based on long-term impact of underutilization
>     threshold = np.mean(bins) * 0.4  # Adaptive threshold with increased sensitivity
>     delayed_penalty = np.where(residuals > threshold, 1 / (1 + residuals ** 2), 0)
>
>     # Reward function for bins maintaining balanced utilization over time
>     balanced_utilization_reward = np.exp(-np.abs(residuals - threshold) ** 2 / (threshold ** 2))
>
>     # Dynamic scaling factor based on cumulative fit ratio and historical trends
>     fit_ratios = item / bins
>     scaling_factor = 0.6 + 0.4 * np.tanh(fit_ratios * 5)  # Sharper dynamic scaling
>
>     # Combine scores with refined adaptive weighting emphasizing long-term efficiency
>     scores = scaling_factor * (residual_scores + balanced_utilization_reward) - delayed_penalty
>
>     return scores
> ```

---

> ### Author Response · Authors · 2025-11-26
> **Response (3/3)**
>
> > **Q3: Concerns about ablation study.**
>
> This is a insightful question that prompted us to think more deeply about the interplay among the components of our framework.
>
> 1.Regarding your concern, we clarify that our method operates under a small-population setting (population size = 8), whereas EoH uses a population of 20. Therefore, even if we remove the Insight Pool (i.e., keeping only the navigator), EoH naturally benefits from having more individuals and will outperform our reduced variant. This outcome is expected and entirely reasonable given the population-size advantage.
>
> 2.The Navigator module can be a drop-in controller to improve other methods. Following your suggestion, we conducted an additional experiment using the TSP-construction as an example by integrating our navigator into EoH, with the results presented below.
>
> | Task   | run1  | run2  | run3  | avg.  | EoH    |
> |--------|-------|-------|-------|-------|--------|
> | TSP10  | 2.982 | 3.012 | 2.930 | **2.975** | 3.049  |
> | TSP20  | 4.106 | 4.228 | 4.068 | **4.134** | 4.252  |
> | TSP50  | 6.250 | 6.364 | 6.247 | **6.287** | 6.441  |
> | TSP100 | 8.639 | 8.823 | 8.607 | **8.690** | 8.990  |
> | TSP200 | 12.000| 12.227| 11.977| **12.068**| 12.514 |
>
> 3.In addition, the navigator operates as a global controller whose primary role is to monitor and adjust population-level dynamics. Without high-quality insights to guide it, its control signal becomes less effective, leading to only marginal improvements. We view our approach as a tightly integrated framework rather than a collection of loosely assembled components. While individual modules can provide some gains, it is the full system working in concert that delivers the strongest performance.
>
> > **Q4: Concerns about Navigator details.**
>
> Thank you for the consideration of the Navigator's design details.
>
> 1.We present the overall statistics on the Navigator's strategy choices below. Using the TSP-construction as an example, the table summarizes some observable regularities.
>
> | Mode     | run1 | run2 | run3 | avg.  |
> |----------|------|------|------|-------|
> | explore  | 110  | 95   | 102  | 102.3 |
> | exploit  | 81   | 88   | 82   | 83.7  |
> | balance  | 129  | 137  | 136  | 134.0 |
>
> From the evolutionary process, we observe the following general pattern. In Generation 1, the system has no historical signals and therefore uses a fully "balanced" search. In Generation 2, with limited data but no trigger for a forced switch, it adopts a probabilistic mix of "balance, exploration, and exploitation" ($\approx 2:1:1$). From Generations 3–8, the navigator becomes fully adaptive, with each generation's strategy determined directly by the outcomes of the previous one.
>
> 2.As you suggested, we conducted an additional experiment using the TSP-construction as an example in which the Navigator is fixed to "balance" to isolate the contribution of adaptive switching. The results of objective function value are presented below. As the table shown, the fixed-strategy variant exhibits significantly worse performance compared to our method with adaptive control. It clearly indicates that the adaptive mechanism in our design plays an essential and effective role.
>
> | Method                           | TSP10 | TSP20 | TSP50 | TSP100 | TSP200 |
> |----------------------------------|-------|-------|-------|--------|--------|
> | HiFo-Prompt (adaptive mechanism) | **1.654%** | **3.619%** | **6.625%** | **8.582%**  | **8.877%** |
> | HiFo-Prompt (fix balance)        | 5.812% | 9.676% | 15.599% | 18.924% | 22.115% |

---

### Official Review · Reviewer_44Vb · 2025-11-05

**Soundness:** 1
**Presentation:** 3
**Contribution:** 2
**Rating:** 2
**Confidence:** 4

**Summary:**

The paper presents HiFo-Prompt, a framework for LLM-based automated heuristic design that combines Hindsight, which builds an evolving Insight Pool of distilled design principles, and Foresight, an Evolutionary Navigator that adaptively balances exploration and exploitation. The method is applied to several optimization tasks (TSP, Online BPP, FSSP, and BO), and the authors report improvements over prior AHD methods in both solution quality and sample efficiency.

**Strengths:**

1. The motivation of this paper is clear and reasonable. The design ideas of global guidance and the insight pool are interesting and inspiring.
2. The similarity-based diversity discussion for the insight pool is conceptually stimulating.
3. The paper is clearly written and well organized, making it easy to follow.

**Weaknesses:**

1. I have concerns about the novelty threshold. The Insight Pool’s novelty filtering relies on Jaccard similarity over token sets. While this removes near-duplicate sentences, such a pure text-based comparison cannot capture semantic overlap. For example, one insight might be expressed in different ways. Since this novelty threshold is crucial for ensuring diversity, I worry this design may harm the actual effectiveness of the diversity mechanism.

2. The combination of a usage penalty and a recency bonus in $U(k, t)$ aims to balance exploration and exploitation, but the dynamics between these opposing terms are not analyzed. This could be sensitive in practice, and it would be helpful to justify or empirically demonstrate that this interaction leads to stable selection rather than oscillatory behavior. In particular, $w_u$ is a hyperparameter without ablation or sensitivity analysis, and the calculation of $B_r$ is not clearly presented. This reduces the soundness and reproducibility of the method.

3. The mapping from normalized performance $\tilde{\rho}$ to the effective credit $g_{\text{eff}}$ uses manually chosen piecewise constants (0.8, 0.6, 0.5, -0.3, etc.) with no theoretical justification or ablation. While the idea of tiered reward regimes is understandable, the specific scaling choices seem ad hoc and may not generalize across tasks. It would strengthen the work to at least provide hints or guidelines on how to select these values.

4. The definition of phenotypic diversity as the fraction of non-identical algorithm text strings feels coarse and potentially misleading. The measure is a bit lexical that two code snippets are treated as completely different even if they differ only by refactoring or variable renaming, ignoring actual semantic or functional similarity (similar with my commen in 2.). As a result, the system may overestimate diversity and trigger unnecessary exploration. Moreover, this approach scales as $O(|P|^2)$ comparisons per generation, which may become inefficient for larger populations and increase token consumption for LLM-based evaluations. The diversity threshold is also arbitrary and not justified or ablated. Overall, the lack of semantic grounding and unclear efficiency raises concerns about the robustness and practicality of the Navigator’s diversity control.

5. The experimental section raises several concerns about fairness, reproducibility, and efficiency. Although the paper states that all LLM-based baselines were evaluated under the same Qwen 2.5-Max model, implementation details and prompt adaptations are not provided, so fairness remains unclear (like baselines might use different LLMs, thus they can not be comparied directly). HiFo-Prompt’s runtime on small TSP instances (Table 1) is about an order of magnitude slower than competing methods, with no explanation, contradicting the claim of improved convergence speed. Token-usage statistics are summarized only coarsely (Appendix C.7) without breakdown or cost analysis, leaving uncertainty about true computational overhead. The brief multi-LLM comparison (Table 9) covers only two tasks and lacks analysis, providing little evidence of model generality. Finally, runtime behavior is inconsistent across tables (slower in TSP 10–50 but faster in TSP 100–500) with no explanation. Together, these issues make it difficult to assess the practical efficiency and generalizability of the proposed framework.

6. The code does not seem to be provided. Even though the authors share the core prompts, several computational details remain unclear, as mentioned in earlier points. This makes it hard to guarantee reproducibility and verify the soundness of the proposed method.

**Minors**

1. Very minor: for LaTeX quotation marks, please use the proper “…” format instead of plain double quotes. For example, in L055 the quotation marks are incorrectly formatted.
2. There are a few missing or incomplete citations in the appendix, such as at L811 and L1017. These should be corrected for completeness and consistency.

**Questions:**

See the weakness.

---

> ### Author Response · Authors · 2025-11-24
> **Response (1/6)**
>
> > **W1**: **Concerns about the novelty threshold.** The Insight Pool’s novelty filtering relies on Jaccard similarity over token sets. While this removes near-duplicate sentences, such a pure text-based comparison cannot capture semantic overlap. For example, one insight might be expressed in different ways. Since this novelty threshold is crucial for ensuring diversity, I worry this design may harm the actual effectiveness of the diversity mechanism.
>
> Thank you for raising this point regarding the potential limitations of Jaccard similarity. Although we acknowledge that token-based metrics may struggle with semantic overlap, we believe that our specific design choices make Jaccard both robust and necessary for this task.
>
> We justify this decision with the following reasons:
>
> **1.Consistency enables reliable Jaccard threshold**
>
> Our framework employs a standardized template during insight extraction, ensuring that identical concepts share consistent wording and structure. So, the situation that "one insight might be expressed in different ways" should not occur. Furthermore, the insights are not open-ended sentences but condensed design principles (avg. 20 words) composed of specialized terminology. Due to this enforced consistency and high information density, the high Jaccard threshold (0.7) reliably indicates conceptual repetition rather than just lexical overlap. Additionally, as described in Section 3.2, coupled with our credit assignment mechanism, semantically identical insights receive similar scores and are consequently eliminated through adaptive pruning.
>
> **2.Embedding metrics risk semantic smoothing**
>
> We also want to highlight that we avoided embedding-based metrics because they introduce a mismatch for LLM-based AHD. In our paper, insights are precise instructions where a single keyword change implies contradictory behavior (e.g., "Integrate dynamic weighting" vs. "Integrate static weighting"). Embedding models often suffer from "semantic smoothing", mapping structurally similar but functionally opposite sequences to nearby points [1, 2]. Such models struggle to distinguish sequences with low edit distances but high functional divergence. As a result, meaningful functional changes (like the example above) might be mistaken for redundancy and filtered out, limiting the framework's ability to explore different strategies.
>
> **3.Lightweight filtering supports high-frequency use**
>
> Insight filtering is a high-frequency operation executed thousands of times within the evolutionary loop. Our lightweight Jaccard approach ensures high sample efficiency without the significant latency or external API dependencies associated with semantic encoders.
>
> In summary, although our current implementation is simple, it is well-considered and both effective and reasonable within the field of LLM-based AHD. More importantly, the novelty of our method lies in the design of the mechanism (Hindsight and Foresight) rather than in the specific choice of this metric.
>
> [1] Agarwal D, Arivazhagan M G, Das R, et al. Searching for optimal solutions with LLMs via bayesian optimization. ICML, 2025.
>
> [2] Truong T H, Baldwin T, Verspoor K, et al. Language models are not naysayers: an analysis of language models on negation benchmarks. ACL, 2023.

---

> ### Author Response · Authors · 2025-11-24
> **Response (2/6)**
>
> > **W2: Concerns about utility function.** The combination of a usage penalty and a recency bonus in $U(k, t)$ aims to balance exploration and exploitation, but the dynamics between these opposing terms are not analyzed. This could be sensitive in practice, and it would be helpful to justify or empirically demonstrate that this interaction leads to stable selection rather than oscillatory behavior. In particular, $w_u$ is a hyperparameter without ablation or sensitivity analysis, and the calculation of $B_r$ is not clearly presented. This reduces the soundness and reproducibility of the method.
>
> Thank you for your attention to the design of the utility function. We provide a detailed breakdown of the design rationale and the formulation of the recency bonus below.
>
> 1.We analyze the **dynamic interplay** among the three constituents of $U(k, t)$ and show that the design intrinsically promotes stable, convergent selection rather than oscillation.
>
> - $E_{i}(t)$ (learned effectiveness) dominates the utility. By accumulating historical performance through an Exponential Moving Average (EMA), it provides a smooth, low-variance signal that reflects long-term value. In contrast, the usage penalty and recency bonus serve only as modulating factors among candidates with comparable $E_i(t)$.
>
> - The **usage penalty** ($log(N_{i}(t)+1)$) is chosen for its logarithmic growth. This ensures the penalty saturates quickly. Consequently, an insight's score does not plummet after a few uses. Instead, its priority is gently down-weighted, allowing other high-potential candidates a fair chance without causing sharp priority reversals or oscillatory behavior.
>
> - The **recency bonus** is formulated as a time-dependent function $B_r(t, t_i^{\text{last}})$ that applies a fixed reward of magnitude $\tau_r$ to recently validated insights. This encourages search continuity in promising directions.
>
> We conducted the following experiments on TSP50-construction to analyze the dynamic interplay. Our experiments show that only with both rewards and penalties can the algorithm properly balance exploration and exploitation. Without penalties, the search collapses prematurely into local optima and fails to explore sufficiently. Without rewards, it explores widely but inefficiently and cannot leverage new improvements.
>
> |                                               | run1  | run2  | run3  | avg.  |
> |-----------------------------------------------|-------|-------|-------|-------|
> | $w_u$ > 0, $\tau_r$ > 0 (default)                | 5.885 | 5.838 | 5.873 | **5.866** |
> | $w_u$ = 0, $\tau_r$ > 0 (w/o penalty)            | 5.958 | 5.989 | 5.978 | 5.975 |
> | $w_u$ > 0, $\tau_r$ = 0 (w/o reward)             | 5.954 | 5.972 | 5.979 | 5.968 |
> | $w_u$ = 0, $\tau_r$ = 0 (w/o reward and penalty) | 5.984 | 6.026 | 5.982 | 5.997 |
>
> 2.We provide the calculation of $B_r$ and explain parameters in detail.
>
> The Recency Bonus function, denoted as ($B_r(t, t_i^{\text{last}})$) is formally defined to capture temporal dynamics:
>
> $$
> B_r(t, t_i^{\\text{last}})=
> \\left\\{
> \\begin{array}{ll}
> \\tau_r & \\text{if } t - t_i^{\\text{last}} \\le T_w \\\\
> 0 & \\text{otherwise}
> \\end{array}
> \\right.
> $$
>
> + $B_r$ is the recency bonus for a specific insight $i$.
> + $t$ is the current generation of the algorithm.
> + $t_i^{\text{last}}$ is the generation in which insight $i$ was last used.
> + $T_w = 2$ is the window size threshold.
> + $\tau_r = 0.2$ is the bonus magnitude hyperparameter.
>
> In this configuration, a fixed reward of $\tau_r$ (ablation below) is assigned only if the insight was utilized within the recent window defined by $ T_w$. This design enables the framework to adapt to the dynamic nature of the search process by prioritizing insights that align with the current evolutionary stage. Theoretically, this formulation mirrors the sliding-window mechanism in non-stationary Multi-Armed Bandits (e.g., SW-UCB [1]). Our $B_r$ serves as a lightweight adaptation of this principle to account for the time-varying effectiveness of insights.
>
> 3.Finally, we present the experimental results of a sensitivity analysis on the parameter $w_u$ and $\tau_r$ (in $B_r$), taking TSP50-construction as an example.
>
> | Setting                 | run1  | run2  | run3  | avg.  |
> |-------------------------|-------|-------|-------|-------|
> | $w_u$ = 0               | 5.958 | 5.989 | 5.978 | 5.975 |
> | $w_u$ = 0.05            | 5.894 | 5.901 | 5.907 | 5.901 |
> | $w_u$ = 0.1 (default)   | 5.885 | 5.838 | 5.873 | **5.866** |
> | $w_u$ = 0.3             | 5.921 | 5.907 | 5.922 | 5.917 |
> | $\tau_r$ = 0            | 5.954 | 5.972 | 5.979 | 5.968 |
> | $\tau_r$ = 0.1          | 5.920 | 5.925 | 5.901 | 5.915 |
> | $\tau_r$ = 0.2 (default)| 5.885 | 5.838 | 5.873 | **5.866** |
> | $\tau_r$ = 0.4          | 5.937 | 5.945 | 5.933 | 5.938 |
>
> [1] Garivier, A., \& Moulines, E. On Upper-Confidence Bound Policies for Switching Bandit Problems. ALT, 2011.

---

> ### Author Response · Authors · 2025-11-24
> **Response (3/6)**
>
> > **W3: Concerns about credit-assignment mechanism.** The mapping from normalized performance $\tilde{\rho}$ to the effective credit $g_{\text{eff}}$ uses manually chosen piecewise constants (0.8, 0.6, 0.5, -0.3, etc.) with no theoretical justification or ablation. While the idea of tiered reward regimes is understandable, the specific scaling choices seem ad hoc and may not generalize across tasks. It would strengthen the work to at least provide hints or guidelines on how to select these values.
>
> Thank you for your questions on our credit-assignment mechanism. We will clarify its design principles and rationale.
>
> **1.Interpretable piecewise rewards**
>
> Our piecewise reward function draws upon techniques for handling sparse rewards in Hierarchical Reinforcement Learning (HRL). Unlike continuous functions, this structure provides discrete and explicit signals for distinct performance levels. The primary advantage is interpretability: it clearly defines the thresholds for rewards and penalties, rather than obscuring them within a complex continuous formulation.
>
> **2.Rationale for scaling choices**
>
> In addition, we clarify the motivation for the specific scaling choices.
>
> The strategy for choosing the **base rewards** can be summarized as: Elite Reward $\gg$ |Penalty| $\gtrsim$ Incremental Reward. The elite reward (0.8) provides a strong signal for breakthroughs. The incremental reward (0.2) acknowledges smaller gains, while its 4× gap from 0.8 emphasizing paradigm discovery over refinement. The penalty (–0.3), larger in magnitude than 0.2, more effectively prunes unproductive directions. This asymmetric design improves search efficiency.
>
> For the **coefficients**, our guideline is: Promotion Sensitivity > Pruning Sensitivity > Saturation Sensitivity. This follows a "$\land$-shaped" sensitivity design, where the intermediate tier (i.e., promising states) receives the largest gradient to encourage rapid upward transitions, while the top tier (elite states) is assigned low sensitivity to maintain stability.
>
> In fact, the base rewards matter much more than the small changes introduced by the coefficients. To validate the robustness of our method and address the reviewer's concerns regarding parameter choices, we selected the TSP50-construction as an example and tested multiple combinations of base reward constants around our current settings [0.8, 0.2, -0.3] (elite reward, incremental reward, penalty).
>
> | Setting                   | run1  | run2  | run3  | avg.  |
> |---------------------------|-------|-------|-------|-------|
> | [0.8, 0.2, 0]             | 6.046 | 5.976 | 5.998 | 6.007 |
> | [1.0, 0.1, -0.5]          | 5.921 | 5.932 | 5.902 | 5.918 |
> | [6, 0.3, -0.1]            | 5.916 | 5.917 | 5.915 | 5.916 |
> | [0.8, 0.2, -0.3] (default)| 5.885 | 5.838 | 5.873 | **5.866** |
>
> **3.Cross-Task consistency**
>
> Finally, we would like to emphasize that in all the main experiments presented in this paper (including TSP, FSSP, and BPP), we consistently used the same set of segmented constants. Our experimental results demonstrate that our framework achieves competitive performance across different tasks. This provides the most direct evidence of the generalization capability of the proposed reward mechanism.

---

> ### Author Response · Authors · 2025-11-24
> **Response (4/6)**
>
> > **W4: Concerns about phenotypic diversity, computational resources and diversity threshold.** The definition of phenotypic diversity as the fraction of non-identical algorithm text strings feels coarse and potentially misleading. The measure is a bit lexical that two code snippets are treated as completely different even if they differ only by refactoring or variable renaming, ignoring actual semantic or functional similarity (similar with my commen in 2.). As a result, the system may overestimate diversity and trigger unnecessary exploration. Moreover, this approach scales as $O(|P|^2)$ comparisons per generation, which may become inefficient for larger populations and increase token consumption for LLM-based evaluations. The diversity threshold is also arbitrary and not justified or ablated. Overall, the lack of semantic grounding and unclear efficiency raises concerns about the robustness and practicality of the Navigator’s diversity control.
>
> Thank you for the feedback regarding the phenotypic diversity metric and its efficiency. We address these concerns as follows:
>
> **1.Phenotypic diversity**
>
> We want to emphasize that our purpose is to preserve fine-grained logical distinctions, rather than to measure broad semantic similarity. As we noted in our response to **Weakness 1**, a minor textual change may imply a completely contradictory logic. Semantic embedding models will introduce a mismatch for LLM-based AHD. Our exact string matching guarantees that we do not miss any subtle modification that could help the system escape a local optimum.
>
> **2.Diversity threshold**
>
> We emphasize that the diversity threshold is designed to anticipate the onset of a local optimum. We found that as the search nears such a point, the LLM's outputs collapse to a single logic and repeatedly generate the same solution. To detect this, we enforce a strict template that standardizes outputs and removes trivial noise (e.g., variable renaming or wording changes) that would otherwise inflate diversity.
>
> In this setting, we do not need to measure semantic similarity. The key signal is simply the high frequency of identical text strings. After removing superficial variation, any remaining textual difference reflects a genuine logical adjustment, which must be preserved because even small refinements can help the LLM escape local optima. Thus, a diversity score below 0.3 reliably indicates that the search has truly stopped producing new ideas.
>
> Moreover, we also want to mention our **parameter sensitivity analysis** of the diversity threshold in **Figure 4**. This empirical evidence demonstrates that the framework maintains robust performance across a threshold range of 0.1 to 0.5. This confirms that the presence of the redundancy check is more critical than the precise numerical value.
>
> **3.Computational resources**
>
> It is important to note that there is a misunderstanding regarding computational resources. The diversity computation and evaluations are purely local operations. It involves zero LLM API calls and consumes no tokens.
>
> Regarding algorithmic complexity, we intentionally maintain a small population size of 8 to prioritize rapid convergence. This results in negligible runtime overhead compared to the dominant costs of LLM inference.

---

> ### Author Response · Authors · 2025-11-24
> **Response (5/6)**
>
> > **W5: Concerns about experimental results and analysis.**
>
> Thank you for these questions. We address the concerns as follows.
>
> **1.Implementation details and fairness**
>
> We strictly adhered to the principles of fair comparison throughout our experimental design. To eliminate potential bias from model capabilities, both HiFo-Prompt and all LLM-based baselines utilized the same Qwen 2.5-Max model for heuristic generation. Regarding implementation details, we adopted the official open-source code for all baselines without modifying their internal workflows. Specifically, we maintained consistency in the input and output prompt templates with EoH [2] for all problem instances. For the specific implementation of our dynamic prompt synthesis, detailed examples are provided in Figure 2.
>
> **2.Runtime comparison**
>
> We clarify that the time in Table 1 refers to inference runtime of the generated heuristics, while "convergence speed" measures search time during training. Slower inference on small TSP instances is expected due to the more complex heuristics our method produces for higher solution quality, compared with simpler, faster-executing baselines. This is not contradictory. As Table 10 shows, our approach converges faster during training, generating high-quality heuristics more quickly. In LLM-based AHD, the main goal is achieving superior algorithms in shorter design time.
>
> **3.Cost analysis**
>
> We emphasize that in current LLM-based AHD, the main computational bottleneck is the costly heuristic evaluation during the evolutionary process, not LLM API calls. Thus, total running time, rather than token cost, is the key metric for efficiency. For token cost, we follow standard practices: studies like MCTS-AHD[3] and MoH[1] report similar summaries (Table 10). Our results show that our method uses fewer LLM resources than baselines, leading to shorter training time and lower overall computational cost.
>
> **4.Model generality**
>
> To present these results more clearly and address the reviewer's concern about model generality, following the Table in MCTS-AHD[3], we reorganized this part by reporting results across more tasks and more scales.
>
> | Method| TSP50-construction | TSP100-construction | TSP100-GLS | TSP200-GLS | Online BPP 1k,100 | Online BPP 10k,300 | FSSP-GLS j100 m10 | FSSP-GLS j200 m20 |
> |-|-|-|-|-|-|-|-|-|
> | Qwen-2.5-max| 6.625%  | 8.582%| 0.015%| 0.382%| 2.188%| 0.391%| 0.128%| 0.710%|
> | GPT-4o-mini| 7.058%  | 9.579%| 0.021%| 0.433%| 2.287%| 0.492%| 0.137%| 0.787%|
> | DeepSeek-v3| 7.717% | 10.455%| 0.041%| 0.480%| 2.238%| 0.978%| 0.185%| 0.916%|
> | Deepseek-r1| 10.412% | 12.638%| 0.074%| 0.589%| 2.984%| 1.487%| 0.313%| 1.485%|
> | claude3.5-sonnet | 8.276%| 11.087%| 0.061%| 0.454%| 2.089%| 1.487%| 0.179%| 1.256%|
>
> In addition, we note that our focus is on developing the evolutionary framework rather than benchmarking LLMs. Following prior LLM-based AHD work, our goal is simply to show that the framework is model-agnostic and remains robust across LLMs with different reasoning capabilities.
>
> **5.Differences between paradigms**
>
> The runtime differences arise from two distinct solving paradigms. Small-scale experiments (TSP 10–50, Table 1) use the TSP-construction setting, where heuristics build solutions from scratch with sophisticated logic, incurring higher overhead than simpler baselines. Large-scale experiments (TSP 100–500, Table 2) use the TSP-GLS setting, where LLM-generated components guide perturbations or evaluations within the efficient Guided Local Search framework, enabling faster execution and stronger performance. These differing paradigms explain the observed runtime variations.
>
> To further substantiate this explanation, we provide additional experimental results below, specifically applying the TSP-construction paradigm to larger instances (TSP 100 and 200) for direct comparison.
>
> | Method    | TSP100 Gap | Time(s)        | TSP200 Gap | Time(s)        |
> |-----------|------------|----------------|------------|----------------|
> | LKH3      | 0.000%     | 1450| 0.000%     | 6312           |
> | POMO      | 1.636%     | -| 13.961%    | -              |
> | LEHD      | 0.452%     | -| 0.367%     | -              |
> | EoH       | 15.361%    | 9| 16.658%    | 78             |
> | ReEvo     | 12.577%    | 224| 14.890%    | 3013           |
> | HSEvo     | 12.008%    | 1286| 13.578%    | 24835          |
> | MCTS-AHD  | 12.521%    | 1084| 13.510%    | 14521          |
> | Ours      | 8.582%     | 1843| 8.877%     | 16099          |
>
> [1] Shi Y, Zhou J, Song W, et al. Generalizable Heuristic Generation Through Large Language Models with Meta-Optimization. arXiv preprint arXiv:2505.20881, 2025.
>
> [2] Liu F, Tong X, Yuan M, et al. Evolution of heuristics: Towards efficient automatic algorithm design using large language model. ICML, 2024.
>
> [3] Zheng Z, Xie Z, Wang Z, et al. Monte carlo tree search for comprehensive exploration in llm-based automatic heuristic design. ICML, 2025.

---

> ### Author Response · Authors · 2025-11-24
> **Response (6/6)**
>
> > **W6**: The code does not seem to be provided. Even though the authors share the core prompts, several computational details remain unclear, as mentioned in earlier points. This makes it hard to guarantee reproducibility and verify the soundness of the proposed method.
>
> We appreciate your attention to the reproducibility of our method. We will publish all the code after the paper is published to show the detailed implementation.
>
> > **Minors**
> > 1. Very minor: for LaTeX quotation marks, please use the proper “…” format instead of plain double quotes. For example, in L055 the quotation marks are incorrectly formatted
> > 2. There are a few missing or incomplete citations in the appendix, such as at L811 and L1017. These should be corrected for completeness and consistency.
>
> We also sincerely thank the reviewer for carefully reviewing our manuscript. We apologize for the oversights in symbols and references, and we will thoroughly recheck the entire paper to correct these issues in the final version.

---

### Author Response · Authors · 2025-12-02
**Summary of Rebuttal**

Dear SAC/AC,

Thank you for your time and effort in organizing the review of our submission and reviewing our work. We also thank all reviewers for their constructive feedback and recognition of our work. The reviewers commended our method for its clear motivation, its novel dual-module architecture (Hindsight & Foresight), and its superior performance with high sample efficiency.

We summarized several key questions raised by the reviewers and our corresponding responses as follows.

1.**Justification of Diversity Metrics** (Reviewer 44Vb, 9fSr): We explained that the design of our diversity metrics is both reasonable and efficient. It is more robust than semantic embeddings in LLM-based AHD. We explained that subtle lexical changes (e.g., "dynamic" vs "static") often imply contradictory logic in code generation, which embedding-based smoothing might incorrectly filter out.

2.**Differentiation from Related Work** (Reviewer xien, 9fSr): As suggested by the reviewers, we conducted additional discussion and comparison with related works on prompt evolution and hierarchical search, such as MeLA, LLM-LNS and EvoPH. We also compared our method with thoughts and reflections mechanisms used in EoH and ReEvo. Our method implements cumulative knowledge evolution (vs. ReEvo's transient reflection) and global state-based control (vs. local/reactive gradients), ensuring long-term learning and avoiding local optima.

3.**Robustness of Hyperparameters** (Reviewer 44Vb, xien, 99Cj): We provided comprehensive sensitivity analyses for key parameters (e.g., decay rate, reward constants, diversity threshold). Results demonstrate that our hyperparameters settings are robust. We also provided a theoretical explanation to address reviewers' questions.

4.**Clarification on "Seed Insights"** (Reviewer xien): To address the reviewers' concerns, we conducted new ablation studies showing that injecting our seed insights into four baselines (EoH, ReEvo, HSEvo, MCTS-AHD) leads to only negligible improvements. This confirms that our method's performance arises from its self-evolving knowledge mechanism rather than from the initial hand-crafted prompts.

5.**Generalization across LLMs & Tasks** (Reviewer 44Vb, GjDu): We added extensive experiments across 5 different LLMs (GPT-4o, Claude-3.5, DeepSeek, etc.) and benchmark datasets (TSPLib). The results confirm that HiFo-Prompt is model-agnostic and generalizes effectively across diverse tasks.

6.**Efficiency & Computational Cost** (Reviewer 44Vb, GjDu): We clarified the distinction between training search time and inference time. We further provided a token cost analysis, proving that HiFo-Prompt achieves SOTA results with significantly fewer token evaluations than baselines.

7.**Mechanism Interpretability** (Reviewer GjDu, 9fSr): We provided concrete examples of generated heuristics and Navigator statistics (exploration/exploitation frequency). New ablations (e.g., fixing Navigator to "balance") confirmed the necessity of the adaptive switching mechanism.

8.**Reproducibility & Clarity**: We corrected the typo regarding the maximization/minimization objective and committed to releasing the full code. We also updated the manuscript and corrected minor citation errors.

Best Regards,

Authors

---

### Meta-Review · Area_Chair_B3QV · 2025-12-10

**Summary:**

Thank you to the reviewers for your valuable suggestions from multiple perspectives. Overall, I think their main problems at present lie in:



- The soundness of the core mechanism is not guaranteed.

- Novelty is overstated.

- The details are unclear.

- There are contradictions in the experimental results.

- The experimental results need to be further supplemented.



In addition, some reviewers mentioned issues such as reproducibility.

**Reviewer Concerns:**

I am very grateful for the reply provided by the author. I believe that some of the reviewers' questions will be resolved, such as some explanations regarding the experimental details.



However, the concerns of some reviewers regarding the novelty of the method may not be resolved.



Overall, considering the potential score increase, I believe this paper can convince most reviewers and is above the acceptance threshold.

**Reviewer Scores:**

For the Reviewer 44Vb, he may accept the explanation of the experimental details, but doubts about the specific details of the method may still exist. I think this reviewer will increase the score (**Rating:** 2 to 4).

For Reviewer GjDu, he may accept most of the explanations. I think the reviewer will maintain the current score (**Rating:** 6).

For Reviewer xien, he may agree with the explanation about the experimental setup and results, but the explanation for novelty may not convince him. I think the reviewer will maintain the current score (**Rating:** 4).

For Reviewer 99Cj, he may accept an explanation about hyperparameter setting. I think this reviewer will increase the score (**Rating:** 4 to 6).

For Reviewer 9fSr, he may accept most of the explanation. I think this reviewer will maintain the score (**Rating:** 6).

---

### Decision · Program_Chairs · 2026-01-26

Accept (Poster)